# Dynamic Correction of Erroneous State Estimates via Diffusion Bayesian Exploration

## Abstract

In emergency response and other high-stakes societal applications, early-stage state estimates critically shape downstream outcomes. Yet, these initial state estimates—often based on limited or biased information—can be severely misaligned with reality, constraining subsequent actions and potentially causing catastrophic delays, resource misallocation, and human harm. Under the stationary bootstrap baseline (zero transition and no rejuvenation), bootstrap particle filters exhibit Stationarity-Induced Posterior Support Invariance (S-PSI), wherein regions excluded by the initial prior remain permanently unexplorable, making corrections impossible even when new evidence contradicts current beliefs. While classical perturbations can in principle break this lock-in, they operate in an always-on fashion and may be inefficient. To overcome this, we propose a diffusion-driven Bayesian exploration framework that enables principled, real-time correction of early state estimation errors. Our method expands posterior support via entropy-regularized sampling and covariance-scaled diffusion. A Metropolis–Hastings check validates proposals and keeps inference adaptive to unexpected evidence. Empirical evaluations on realistic hazardous-gas localization tasks show that our approach matches reinforcement learning and planning baselines when priors are correct. It substantially outperforms classical SMC perturbations and RL-based methods under misalignment, and we provide theoretical guarantees that DEPF resolves S-PSI while maintaining statistical rigor.

## 1 Introduction

In disaster emergency management, early decisions play a pivotal role in shaping the outcome of a crisis Steel (2010); Kruschke (2010). An initial state estimate—for example, an early guess of a hazardous leak's location—is often made with scant data under intense time pressure. Such early estimates carry great weight in guiding subsequent actions, yet they are highly vulnerable to error due to uncertainty and human subjective judgment. If the initial assumption is mistaken, the entire response effort can become locked into a false premise, failing to adjust even as new data arrives. This lock-in effect in high-stakes scenarios leads to catastrophic delays, misallocation of resources, and heightened risks to society. The fragility of such initial state assumptions thus poses a serious challenge for social decision-making processes, revealing the need for methods that can rapidly correct initial mistakes in real time.

One major source of initial state estimation error is the inherent uncertainty and bias in early-stage human assessments. When decision-makers narrow their focus based on incomplete or misleading information, they risk excluding the true state of the world from consideration. Once an erroneous initial state estimate anchors the process, it creates a path-dependent trap: monitoring systems and responders may remain fixated on the wrong location or strategy, even as contradictory evidence mounts. In practice, this means a search algorithm could ignore critical regions outside the presumed hazard zone, or resources might continue to be deployed ineffectively, compounding the crisis's social and economic impacts. To avoid such outcomes, it is crucial to develop adaptive algorithms that revisit and revise early assumptions as new observations arrive.

From a robotics and Bayesian inference perspective, the above scenario is essentially a state estimation problem: an agent must form an initial belief (prior) about the latent state (e.g., the hazard's location) and update this belief as new sensor data arrives. If the initial belief is mis-specified, the goal

is to correct this incorrect state estimate in real time by incorporating incoming evidence. For clarity, we use the term initial policy error to denote such a mistaken initial state estimate under prior uncertainty, which our approach aims to correct. In recent years, particle filtering (PF) has become a go-to approach for sequential Bayesian state estimation in these settings Ristic (2013); Gordon et al. (1993b); Doucet et al. (2001a). PF offers a principled way to update beliefs by fusing sensor data with prior knowledge, making it a representative framework to tackle the problem of erroneous initial state estimates in emergency response and other high-stakes decision domains. However, existing Bayesian filtering methods Fox et al. (2003); Smidl & Quinn (2008) struggle with erroneous initial state estimates, often failing to recover when the true state lies outside the initial belief. 1) Bootstrap particle filters Candy (2007); Gordon et al. (1995) tend to remain confined to the support of the initial prior, hindering exploration beyond the originally assumed region when operated under a stationary bootstrap baseline with zero transition and no rejuvenation. We formalize this baseline-specific lock-in as Stationarity-Induced Posterior Support Invariance (S-PSI): under the zero-transition, no-rejuvenation baseline, the posterior cannot escape the initial support. Importantly, S-PSI is not an inherent limitation of PF; classical countermeasures (e.g., jittering, roughening) can in principle expand support (we include these as baselines in our experiments). 2) More advanced PF variants,such as auxiliary particle filters Mountney et al. (2009); Branchini & Elvira (2021) or filters with optimal proposal distributions, can partially mitigate bias when the true state has a small non-zero prior probability Fox (2001); Douc & Cappé (2005); Liu & Chen (1998); Doucet et al. (2000); Arulampalam et al. (2002), but they fundamentally cannot handle cases where the true state was assigned zero initial probability. Under the standard Bayesian update, any region with zero prior mass will remain at zero posterior mass indefinitely, meaning the algorithm can never discover a completely excluded possibility under this baseline. 3) Attempts to address these issues by augmenting particle filters have had limited success. Some works inject noise or broaden the prior artificially, and others integrate reinforcement learning (RL) with PF Shi et al. (2024); Zhao et al. (2022); Park et al. (2022b) to actively guide sensor exploration. While such approaches can improve data collection, they may inherit the same blind spots from a mis-specified prior and often introduce significant complexity and resource demands. Without a new perspective, Bayesian trackers and decision methods remain at risk of locking onto an incorrect initial belief, especially under the S-PSI.

To overcome these challenges, we propose a novel approach called Diffusion-Enhanced Particle Filtering (DEPF) that dynamically corrects erroneous initial state estimates via a diffusion-driven Bayesian exploration mechanism. The key insight of DEPF is to expand the particle filter's support in response to observation feedback, allowing the algorithm to break out of the constraints imposed by a flawed initial prior. Instead of passively accepting the prior's limits, our method systematically injects a small number of exploratory particles into regions outside the currently believed range. This injection is guided by indicators of model inconsistency when incoming sensor data strongly contradicts the filter's predictions (e.g., high error or entropy). A controlled stochastic diffusion process then spreads these exploratory particles into previously neglected areas, effectively probing the hypothesis that the true state might lie beyond the old bounds. We incorporate a Bayesian validation step to ensure that the expanded support remains statistically coherent. Through this belief-triggered diffusion-and-validation cycle, DEPF augments the PF inference layer and mitigates S-PSI when it arises under the stationary bootstrap baseline.

The main contributions of this work are as follows: (1) We identify and formally define the Stationarity-Induced Posterior Support Invariance (S-PSI) under the zero-transition, no-rejuvenation bootstrap baseline, characterizing it as a diagnostic condition rather than a universal PF limitation (§3.4). (2) We propose the DEPF framework, a particle filtering method that introduces a principled, belief-triggered technique to dynamically expand inference support beyond initial belief constraints (§4). (3) We demonstrate via theory and experiments (hazardous gas leak scenarios) that DEPF can effectively correct initial state estimation errors across different scales and error severities, substantially improving localization and response efficiency over existing methods, including RL/planning baselines and classical perturbations (§5).

## 2 RELATED WORK

In emergency localization scenarios Wu et al. (2021); Hite (2019), *Bayesian filtering* Fox et al. (2003); Smidl & Quinn (2008); Quinlan & Middleton (2010) methods such as the bootstrap particle filter leverage Bayesian inference to iteratively update state estimates but typically assume correctly-

specified initial priors, thus *may become ineffective under severely misaligned early assumptions when operated under a stationary bootstrap baseline (zero transition, no rejuvenation), due to Stationarity-Induced Posterior Support Invariance (S-PSI)* Gordon et al. (1995); Candy (2007). *S-PSI is a baseline-specific diagnostic rather than an inherent limitation of PF; classical countermeasures such as jittering/roughening or resample–move can, in principle, expand support, and we include them as baselines in §5.* Advanced particle filter variants, including *auxiliary particle filters* Mountney et al. (2009); Branchini & Elvira (2021) and filters using *optimal proposal distributions*, improve proposal quality and sample efficiency but still fail when the initial belief assigns *zero* prior probability to the true state Fox (2001); Arulampalam et al. (2002), i.e., in the presence of a zero-prior barrier without explicit support expansion. Meanwhile, *information-theoretic methods* like *Infotaxis* Vergassola et al. (2007), *Entrotaxis* Hutchinson et al. (2018b) and *DCEE* Chen et al. (2021) focus sensor motions on maximizing expected information gain or reducing entropy, yet they typically operate within the belief support induced by the initial prior and thus cannot systematically correct severe prior misalignment in real time without a support-expanding inference layer. More recently, integrated *reinforcement learning and particle filtering (RL-PF)* methods have emerged—e.g., *AGDC* and its variants using KL-divergence or entropy-based intrinsic rewards Shi et al. (2024), *PC-DQN* Zhao et al. (2022), and *GMM-PFRL* Park et al. (2022b). While these RL-driven methods exhibit stronger adaptive exploration, they can still inherit the zero-prior barrier when the underlying filtering layer does not expand support (cf. the S-PSI baseline). In contrast, our proposed *DEPF* explicitly addresses this gap by introducing a belief-triggered, validated support-expansion mechanism that operates at the inference layer, making it complementary to proposal-improvement filters, classical perturbations (jittering/roughening/rejuvenation), and planning/RL controllers, and yielding superior robustness under severely misaligned early assumptions.

## 3 PROBLEM FORMULATION AND PRELIMINARIES

### 3.1 PROBLEM SETUP

Consider a two-dimensional spatial domain $\Omega \subset \mathbb{R}^2$ with a stationary hazardous gas source. We describe the unknown source term by the parameter vector at time step $k$: $\Theta_k = [x_s, y_s, q_s, u_s, \phi_s, d_s, \tau_s]^\top \in \mathbb{R}^7$ where $\boldsymbol{p}_s = (x_s, y_s) \in \Omega \subset \mathbb{R}^2$ represent the Cartesian coordinates of the source, $q_s \in \mathbb{R}^+$ denotes the scaled release strength, representing the true emission rate adjusted by an unknown sensor calibration factor, $u_s \in \mathbb{R}^+$ and $\phi_s \in [0, 2\pi)$ represent the wind speed and wind direction respectively, $d_s \in \mathbb{R}^+$ describes the diffusivity of the gas in air, $\tau_s \in \mathbb{R}^+$ indicates the effective lifetime of the gas. At each discrete time step $k$, a mobile robot equipped with a gas sensor occupies position $\boldsymbol{p}_k = (x_k, y_k) \in \Omega$ and records a scalar sensor output $z_k \in \mathbb{R}^+$, which represents the raw *voltage signal* from the gas sensor. This signal serves as the *observation* in the Bayesian filtering model, linking sensor data to the hidden source state $\Theta$. The cumulative sensor readings up to step $k$ are denoted as $z_{1:k} = \{z_1, z_2, \ldots, z_k\}$. **Our objective** is to estimate the posterior distribution $p(\Theta_k \mid z_{1:k})$ given the observed sensor signals $z_{1:k}$ (i.e., raw voltage measurements), and robot locations, under the assumption of source stationarity, i.e., $\Theta_{k+1} = \Theta_k$. To handle nonlinearities and intermittency in sensor readings, a particle filter is adopted to iteratively approximate this posterior.

### 3.2 OBSERVATION MODEL

We adopt a simplified analytical plume model derived from the advection–diffusion equation to represent gas transport from the source to the sensor location. The expected sensor output at location $p_k$ under source parameters $\Theta$ is given by: $h(p_k; \Theta) = \frac{q_s}{4\pi d_s \|p_k - p_s\|} \cdot \exp\left(-\frac{\|p_k - p_s\|}{\lambda} - \frac{\psi}{2d_s}\right)$ with $\psi = (x_k - x_s)u_s \cos\phi_s + (y_k - y_s)u_s \sin\phi_s$ and $\lambda = \sqrt{d_s \tau_s / [1 + (u_s^2 \tau_s / 4d_s)]}$.

Since we use a low-cost metal oxide (MOX) sensor, the sensor output is not a calibrated concentration but a voltage value subject to significant uncertainty and miss-detection. Thus, the final measurement model is: $z_k = D_k \cdot \left(h(p_k; \Theta) + \bar{v}_k\right) + (1 - D_k) \cdot v_k$. where $\bar{v}_k \sim \mathcal{N}(0, \bar{\sigma}_k^2)$ is additive sensor noise, variance $\bar{\sigma}_k^2$, $v_k \sim \mathcal{N}(0, \sigma_k^2)$ is background noise in clean air, variance $\sigma_k^2$, $D_k \in \{0, 1\} \sim$ Bernoulli$(P_d)$ encodes whether the sensor successfully detects the gas, $P_d$ is the probability of detection, reflecting turbulence, dilution, or sensor failure. The resulting *Gaussian mixture likelihood function* becomes: $p(z_k \mid \Theta) = (1 - P_d) \cdot \mathcal{N}(z_k; 0, \sigma_k^2) + P_d \cdot \mathcal{N}(z_k; h(p_k; \Theta), \bar{\sigma}_k^2)$

### 3.3 SEQUENTIAL PARTICLE FILTERING

Particle filtering offers a non-parametric, sequential Bayes estimator of the posterior $p(\Theta_k \mid z_{1:k})$. We represent that posterior by a set of $N$ weighted particles $\{\Theta_k^{(i)}, w_k^{(i)}\}_{i=1}^N$, $p(\Theta_k \mid z_{1:k}) \approx \sum_{i=1}^N w_k^{(i)} \delta(\Theta_k - \Theta_k^{(i)})$, $\sum_{i=1}^N w_k^{(i)} = 1$, with $\delta(\cdot)$ the Dirac delta. In general, particles propagate via a transition kernel $p(\Theta_k \mid \Theta_{k-1})$ and are sampled from a proposal $q(\Theta_k \mid \Theta_{k-1}, z_k)$. The importance weights then update as $\tilde{w}_k^{(i)} = w_{k-1}^{(i)} \frac{p(z_k \mid \Theta_k^{(i)}) \, p(\Theta_k^{(i)} \mid \Theta_{k-1}^{(i)})}{q(\Theta_k^{(i)} \mid \Theta_{k-1}^{(i)}, z_k)}$, $w_k^{(i)} = \tilde{w}_k^{(i)} / \sum_{j=1}^N \tilde{w}_k^{(j)}$, with the likelihood $p(z_k \mid \Theta)$ given in §3.2. We trigger resampling when the effective sample size $\mathrm{ESS}_k = 1 / \sum_{i=1}^N (w_k^{(i)})^2$ falls below a threshold $\eta$. In the widely used *bootstrap filter*, the proposal equals the transition, $q(\Theta_k \mid \Theta_{k-1}, z_k) = p(\Theta_k \mid \Theta_{k-1})$, so the weight update simplifies to $\tilde{w}_k^{(i)} = w_{k-1}^{(i)} p(z_k \mid \Theta_k^{(i)})$. In our setting the source term $\Theta$ is assumed *static during the response horizon*. Without a natural dynamical law, a common and widely used reference method is the *bootstrap filter*, where particles are simply carried forward and only their weights are updated by the likelihood of new sensor data.

### 3.4 STATIONARITY-INDUCED POSTERIOR SUPPORT INVARIANCE (S-PSI)

The simplicity of the bootstrap filter, while natural under static parameters, reveals a structural vulnerability: particles remain fixed in parameter space and can never leave the initial prior region. Even when new observations strongly contradict the prior, the filter cannot escape this confinement. We formalize this lock-in effect as *Stationarity-Induced Posterior Support Invariance (S-PSI)*.

Let the prior be $p_0(\Theta)$ with support $\mathcal{S}_{\mathrm{prior}} := \mathrm{supp}\, p_0(\Theta) = \{\Theta : p_0(\Theta) > 0\} \subset \mathbb{R}^7$.

**Assumption 3.1** (S0: zero transition, no rejuvenation). $p(\Theta_k \mid \Theta_{k-1}) = \delta(\Theta_k - \Theta_{k-1})$ and no rejuvenation step (e.g., jittering, roughening, resample–move) is applied.

**Proposition 3.2** (S-PSI under O.1). *If particles are initialized within $\mathcal{S}_{\mathrm{prior}}$, then for all $k$, $\mathrm{supp}(p(\Theta \mid z_{1:k})) \subseteq \mathcal{S}_{\mathrm{prior}}$. In words, the posterior support remains permanently trapped inside the initial prior region. As a direct consequence, if the true source $\Theta^*$ lies outside the prior, then $\Theta^* \notin \mathcal{S}_{\mathrm{prior}} \Rightarrow p(\Theta^* \mid z_{1:k}) = 0$, $\forall k$, i.e., the filter fundamentally cannot discover it, not due to likelihood mismatch, but simply because no particles ever enter that excluded region.*

### 3.5 POMDP FORMULATION FOR BELIEF-AWARE SENSOR PLANNING

The interaction between the mobile robot and the unknown gas plume can be framed as a *partially observableMarkov decision process* (POMDP) $\mathcal{M} = (\mathcal{S}, \mathcal{A}, \Omega, T, O, R, \gamma)$, where $\mathcal{S}$ is the latent state space containing the stationary source vector $\Theta = (x_s, y_s, q_s, u_s, \phi_s, d_s, \tau_s)^\top$; $\mathcal{A}$ is the set of motion commands that move the robot in the plane; $\Omega$ is the observation space, where an observation at time $k$ is $o_k = (p_k, z_k)$ with $p_k \in \mathbb{R}^2$ the robot position and $z_k \in \mathbb{R}$ the sensor voltage. The transition kernel factorises as $T((p', \Theta') \mid (p, \Theta), a) = T_p(p' \mid p, a)\, \delta(\Theta' - \Theta)$: robot kinematics are deterministic while the source parameters remain unchanged. The observation model is $O(o_k \mid (p_k, \Theta), a_{k-1}) = p(z_k \mid p_k, \Theta)$, where the likelihood $p(z_k \mid p_k, \Theta)$ is the mixture-Gaussian plume sensor model of §3.2. At decision time the agent cannot access $\Theta$ directly, so it reasons with the belief $b_k = p(\Theta \mid z_{1:k})$ supplied by the particle filter. We therefore feed the RL policy with the augmented information state $s_k^{\mathrm{RL}} = (p_k, z_k, b_k)$. The reward is chosen as the expected one-step information gain $R_k = \mathbb{E}_{o_{k+1}}[D_{\mathrm{KL}}(b_{k+1} \parallel b_k)]$, encouraging actions that shrink posterior uncertainty, and future rewards are discounted by $\gamma \in (0, 1)$. The objective is to learn a policy $\pi^*$ that maximises the expected discounted return $J = \mathbb{E}[\sum_{t=0}^\infty \gamma^t R_{k+t}]$, thereby steering the robot along paths that are most informative about the hidden source. The details are provided in Appendix § P.

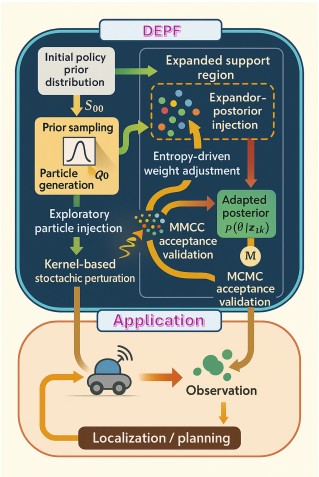

Figure 1: Flowchart of DEPF.

## 4 DIFFUSION-DRIVEN SUPPORT RANGE EXPANSION

Particle filtering provides a sequential Bayesian framework with weighted particles approximating the posterior. Under the *S-PSI baseline* introduced in §C—i.e., a **zero-transition, no-rejuvenation** bootstrap setting—the posterior support cannot escape the initial prior support $\mathcal{S}_{\text{prior}}$. We treat this as a *didactic baseline*, not an inherent limitation of PF. To mitigate S-PSI *when it arises*, we propose a diffusion-enhanced correction module (Fig. 1) that **(i)** injects a small fraction of exploratory particles, **(ii)** applies covariance-scaled stochastic diffusion, and **(iii)** validates proposals via Metropolis–Hastings (MH), thereby enabling *minimal-bias*, data-triggered support expansion.

**Adaptive Diffusion via Exploratory Particles:** At each time step, a subset of particles is designated as *exploratory particles*, which introduce a uniform diffusion process into the framework. These particles are sampled from an adaptively extended bounding region $\mathcal{B}_k$, dynamically adjusted according to the current particle distribution to cover regions beyond the initial prior boundary $\mathcal{S}_{\text{prior}}$: $\mathcal{B}_k = [0, x_{\max} + \delta] \times [0, y_{\max} + \delta]$, where $x_{\max}$ and $y_{\max}$ denote the current maximum particle positions along each spatial dimension and $\delta$ is an adaptively determined margin parameter. Exploratory particles are then uniformly sampled from this expanded bounding region: $\Theta_k^{(j)} \sim \mathcal{U}(\mathcal{B}_k), j \in \mathcal{E}$, where $\mathcal{E}$ represents the indices of exploratory particles. The exploratory particles are initialised with small weights: $w_k^{(j)} = \frac{\epsilon}{|\mathcal{E}|}, \epsilon \ll 1$. This mechanism enables the bootstrap filter to sample states outside the original support range $\mathcal{S}_{\text{prior}}$, thereby increasing the likelihood of reaching states $\Theta^* \notin \mathcal{S}_{\text{prior}}$.

**Entropy-Driven Diffusion Regularisation:** To ensure that the exploratory diffusion does not collapse prematurely, an entropy regularisation term is added during the weight update step. This regularisation diffuses the weights across all particles, encouraging exploration of low-probability regions: $w_k^{(i)} \leftarrow w_k^{(i)} + \beta \cdot H(w_k)$, where $H(w_k)$ is the entropy of the weight distribution, defined as: $H(w_k) = -\sum_{i=1}^{N} w_k^{(i)} \log(w_k^{(i)} + \epsilon)$. The regularisation parameter $\beta$ is adaptively chosen based on the discrepancy between the current entropy and a predefined target entropy $H_{\text{target}}$: $\beta = \max\left(\beta_{\min}, \min\left(\beta_{\max}, \frac{H_{\text{target}} - H(w_k)}{H_{\text{target}}}\right)\right)$, where $\beta_{\min}$ and $\beta_{\max}$ represent the predefined minimum and maximum regularisation strengths, respectively. By penalising weight distributions that become overly concentrated, this adaptive entropy-based mechanism promotes balanced diffusion across the state space. The diffusion of weights helps exploratory particles retain influence, thus effectively encouraging the discovery and sustained exploration of regions beyond $\mathcal{S}_{\text{prior}}$.

**Kernel-Induced Stochastic Diffusion:** To further expand the particle support range dynamically, we introduce a stochastic diffusion mechanism based on kernel perturbations. Each particle $\Theta_k^{(i)}$ is perturbed by a Gaussian kernel that models diffusion within the local neighbourhood: $\Delta\Theta_k^{(i)} \sim h_{\text{opt}} \cdot \mathcal{L} \cdot \mathcal{N}(0, I)$, where $h_{\text{opt}} = A \cdot N^{-\frac{1}{n+4}}$ is the optimal kernel bandwidth dynamically adjusted to balance exploration and precision $\mathcal{L}$ is the lower triangular matrix obtained from the Cholesky decomposition of the covariance matrix $\Sigma$, ensuring diffusion adapts to the local particle distribution. The covariance matrix $\Sigma$ is computed dynamically: $\Sigma = \sum_{i=1}^{N} w_k^{(i)} (\Theta_k^{(i)} - \mu)(\Theta_k^{(i)} - \mu)^T + \lambda I$, where $\mu = \sum_{i=1}^{N} w_k^{(i)} \Theta_k^{(i)}$ is the weighted mean, and $\lambda > 0$ ensures positive definiteness of $\Sigma$.

This perturbation mechanism expands the effective support range by introducing stochastic diffusion, allowing particles to explore new regions iteratively: $\Theta_k^{(i)} \leftarrow \Theta_k^{(i)} + \Delta\Theta_k^{(i)}$.

**Diffusion-Driven Validation via MCMC:** To ensure consistency with the target posterior distribution, a Metropolis-Hastings acceptance criterion Hastings (1970) validates the diffused particles. For each perturbed particle $\Theta_k^{(i)}$, the acceptance probability is: $\alpha_i = \frac{w_{\text{new}}^{(i)}}{w_{\text{old}}^{(i)}} \cdot \exp\left(-\frac{1}{2}\Delta\Theta_k^{(i)^T} \Sigma^{-1} \Delta\Theta_k^{(i)}\right)$. A uniformly sampled random variable $u_i \sim \mathcal{U}(0, 1)$ determines whether the particle is accepted: $\Theta_k^{(i)} = \Theta_k^{(i)} - \Delta\Theta_k^{(i)} \cdot \mathbb{I}(\alpha_i < u_i)$, where $\mathbb{I}(\alpha_i < u_i)$ is the indicator function, which equals 1 when $\alpha_i < u_i$ is true, and 0 otherwise. This step ensures that the diffusion-driven expansion aligns with the posterior distribution, preserving the accuracy of the particle filter.

**Diffusion-Enhanced Particle Filtering:** By integrating exploratory particles, entropy-driven diffusion regularisation, and kernel-induced stochastic perturbations, the proposed framework creates a dynamic diffusion process that iteratively expands the effective support range. The recursive relationship for the support range becomes: $\mathcal{S}_{k+1} = (\mathcal{S}_k \cup \mathcal{B}) \oplus h_{\text{opt}}$, where $\oplus h_{\text{opt}}$ represents kernel-induced

stochastic diffusion. This diffusion framework effectively mitigates the posterior support invariance by continuously extending the particle filter's exploration capability, enabling robust state estimation for target states $\Theta^* \notin \mathcal{S}_{\text{prior}}$. The detailed theoretical analysis and justification of the effectiveness of our proposed method are provided in Appendix §D.

## 5 EXPERIMENT

To evaluate the ability of our method to dynamically recover from severe prior misalignment, we conduct experiments using the ISLC environments (ISLCenv), a simulation suite designed for emergency gas leak localization under varying levels of initial policy error. As detailed in §G, ISLCenv models a multi-source Gaussian plume and simulates noisy sensor observations without explicit reward signals. This setup allows us to rigorously assess the capacity of DEPF and competing baselines to overcome posterior support limitations and adaptively **infer the full 7-D parameter vector $\Theta$** in real time under realistic operational constraints, **rather than only the source coordinates**.

### 5.1 EVALUATION METRICS AND BASELINE ALGORITHMS

To evaluate our proposed approach and compare it against baseline algorithms, we use four distinct metrics: *Operational Completion Efficacy (OCE)*, which measures how frequently emergency response missions meet their goals, with higher scores indicating better deployment effectiveness; *Average Deployment Efficiency (ADE)*, representing the average distance traveled by response units, where shorter distances imply more efficient routing; *Response Execution Velocity (REV)*, quantifying the time duration from deployment to task completion, with faster times signifying more efficient operations; and *Localization Precision Score (LPS)*, assessing the accuracy of source localization by computing the average discrepancy between estimated and actual source locations, with smaller values denoting higher accuracy. Our proposed method is evaluated alongside various baseline algorithms, grouped according to their methodological foundations. The **first group** merges reinforcement learning with Bayesian inference and includes a single representative, *AGDC* Shi et al. (2024), which leverages the particle-filter RL posterior and uses intrinsic rewards derived from belief updates to guide exploration. Additionally, we include two other reinforcement learning approaches, *PC-DQN* Zhao et al. (2022) and *GMM-PFRL* Park et al. (2022b), which independently leverage particle filtering parameters as states for RL training. The **second group** integrates planning and Bayesian inference approaches, represented by *Infotaxis* Vergassola et al. (2007), *Entrotaxis* Hutchinson et al. (2018b), and *DCEE* Chen et al. (2021). Finally, we include a **third group** of classical SMC perturbation baselines that are not subject to S-PSI constraints: *PF+Jittering* Liu & Chen (1998); Doucet et al. (2000), *PF+Roughening* Gordon et al. (1993b; 1995), and *PF+Rejuvenation* Hastings (1970); Doucet et al. (2000). Implementation details and hyperparameter grids are given in §I and J.

### 5.2 SCENARIO PARAMETERIZATION AND EVALUATION

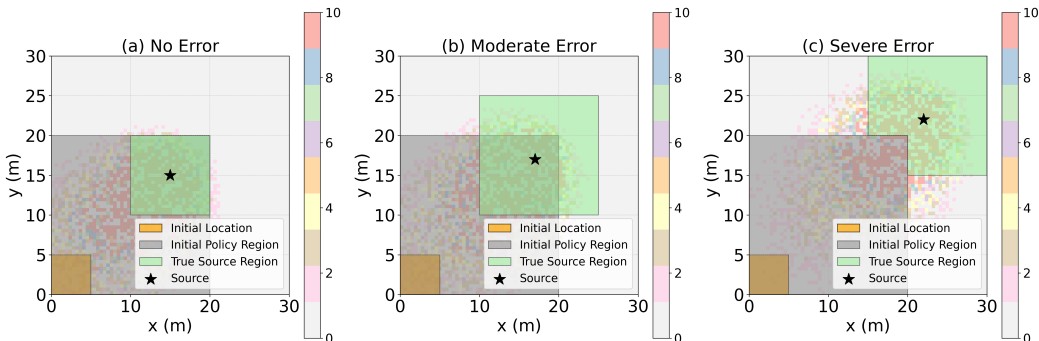

Figure 2: Experimental Scenarios for Policy Errors in Emergency Response.

We evaluate our proposed approach under three distinct scenarios representing different levels of initial policy error made by emergency response planners. In all scenarios, the agents operate in a

Table 1: Performance Comparison under Different Initial Policy Errors and Environment Scales

| Metric | Method | Small-scale Environment (1:30) | | | Large-scale Environment (1:300) | | |
|---|---|---|---|---|---|---|---|
| | | No Error | Moderate Error | Severe Error | No Error | Moderate Error | Severe Error |
| OCE↑ | DEPF (ours) | **0.90±0.03** | **0.90±0.03** | **0.89±0.03** | **0.90±0.03** | **0.90±0.03** | **0.88±0.03** |
| | AGDC | 0.90±0.03 | 0.45±0.04 | $< 0.05$ | 0.87±0.04 | 0.42±0.05 | $< 0.05$ |
| | PC-DQN | 0.80±0.04 | 0.40±0.04 | $< 0.05$ | 0.79±0.04 | 0.38±0.04 | $< 0.05$ |
| | GMM-PFRL | 0.80±0.04 | 0.40±0.04 | $< 0.05$ | 0.77±0.04 | 0.35±0.05 | $< 0.05$ |
| | Infotaxis | 0.85±0.04 | 0.25±0.02 | $< 0.05$ | $< 0.05$ | $< 0.05$ | $< 0.05$ |
| | Entrotaxis | 0.26±0.02 | 0.13±0.02 | $< 0.05$ | $< 0.05$ | $< 0.05$ | $< 0.05$ |
| | DCEE | 0.62±0.03 | 0.21±0.03 | $< 0.05$ | $< 0.05$ | $< 0.05$ | $< 0.05$ |
| | PF+Jittering | 0.88±0.03 | 0.40±0.04 | 0.06±0.02 | 0.85±0.04 | 0.36±0.05 | $< 0.05$ |
| | PF+Roughening | 0.89±0.03 | 0.48±0.04 | 0.10±0.03 | 0.86±0.04 | 0.44±0.05 | 0.08±0.03 |
| | PF+Rejuvenation | 0.89±0.03 | 0.52±0.04 | 0.16±0.03 | 0.86±0.04 | 0.48±0.05 | 0.12±0.03 |
| ADE↓ | DEPF (ours) | **19±0.8** | **22±1.2** | **27±1.7** | **167±15** | **200±10** | **255±15** |
| | AGDC | 18±0.9 | 59±11 | timeout | 168±15 | 235±20 | timeout |
| | PC-DQN | 20±1.0 | 60±11 | timeout | 193±16 | 246±28 | timeout |
| | GMM-PFRL | 19±0.9 | 60±20 | timeout | 200±16 | 250±36 | timeout |
| | Infotaxis | 40±2.0 | 70±30 | timeout | timeout | timeout | timeout |
| | Entrotaxis | 50±2.5 | 75±25 | timeout | timeout | timeout | timeout |
| | DCEE | 45±2.2 | 55±3.5 | timeout | timeout | timeout | timeout |
| | PF+Jittering | 22±1.0 | 65±8 | timeout | 180±15 | 250±22 | timeout |
| | PF+Roughening | 21±0.9 | 55±6 | timeout | 175±15 | 235±20 | timeout |
| | PF+Rejuvenation | 21±0.9 | 50±5 | 90±12 | 170±14 | 225±18 | 285±35 |
| REV↓ | DEPF (ours) | **0.10±0.05** | **0.10±0.05** | **0.10±0.05** | **0.10±0.05** | **0.10±0.05** | **0.10±0.05** |
| | AGDC | 0.10±0.05 | 0.10±0.05 | 0.10±0.05 | 0.12±0.05 | 0.40±0.15 | 1.50±0.30 |
| | PC-DQN | 0.12±0.05 | 0.12±0.05 | 0.12±0.05 | 0.15±0.07 | 0.45±0.17 | 1.50±0.30 |
| | GMM-PFRL | 0.11±0.04 | 0.11±0.04 | 0.11±0.04 | 0.13±0.06 | 0.42±0.16 | 1.50±0.30 |
| | Infotaxis | 1.30±0.06 | 1.30±0.06 | 1.30±0.06 | 1.80±0.08 | 3.00±0.15 | 4.00±0.20 |
| | Entrotaxis | 1.30±0.05 | 1.30±0.05 | 1.30±0.05 | 1.60±0.08 | 2.80±0.14 | 4.50±0.22 |
| | DCEE | 1.30±0.05 | 1.30±0.05 | 1.30±0.05 | 1.50±0.07 | 2.50±0.12 | 4.20±0.21 |
| | PF+Jittering | 0.11±0.05 | 0.30±0.12 | 1.80±0.35 | 0.12±0.05 | 0.90±0.25 | 2.80±0.45 |
| | PF+Roughening | 0.11±0.05 | 0.22±0.10 | 1.60±0.30 | 0.12±0.05 | 0.70±0.20 | 2.40±0.40 |
| | PF+Rejuvenation | 0.11±0.05 | 0.20±0.08 | 1.20±0.25 | 0.12±0.05 | 0.60±0.18 | 2.00±0.35 |
| LPS↓ | DEPF (ours) | **0.20±0.01** | **0.20±0.01** | **0.20±0.02** | **0.20±0.01** | **0.20±0.01** | **0.20±0.01** |
| | AGDC | 0.20±0.01 | 2.60±0.03 | 12.50±0.15 | 0.20±0.01 | 2.60±0.03 | 12.60±0.15 |
| | PC-DQN | 0.23±0.02 | 2.64±0.04 | 12.60±0.18 | 0.23±0.02 | 2.65±0.04 | 12.70±0.19 |
| | GMM-PFRL | 0.25±0.01 | 2.65±0.03 | 12.55±0.16 | 0.25±0.01 | 2.68±0.03 | 12.57±0.16 |
| | Infotaxis | 0.60±0.03 | 3.30±0.03 | 12.50±0.20 | 0.62±0.03 | 3.32±0.03 | 12.70±0.20 |
| | Entrotaxis | 0.70±0.03 | 3.40±0.03 | 13.00±0.23 | 0.72±0.03 | 3.42±0.03 | 13.30±0.23 |
| | DCEE | 0.65±0.03 | 3.55±0.04 | 12.80±0.22 | 0.68±0.03 | 3.58±0.05 | 13.10±0.23 |
| | PF+Jittering | 0.26±0.02 | 3.20±0.12 | 11.8±0.4 | 0.28±0.02 | 3.40±0.15 | 12.0±0.5 |
| | PF+Roughening | 0.24±0.02 | 2.90±0.10 | 10.5±0.4 | 0.26±0.02 | 3.10±0.12 | 11.0±0.5 |
| | PF+Rejuvenation | 0.23±0.02 | 2.70±0.09 | 9.0±0.3 | 0.24±0.02 | 2.90±0.10 | 10.0±0.4 |

$30 \times 30$ spatial domain, and the gas source location, wind speed, and wind direction are sampled according to the distributions specified in Table 7 of Appendix §G. Each training and testing instance uses entirely different plume parameters, with 1000 training and 500 testing instances. Agents start uniformly distributed within the initial subregion $(0, 5) \times (0, 5)$ and move with unit-length steps.

The three initial estimation error scenarios in Figure 2 represent increasing levels of prior misalignment between the initial belief and the true disaster source. **(1) No Error (Ideal)** corresponds to an ideal case where the initial particle distribution fully covers the true source region, providing a best-case baseline without decision uncertainty. **(2) Moderate Error (Partial Misalignment)** models a realistic situation where the initial assumed region partially overlaps with the true source area, testing whether each method can adapt to moderate initial mistakes. **(3) Severe Error (Complete Misalignment, PSI)** is the most challenging case, where the initial prior support is entirely disjoint from the true source location, creating a strict zero-prior barrier. This PSI scenario explicitly probes whether an algorithm can expand its posterior support and recover from severely misaligned initial state estimates, a setting in which standard bootstrap particle filters typically fail. The exact spatial layouts and parameter ranges for these three scenarios are provided in the appendix F.

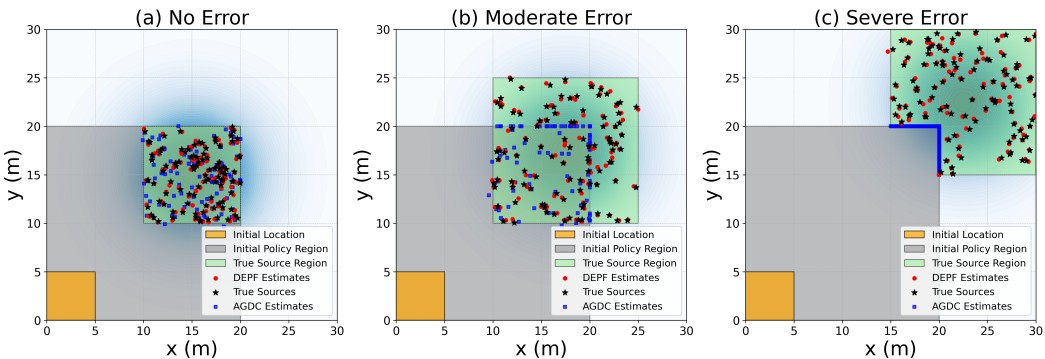

Figure 3: Visualization of Policy Error Scenarios in Emergency Response.

## 5.3 Performance Analysis Across Error Conditions and Environment Scales

We conduct a detailed evaluation of DEPF and nine baseline algorithms under varying levels of initial policy error—*No Error*, *Moderate Error*, and *Severe Error*—each tested in both *small-scale* (agent-domain ratio 1:30) and *large-scale* (1:300) environments. Performance is assessed using four metrics: OCE, ADE, LPS, and REV. Quantitative results are presented in Table 1, with spatial illustrations provided in Figure 3, which visualizes the posterior estimates from 100 test trials of DEPF and AGDC under each policy error condition.

**No Error (Ideal).** When the initial prior support fully covers the true source region, all methods perform well. DEPF attains **OCE = 0.90**, **LPS = 0.20**, **ADE = 19** (small-scale) and **ADE = 167** (large-scale), with **REV = 0.10** in both scales. AGDC closely matches DEPF on all metrics (**OCE = 0.90/0.87, LPS = 0.20, ADE = 18/168**). RL baselines PC-DQN and GMM-PFRL also converge (OCE $\approx 0.80$), while information-theoretic planners show varying success (e.g., Infotaxis achieves OCE $0.85$ in small-scale but degrades in large-scale). The three classical SMC perturbation baselines (PF+Jittering/Roughening/Rejuvenation) behave similarly or slightly worse than DEPF due to always-on diffusion incurring mild over-exploration cost in ideal conditions. **Moderate Error (Partial misalignment).** With partial overlap between prior support and ground truth, DEPF maintains high performance: **OCE = 0.90**, **LPS = 0.20**, **ADE = 22** (small) and **ADE = 200**, **REV = 0.10** (large). AGDC degrades substantially (**OCE = 0.45/0.42, LPS = 2.60, ADE = 59/235**) with increased timeouts at larger scale. PC-DQN and GMM-PFRL show similar or worse drops (OCE $\approx 0.40$). Among planners, Infotaxis/Entrotaxis/DCEE underperform markedly (see Table). Notably, the classical PF perturbation baselines recover *part* of the gap relative to RL/planning baselines: PF+Rejuvenation > PF+Roughening > PF+Jittering in OCE/LPS, consistent with resample–move providing the strongest rejuvenation. However, all three remain *well below* DEPF in both accuracy and efficiency, especially in the large-scale setting where path lengths and timeouts increase. **Severe Error (Complete misalignment; S-PSI test).** This scenario creates a strict zero-prior barrier: the initial prior support is disjoint from the true source region. DEPF remains stable with **OCE = 0.89** (small) and **0.88** (large), **LPS = 0.20**, and **ADE = 27/255**—all below the 100/300 step timeout thresholds and with **REV = 0.10**. In contrast, *bootstrap PF under the S-PSI baseline* (zero transition, no rejuvenation) cannot escape the prior support; accordingly, AGDC, PC-DQN, GMM-PFRL, and information-theoretic planners collapse (**OCE** $< 0.05$, **LPS** $> 12.5$; frequent timeouts). The classical SMC perturbation baselines alleviate this limitation to a degree: PF+Rejuvenation achieves the highest non-zero success among the three (occasionally localizing in the small-scale severe case), followed by PF+Roughening and then PF+Jittering; nevertheless, their success rates remain low and many runs still timeout—especially in the large-scale severe case—highlighting the cost of always-on diffusion and the absence of belief-triggered control. **Scaling with domain size.** Across error conditions, most baselines suffer pronounced degradation when moving from 1:30 to 1:300: OCE drops, ADE/REV increase, and timeouts spike, reflecting insufficient long-range exploration when prior guidance is misleading. DEPF sustains **OCE = 0.88–0.90** with **LPS = 0.20** and **REV = 0.10** across scales, indicating robust scalability. Its advantage is clearest under Moderate/Severe misalignment, where belief-triggered support expansion and covariance-aligned diffusion provide minimal-bias exploration precisely when needed, in contrast to always-on perturbations. **Takeaways.** (i) In ideal settings, DEPF is competitive with the best baselines. (ii) Under partial misalignment,

DEPF preserves accuracy/efficiency while RL/planning baselines and classical PF perturbations degrade, with PF+Rejuvenation the strongest among the latter yet still below DEPF. (iii) Under complete misalignment (S-PSI test), DEPF consistently sustains high OCE and low LPS at both scales; all other methods either fail or achieve only occasional recovery (PF+Rejuvenation) with substantial timeouts. These results substantiate DEPF as a principled, data-triggered fallback when early-stage priors are unreliable.

## 5.4 ABLATION AND SENSITIVITY STUDIES

We perform a series of ablations and sensitivity studies to understand how different components and hyperparameters of DEPF affect performance under the *Severe Error* setting, where the true source lies entirely outside the initial prior support.

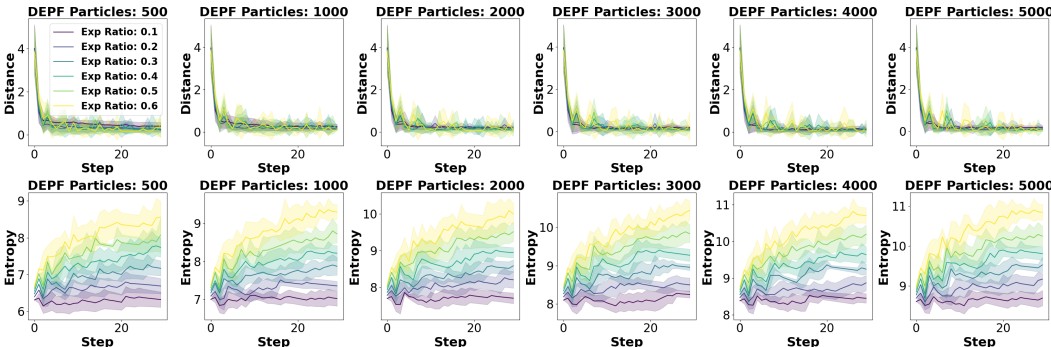

Figure 4: Impact of the number of particles and $\delta$ on DEPF performance under *Severe Error*.

**Sensitivity to particle number and margin parameter $\delta$.** We first analyze how DEPF's performance is affected by the number of particles and the support margin ratio $\delta$, which determines the proportion by which the posterior support region is expanded beyond the initial prior boundary at each step. As shown in Figure 6, increasing $\delta$ generally allows the model to converge more quickly—reflected in lower belief-to-goal distances—because particles are better able to explore outside the constrained prior region. However, this broader support also leads to higher posterior entropy, as particles become more dispersed and the belief distribution less concentrated. This trade-off is particularly pronounced at low particle counts (e.g., $N = 500$–$2000$), where large $\delta$ causes belief imprecision. When using larger particle sets ($N \geq 3000$), DEPF maintains both fast convergence and acceptable entropy levels even at higher $\delta$, indicating that sufficient particle resolution can stabilize exploration-induced uncertainty. Overall, a moderate support ratio ($\delta \in [0.2, 0.4]$) paired with adequate particle numbers achieves a good balance between exploration and localization precision. These results reinforce DEPF's ability to recover from severely misaligned initial state estimates by dynamically expanding support and refining the belief.

Table 2: Component-wise ablation of DEPF under *No Error* and *Severe Error*.

| Idx | Components | | | No Error | | | | Severe Error | | | |
|---|---|---|---|---|---|---|---|---|---|---|---|
| | Entropy | Stoch. Diffusion | MH Valid. | OCE | ADE | REV | LPS | OCE | ADE | REV | LPS |
| 1 | ✓ | ✓ | ✓ | 0.90±0.03 | 19±0.8 | 0.10±0.05 | 0.20±0.01 | 0.88±0.03 | 27±1.7 | 0.10±0.05 | 0.20±0.01 |
| 2 | × | ✓ | ✓ | 0.90±0.03 | 19±0.8 | 0.10±0.05 | 0.20±0.01 | 0.62±0.17 | 34±2.4 | 0.10±0.05 | 3.00±0.45 |
| 3 | ✓ | × | ✓ | 0.90±0.03 | 19±0.8 | 0.10±0.05 | 0.20±0.01 | 0.00 | timeout | 0.10±0.05 | 12.50±0.15 |
| 4 | ✓ | ✓ | × | not work | | | | not work | | | |

**Component-wise ablation.** We next disentangle the contribution of each DEPF component. The component-wise ablation in Table 2 shows that all three mechanisms—entropy-based weight regularization, covariance-scaled stochastic diffusion, and MH-based acceptance—are jointly necessary for robust recovery. Removing stochastic diffusion collapses performance under *Severe Error*, yielding timeouts and large residual errors despite correct behavior in the *No Error* setting. Disabling entropy regularization substantially lowers OCE and increases LPS (e.g., $0.62$ OCE and $\sim 3.0$ LPS), indicating premature weight concentration and insufficient posterior spread. Absent MH validation, the algorithm becomes ill-posed (`not work`), confirming the role of acceptance control in maintaining

Bayesian consistency. These outcomes highlight that DEPF's gains do not stem from ad-hoc noise, but from a calibrated, evidence-triggered expansion pipeline.

**Hyperparameter sensitivity: $\lambda$ and $A$.** We then study the kernel stability parameter $\lambda$, which controls the diagonal regularization added to the empirical covariance, and the KDE bandwidth constant $A$. The two tables below summarize the effect of these hyperparameters.

Table 3: Sensitivity to $\lambda$.

| $\lambda$ | OCE↑ | ADE↓ | REV↓ | LPS↓ |
|---|---|---|---|---|
| $1 \times 10^{-1}$ | 0.87±0.04 | 22±1.5 | 0.25±0.04 | 0.10±0.05 |
| $1 \times 10^{-2}$ | 0.90±0.03 | 19±0.8 | 0.22±0.04 | 0.10±0.05 |
| $1 \times 10^{-3}$ | 0.91±0.05 | 17±1.2 | 0.23±0.04 | 0.10±0.05 |
| $1 \times 10^{-4}$ | 0.89±0.08 | 18±2.1 | 0.25±0.04 | 0.10±0.05 |
| $1 \times 10^{-5}$ | 0.85±0.12 | 25±3.5 | 0.26±0.04 | 0.10±0.05 |

Table 4: Sensitivity to KDE $A$.

| $A$ | Belief Dist.↓ | Entropy↓ | LPS↓ | Steps↓ |
|---|---|---|---|---|
| 0.10 | 2.10 | 0.37 | 0.23 | 25 |
| 0.30 | 1.00 | 0.26 | 0.21 | 23 |
| 0.50 | **0.22** | **0.17** | **0.20** | **18** |
| 1.00 | 0.35 | 0.22 | 0.22 | 20 |
| 2.00 | 1.10 | 0.34 | 0.26 | 27 |

Mid-range values of $\lambda$ ($10^{-3}$–$10^{-2}$) strike the best balance between numerical stability and diffusion accuracy, yielding the lowest ADE and competitive OCE. Too large a $\lambda$ overdamps useful moves and slows convergence; too small a $\lambda$ induces numerical instability and degraded metrics. For the bandwidth constant $A$, an intermediate value $A \approx 0.5$ minimizes belief-to-goal distance, posterior entropy, and convergence steps, whereas either under- or over-diffusion (very small or very large $A$) harms recovery. This confirms the need for bandwidth matching to the evolving posterior covariance, as encoded by DEPF's covariance-scaled perturbations.

**Hyperparameter sensitivity: $\beta$ and exploratory ratio.** Finally, we study the entropy regularization strength $\beta$ and the ratio of exploratory particles per step, summarized in Table 5 and Table 6.

Table 5: Sensitivity to $\beta$.

| $\beta$ | OCE↑ | ADE↓ | REV↓ | LPS↓ |
|---|---|---|---|---|
| 0.1 | $0.88 \pm 0.04$ | $22 \pm 1.2$ | $0.10 \pm 0.05$ | $0.25 \pm 0.03$ |
| 0.2 | $0.89 \pm 0.03$ | $20 \pm 1.0$ | $0.10 \pm 0.05$ | $0.22 \pm 0.02$ |
| 0.3 | $0.90 \pm 0.03$ | $19 \pm 0.8$ | $0.10 \pm 0.05$ | $0.20 \pm 0.01$ |
| 0.4 | $0.91 \pm 0.02$ | $18 \pm 0.7$ | $0.10 \pm 0.05$ | $0.19 \pm 0.01$ |
| 0.5 | $0.91 \pm 0.02$ | $18 \pm 0.6$ | $0.10 \pm 0.05$ | $0.20 \pm 0.01$ |

Table 6: Sensitivity to the ratio of exploratory.

| Ratio (%) | Belief Dist.↓ | Entropy↓ | LPS↓ | Steps↓ |
|---|---|---|---|---|
| 1% | 1.60 | 0.29 | 0.22 | 26 |
| 2% | 0.65 | 0.21 | 0.20 | 20 |
| 5% | **0.22** | **0.17** | **0.20** | **18** |
| 10% | 0.30 | 0.19 | 0.21 | 19 |
| 20% | 0.75 | 0.25 | 0.22 | 22 |

As $\beta$ increases from 0.1 to 0.4, the task OCE steadily improves from 0.88±0.04 to 0.91±0.02, while the path cost (ADE) decreases from 22±1.2 to 18±0.7 and the localization error (LPS) drops from 0.25±0.03 to 0.19±0.01. Execution latency (REV) remains around 0.10±0.05, indicating that entropy regularization primarily enhances convergence efficiency and localization accuracy without introducing additional delays. When $\beta$ is further increased to 0.5, all metrics remain essentially unchanged, suggesting that performance enters a plateau and becomes robust within the range $\beta \in [0.3, 0.5]$. For the exploratory-particle ratio, there is a clear interior optimum around 5% (Table 6). Too few exploratory particles slow support expansion and delay recovery; too many inject excess randomness, raising entropy and slightly reducing stability. Together with the diffusion-related ablations, these results yield a practical recipe: pair a moderate exploratory ratio (about 5%) with a mid-range bandwidth ($A \approx 0.5$), moderate smoothing ($\beta \approx 0.3$–0.4), and a stability parameter $\lambda$ in $[10^{-3}, 10^{-2}]$ to reliably break S-PSI while maintaining efficient convergence.

## 6 CONCLUSION

In this work, we identified and formalized S-PSI, a baseline-specific limitation of bootstrap particle filters that arises under zero transition and no rejuvenation. To address this issue, we proposed the DEPF framework, which dynamically expands posterior support through exploratory particle injection, entropy-driven diffusion, and kernel-based perturbations validated by Metropolis–Hastings. Experiments on hazardous-gas localization tasks demonstrated that DEPF consistently corrects severe prior misalignment, substantially outperforming both classical perturbation strategies and RL-based baselines. By resolving bootstrap-specific lock-in while maintaining Bayesian rigor, DEPF offers a robust and practical solution for decision-support systems in emergency management.

These statements **are not** included in the main body of the paper; according to the **ICLR 2026 Author Guide**, it is recommended to provide them under the **Reproducibility Statement** and **Ethics Statement**.

===== ===== ===== ===== **Reproducibility Statement -Begin** ===== ===== ===== =====

These statements **are not** included in the main body of the paper; according to the **ICLR 2026 Author Guide**, it is recommended to provide them under the Reproducibility Statement.

## Reproducibility Statement

We have taken multiple steps to ensure the reproducibility of our results:

**Detailed Methodology** – Section 3–4 provide a complete mathematical formulation of the problem, including the particle filtering framework, the definition of Stationarity-Induced Posterior Support Invariance (S-PSI), and the components of the proposed Diffusion-Enhanced Particle Filtering (DEPF) algorithm.

**Experimental Setup** – Section 5 and Appendix E describe the ISLC simulation environments, plume dispersion models, and parameter distributions used to generate training and evaluation data. Hyperparameter grids and implementation details of baseline methods are provided in Appendix G–H.

**Evaluation Metrics** – Appendix F provides precise definitions of all evaluation metrics (OCE, ADE, REV, LPS), ensuring clarity in performance comparisons.

**Ablation and Sensitivity Analyses** – Appendix K presents systematic ablation studies and hyper-parameter sensitivity analyses (e.g., entropy regularization strength, kernel stability $\lambda$, exploratory ratio), which demonstrate the robustness of our findings across settings.

**Hardware Transparency** – Appendix L reports the hardware configuration (Linux server, AMD EPYC CPUs, NVIDIA A6000 GPUs, and memory usage), enabling replication of the computational environment.

===== ===== ===== ===== **Reproducibility Statement -End** ===== ===== ===== =====

===== ===== ===== ===== **Ethics Statement -Begin** ===== ===== ===== =====

These statements **are not** included in the main body of the paper; according to the **ICLR 2026 Author Guide**, it is recommended to provide them under the Ethics Statement.

## Ethics Statement

This work addresses decision-making under uncertainty in high-stakes emergency response scenarios, such as hazardous gas leak localization. The proposed methodology is designed to improve robustness and adaptability of inference under erroneous initial assumptions. Our experiments are fully simulation-based and do not involve human participants, personal data, or sensitive information. As such, this research does not raise direct concerns regarding privacy, safety, or fairness in data handling. Nevertheless, we acknowledge that deployment of autonomous decision-support systems in real-world emergency management contexts must be carefully assessed for reliability, safety, and ethical use, particularly when human lives and societal resources are at stake. Our contributions should be viewed as methodological advances that require further validation and regulatory oversight before operational use.

===== ===== ===== ===== **Ethics Statement -End** ===== ===== ===== =====

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

APPENDIX

# A   **THE USE OF LARGE LANGUAGE MODELS**

We employed large language models (LLMs) solely to polish the writing of this paper, such as improving grammar, clarity, and readability. The models were not used for generating original ideas, experiments, analyses, or results. All scientific contributions, methods, and conclusions presented in the paper are entirely the work of the authors.

# B   PARTICLE FILTERING: RECURSIVE APPROXIMATION OF POSTERIOR DISTRIBUTIONS

The particle filtering process recursively approximates the posterior distribution $P(\Theta_k \mid z_{1:k})$ through four main steps: sampling, weight update, normalization, and resampling. These steps iteratively adapt particle sets to new observations, enabling sequential Bayesian inference.

**Step 1. Sampling (Prediction):** Gordon et al. (1993b) Particles are sampled from an importance distribution $q(\Theta_k \mid \Theta_{k-1}, z_k)$, typically approximated by the state transition model $P(\Theta_k \mid \Theta_{k-1})$. The propagation of particles is expressed as:

$$\Theta_k^{(i)} \sim P(\Theta_k \mid \Theta_{k-1}^{(i)}),$$

where $\Theta_k^{(i)}$ denotes the $i$-th particle at time $k$.

**Step 2. Weight Update:** Doucet et al. (2001a) Particle weights are updated to reflect their likelihood given the new observation. The unnormalized weights are calculated as:

$$\tilde{w}_k^{(i)} = w_{k-1}^{(i)} \cdot \frac{P(z_k \mid \Theta_k^{(i)})P(\Theta_k^{(i)} \mid \Theta_{k-1}^{(i)})}{q(\Theta_k^{(i)} \mid \Theta_{k-1}^{(i)}, z_k)},$$

where $P(z_k \mid \Theta_k^{(i)})$ is the observation likelihood, and $q(\Theta_k \mid \Theta_{k-1}, z_k)$ is the importance distribution used in sampling.

**Step 3. Normalization of Weights:** Liu & Chen (1998) Weights are normalized to maintain their probabilistic interpretation:

$$w_k^{(i)} = \frac{\tilde{w}_k^{(i)}}{\sum_{j=1}^{N} \tilde{w}_k^{(j)}}.$$

**Step 4. Resampling:** Doucet et al. (2000) To mitigate weight degeneracy (domination by a few particles), resampling occurs when the effective particle number $N_{\text{eff}}$ falls below a threshold:

$$N_{\text{eff}} = \frac{1}{\sum_{i=1}^{N} (w_k^{(i)})^2}.$$

Particles are then resampled based on current weights, and weights are reset uniformly:

$$w_k^{(i)} = \frac{1}{N}.$$

These recursive steps approximate the posterior $P(\Theta_k \mid z_{1:k})$. However, under the *stationary bootstrap baseline with zero transition and no rejuvenation*, particle trajectories remain confined to the initial prior support. We formalize this diagnostic effect as **Stationarity-Induced Posterior Support Invariance (S-PSI)**: the posterior cannot expand beyond the prior support in this baseline setting. Importantly, S-PSI is *not* an inherent property of particle filtering—classical perturbation strategies such as jittering, roughening, or resample–move can, in principle, relax this constraint, and are included as baselines in our experiments.

# C   STATIONARITY-INDUCED POSTERIOR SUPPORT INVARIANCE (S-PSI)

Let the prior $p_0(\Theta)$ have support $\mathcal{S}_0 := \text{supp}\, p_0(\Theta) = \{\theta \mid p_0(\theta) > 0\} \subset \mathbb{R}^7$. If particles are initialized exclusively within $\mathcal{S}_0$, then under a *stationary bootstrap baseline with zero transition and no rejuvenation*, the posterior distributions at all subsequent time steps will satisfy:

$$\text{supp}(p(\Theta \mid z_{1:k})) \subseteq \mathcal{S}_0, \quad \forall k.$$

We refer to this diagnostic baseline effect as **Stationarity-Induced Posterior Support Invariance (S-PSI)**. It indicates that:

> Even if observations strongly suggest a source location outside the prior support, the particle filter cannot estimate it—**not due to insufficient likelihood**, but because no particles exist in that region under the baseline assumptions.

Consequently, estimation fails if the true source $\Theta^*$ lies outside the initial prior support:

$$\Theta^* \notin \mathcal{S}_0 \quad \Rightarrow \quad p(\Theta^* \mid z_{1:k}) = 0, \quad \forall k.$$

**Proposition C.1** (S-PSI under the stationary bootstrap baseline)**.** *Given particles initialized exclusively within the prior support $\mathcal{S}_0$, the particle support range $\mathcal{S}_k$ at any time $k$ satisfies:*

$$\mathcal{S}_k \subseteq \mathcal{S}_0, \quad \forall k \geq 0,$$

*with the base case:*

$$\mathcal{S}_0 = \mathcal{S}_{prior}.$$

*Thus, under the baseline assumptions, the posterior support remains invariant over time:*

$$\mathcal{S}_k \subseteq \mathcal{S}_{prior}, \quad \forall k \geq 0.$$

*Proof.* By definition, the particle support $\mathcal{S}_k$ at step $k$ is:

$$\mathcal{S}_k = \bigcup_{i=1}^{N} \{\Theta_k \in \mathbb{R}^7 : P(\Theta_k \mid \Theta_{k-1}^{(i)}) > 0\}, \quad \Theta_{k-1}^{(i)} \in \mathcal{S}_{k-1}.$$

For the base case at $k = 0$, by construction:

$$\mathcal{S}_0 = \mathcal{S}_{\text{prior}} = \{\Theta_0 \in \mathbb{R}^7 : p_0(\Theta_0) > 0\}.$$

Assume inductively that $\mathcal{S}_{k-1} \subseteq \mathcal{S}_{\text{prior}}$. Particle propagation at time $k$ depends entirely on particles from step $k - 1$, and under the stationary baseline assumption (no transition noise, no rejuvenation), we have:

$$\mathcal{S}_k \subseteq \mathcal{S}_{k-1}.$$

Combining this with the inductive hypothesis:

$$\mathcal{S}_k \subseteq \mathcal{S}_{\text{prior}}, \quad \forall k \geq 0.$$

Thus, particles cannot move beyond the initial support boundary under this baseline, making the posterior support invariant over time. $\qquad\square$

**Remarks.** S-PSI is *not* an inherent property of particle filtering but a consequence of the stationary bootstrap baseline. Classical perturbation strategies such as jittering, roughening, or resample–move rejuvenation can, in principle, relax this invariance and are included as baselines in §5. Our proposed method DEPF builds upon this insight by introducing belief-triggered exploratory injection, covariance-scaled diffusion, and MH validation, enabling adaptive support expansion only when data contradict the current belief.

# D   THEORETICAL ANALYSIS AND JUSTIFICATION

## D.1   INTRODUCTION

In this appendix, we provide a theoretical analysis of the Diffusion-Enhanced Particle Filtering (DEPF) framework proposed in Section 4, focusing particularly on the rationality, correctness, and asymptotic unbiasedness of its mechanisms: exploratory particle injection, entropy-based weight smoothing, kernel-induced stochastic perturbation, and Metropolis–Hastings (MH) validation.

## D.2 EXPLORATORY PARTICLES AND MODIFIED PRIOR

By injecting exploratory particles uniformly drawn from an expanded region $B_k$ beyond the initial prior support $S_0$, DEPF implicitly modifies the initial prior distribution $p_0(\Theta)$ as:

$$p_{\text{new-prior}}(\Theta) = (1 - \epsilon)p_0(\Theta) + \epsilon U(B_k), \quad \epsilon \ll 1, \tag{1}$$

where $U(B_k)$ denotes the uniform distribution on $B_k$. This is equivalent to relaxing the diagnostic **Stationarity-Induced Posterior Support Invariance (S-PSI)** baseline, which arises under a stationary bootstrap PF with zero transition and no rejuvenation (§3.4). Since $\epsilon$ is small, this injection introduces minimal bias while enabling posterior support expansion outside $S_0$.

## D.3 ENTROPY-BASED WEIGHT SMOOTHING

To preserve particle diversity, DEPF smooths weights toward higher entropy. One implementation is:

$$w_k^{(i)} \leftarrow w_k^{(i)} + \beta H(w_k), \quad H(w_k) = -\sum_{i=1}^{N} w_k^{(i)} \log(w_k^{(i)} + \epsilon), \tag{2}$$

which can be viewed as Bayesian smoothing (Doucet et al., 2001b; Liu & Chen, 1998). In practice, we also adopt a standard *tempering* scheme

$$\tilde{w}_k^{(i)} \propto (w_k^{(i)})^{1/T_k}, \quad w_k^{(i)} = \tilde{w}_k^{(i)} / \sum_j \tilde{w}_k^{(j)},$$

with $T_k \geq 1$ adapted from entropy gaps, ensuring theoretical coherence with standard SMC while mitigating degeneracy.

## D.4 KERNEL-INDUCED STOCHASTIC DIFFUSION

Particles are further perturbed by a Gaussian kernel:

$$\Delta\Theta_k^{(i)} \sim h_{\text{opt}} L \mathcal{N}(0, I), \quad h_{\text{opt}} = A \cdot N^{-\frac{1}{n+4}}, \quad \Theta_k^{(i)} \leftarrow \Theta_k^{(i)} + \Delta\Theta_k^{(i)}, \tag{3}$$

where $LL^\top = \Sigma$ is the Cholesky factorization of the weighted covariance. This is equivalent to a kernel density approximation (Silverman, 2018), and ensures that as $N \to \infty$ the perturbed empirical measure converges to the true posterior.

## D.5 METROPOLIS–HASTINGS VALIDATION

To maintain Bayesian coherence, proposals are validated by a standard MH acceptance step:

$$\alpha_i = \min\left(1, \frac{\pi(\Theta_i') \, q(\Theta_i \mid \Theta_i')}{\pi(\Theta_i) \, q(\Theta_i' \mid \Theta_i)}\right), \tag{4}$$

with $\pi(\Theta) \propto p(z_k \mid \Theta) \, p(\Theta \mid z_{1:k-1})$. For symmetric Gaussian $q$, this reduces to

$$\alpha_i = \min\left(1, \frac{p(z_k \mid \Theta_i')}{p(z_k \mid \Theta_i)}\right). \tag{5}$$

This guarantees detailed balance and convergence to the true posterior distribution (Hastings, 1970).

## D.6 CONVERGENCE AND THEORETICAL GUARANTEES

Combining these mechanisms, DEPF ensures:

1. Minimal bias from exploratory injection, expanding support beyond $S_0$ only when warranted.

2. Stability of particle diversity due to entropy-based weight smoothing.

3. Asymptotic convergence via kernel-induced diffusion validated by MH acceptance.

Therefore, DEPF is asymptotically unbiased:

$$\lim_{N \to \infty} p_{\text{particle}}(\Theta \mid z_{1:k}) = p_{\text{true posterior}}(\Theta \mid z_{1:k}). \tag{6}$$

# E PROOF OF SUPPORT RANGE EXPANSION BEYOND THE PRIOR BOUNDARY

Under the stationary bootstrap baseline with zero transition and no rejuvenation, particle filters exhibit the diagnostic phenomenon we term **Stationarity-Induced Posterior Support Invariance (S-PSI)**: the posterior support remains confined to the prior support set $\mathcal{S}_{\text{prior}}$. This is not an inherent limitation of PF but a consequence of the baseline assumptions (§3.4). Classical remedies such as jittering, roughening, or resample–move can, in principle, expand support, but they do so in an always-on and often inefficient manner. Here, we formally prove that with DEPF's enhancements—exploratory injection, entropy-based regularisation, and kernel-induced stochastic diffusion validated by MH—the support set $\mathcal{S}_k$ can expand beyond $\mathcal{S}_{\text{prior}}$.

**Proposition E.1** (Expansion of Support Range Beyond S-PSI)**.** *With the proposed enhancements, the support range $\mathcal{S}_k$ satisfies the recursion*

$$\mathcal{S}_k^{new} = \mathcal{S}_k \cup \mathcal{B}, \qquad \mathcal{S}_{k+1} = \mathcal{S}_k^{new} \oplus h_{opt},$$

*where $\mathcal{B}$ is the extended bounding box sampled by exploratory particles and $\oplus h_{opt}$ denotes kernel-induced diffusion. Starting from*

$$\mathcal{S}_0 = \mathcal{S}_{prior},$$

*for any target state $\Theta^* \in \mathcal{B}$, there exists a finite step $k$ such that*

$$\Theta^* \in \mathcal{S}_k.$$

*Proof.* **Step 1: Base case.** At $k = 0$, $\mathcal{S}_0 = \mathcal{S}_{\text{prior}}$.

**Step 2: Exploratory injection.** At step $k > 0$, particles sampled from $\mathcal{U}(\mathcal{B})$ enlarge the predictive support:

$$\mathcal{S}_k^{\text{new}} = \mathcal{S}_k \cup \mathcal{B}.$$

**Step 3: Weight-based survival.** Exploratory particles obtain weights proportional to their likelihood:

$$w_k^{(j)} \propto p(z_k \mid \Theta_k^{(j)}).$$

Particles consistent with observations survive resampling, ensuring that regions supported by data persist.

**Step 4: Kernel-induced diffusion.** Surviving particles are further perturbed:

$$\Delta\Theta_k^{(i)} \sim h_{\text{opt}} L \mathcal{N}(0, I), \qquad \Theta_k^{(i)} \leftarrow \Theta_k^{(i)} + \Delta\Theta_k^{(i)},$$

which dilates the support: $\mathcal{S}_{k+1} = \mathcal{S}_k^{\text{new}} \oplus h_{\text{opt}}$.

**Step 5: Induction.** Iterating steps 2–4, if $\Theta^* \in \mathcal{B}$, then with positive probability it survives weighting and resampling, and by induction there exists finite $k$ such that $\Theta^* \in \mathcal{S}_k$. □

**Remarks.** This result shows that DEPF breaks the S-PSI constraint by combining: (i) exploratory injection to seed new regions, (ii) likelihood-driven survival so only data-supported regions expand, and (iii) kernel diffusion to propagate local coverage. Together with MH validation (§3.1), these steps ensure support expansion is both data-triggered and statistically coherent.

# F DETAILED SCENARIO SPECIFICATIONS

The three policy error scenarios are defined as follows: **(1) No Error (Ideal Scenario)** in Figure 2(a) represents an ideal policy scenario in which the initial government decision on the disaster source location is completely accurate. Specifically, the initial particle distribution (gray region: $(0, 20) \times (0, 20)$) accurately covers the true source region (green region: $(10, 20) \times (10, 20)$). This scenario serves as a baseline, assessing the best-case performance without initial decision uncertainty. **(2) Moderate Error (Partial Misalignment Scenario)** in Figure 2(b) simulates a realistic and moderate error scenario where the government's initial policy assumptions about the disaster location (gray region: $(0, 20) \times (0, 20)$) are partially misaligned, overlapping only partially with the true disaster source region (green region: $(10, 25) \times (10, 25)$). This setup tests the ability of our method and baseline algorithms to adaptively correct moderate inaccuracies in initial policy assumptions,

reflecting realistic emergency management conditions. **(3) Severe Error (Complete Misalignment, PSI Scenario)** in Figure 2(c) represents the most challenging and realistic scenario, termed the Posterior Support Invariance scenario, in which the initial policy completely excludes the true disaster location. Here, the initial policy region (gray region: $(0, 20) \times (0, 20)$) and the true disaster source regions (green regions: $(15, 30) \times (20, 30) \cup (20, 30) \times (15, 20)$) are entirely disjoint, creating a strict zero-prior barrier. This scenario explicitly tests each algorithm's robustness and ability to dynamically correct severe initial policy errors—situations where bootstrap particle filtering methods typically fail due to their inherent inability to extend beyond the initial support.

## G  ISLC ENVIRONMENTS

In this section, we describe the simulation environment adapted to a reinforcement learning and planning framework suitable for conducting research as outlined in Section §3 of the paper.

Table 7: Parameter Distributions for the Training Scenarios

| Source Parameter | Distribution |
|---|---|
| location of field source $x_s$ | Uniform $\mathcal{U}(5, 25)$ |
| location of field source $y_s$ | Uniform $\mathcal{U}(5, 25)$ |
| Release Strength $q_s$ | Uniform $\mathcal{U}(10, 3000)$ |
| Wind Speed $u_s$ | Uniform $\mathcal{U}(0, 6)$ |
| Decay parameter $\lambda$ | Uniform $\mathcal{U}(0, 8)$ |
| Diffusivity $d_s$ | Uniform $\mathcal{U}(1, 5)$ |
| Sensor Noise $\epsilon$ | Fixed at 0.5 |
| Environmental Noise $\sigma$ | Fixed at 0.4 |
| Effective Samples $N_{eff}$ | Fixed at 0.6 |

### G.1  ENVIRONMENT OVERVIEW

The environment simulates a pollutant dispersion scenario within a defined two-dimensional spatial boundary. An RL agent operates within this environment by sequentially selecting positions to gather sensor measurements. Importantly, the environment itself does not explicitly provide reward signals. Instead, the agent has access only to its positional coordinates and the concentration measurements obtained at those positions.

The spatial boundaries for the simulation environment are as follows:

- $x$-axis: from 0 to 25 meters.
- $y$-axis: from 0 to 25 meters.

### G.2  POLLUTANT DISPERSION MODEL

Pollutant concentrations are simulated using a Gaussian plume dispersion model in Figure 5, characterized by source emission parameters and environmental conditions:

$$C = \frac{q_s}{4\pi d_s \cdot \text{dist}} \exp\left(-\frac{(x - x_{\text{source}})u\cos\phi + (y - y_{\text{source}})u\sin\phi + \text{dist}}{2d_s}\right),$$

where $q_s$ is the pollutant release rate, $u$ is the wind speed, $\phi$ is the wind direction, $d_s$ represents dispersion coefficients, dist denotes the Euclidean distance between the agent's position and the pollutant source.

### G.3  AGENT OBSERVATIONS AND ACTIONS

At each time step, the RL agent observes:

- Its current two-dimensional coordinates $(x, y)$.

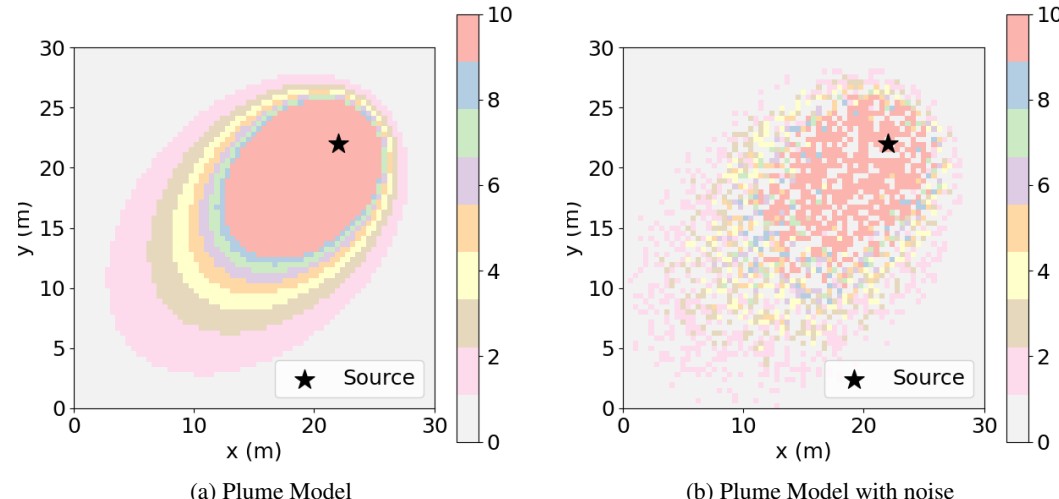

(a) Plume Model

(b) Plume Model with noise

Figure 5: Gaussian Plume Model Visualization.

- The pollutant concentration $z$ measured at that location.

Based on these observations, the agent selects its next position within the spatial boundary constraints. The agent's action space consists of discrete movements within predefined distances, typically in increments of 1 meter in either the $x$ or $y$ direction.

### G.4 NO EXPLICIT REWARD STRUCTURE

Crucially, the environment itself does not provide any explicit reward signals. The agent must indirectly **infer** useful positional information solely from the sequence of measured pollutant concentrations. Consequently, the agent's learning process depends entirely on interpreting these indirect observations, making it a suitable scenario to investigate methods dealing with sparse or implicit rewards, as discussed extensively in Section §3.

## H DETAILED DESCRIPTION OF EVALUATION METRICS

We provide detailed definitions and interpretations for each evaluation metric used in the experiments, accompanied by illustrative examples:

- **Operational Completion Efficacy (OCE)**: This metric quantifies the proportion of emergency response missions successfully completed. For instance, if an algorithm successfully completes 90 out of 100 missions, it achieves an OCE of 0.90, indicating high reliability and effectiveness in operational scenarios.

- **Average Deployment Efficiency (ADE)**: ADE measures the average total distance traveled by response units during their missions. For example, if algorithm A requires units to travel an average of 15 kilometers per mission, while algorithm B achieves the same objectives in only 10 kilometers, algorithm B demonstrates a better (lower) ADE, suggesting superior efficiency in routing and resource usage.

- **Response Execution Velocity (REV)**: REV assesses the total time from initial deployment to mission completion. As an example, consider two algorithms where one completes missions in an average of 20 minutes, while the other requires 30 minutes on average. The first algorithm, with its shorter completion times (higher REV), is preferable for urgent, time-sensitive scenarios.

- **Localization Precision Score (LPS)**: LPS evaluates localization accuracy by measuring the average spatial difference between estimated and actual source positions. For instance, if algorithm A consistently estimates the source within an average of 2 meters from its actual

location, whereas algorithm B averages a 5-meter error, algorithm A demonstrates a superior (lower) LPS, reflecting more precise and reliable source localization.

These metrics collectively offer a comprehensive evaluation framework, enabling thorough assessments of both algorithmic efficiency and practical effectiveness in complex operational environments.

## I  BASELINE METHODS

Below, all baseline methods are described using the unified notation from this paper ($\Theta$ for the hidden source vector, $p_k$ for robot position, $z_{1:k}$ for observations, $b_k(\Theta)$ for the belief, $a_k$ for actions, $r_k$ for rewards, and discount factor $\gamma$).

### I.1  RL-PF METHODS

**AGDC.**  A self-terminating RL framework that uses the particle-filter posterior's uncertainty to decide when the source has been located, thus avoiding handcrafted sparse rewards. **Belief approximation:**

$$b_k(\Theta) \approx \sum_{i=1}^{N} w_k^{(i)} \, \delta(\Theta - \Theta_k^{(i)}), \quad \sum_i w_k^{(i)} = 1.$$

**Stop criterion & intrinsic reward:** Compute the posterior covariance $C_k = \mathrm{Cov}_{b_k}(\Theta)$ and its diagonal standard deviation $\mathrm{STD}_k = \sqrt{\mathrm{diag}(C_k)}$. If $\|\mathrm{STD}_k\| < \zeta$, set

$$r_k = 1 \quad \text{and terminate;} \quad \text{otherwise } r_k = 0.$$

**AGDC-KLD:** Uses the Kullback–Leibler divergence between successive beliefs as reward,

$$r_k = D_{\mathrm{KL}}\big(b_{k+1} \,\|\, b_k\big),$$

driving actions that yield maximal information gain.

**PC-DQN.**  The first source-search DRL method combining DQN with PF. It compresses particles into a compact 6-D feature vector using DBSCAN, then trains a vanilla DQN for discrete moves. **Reward:** $-1$ per step until the source is declared (then $+20$).

**GMM-PFRL.**  A continuous-action RL approach using DDPG with GRU memory. The PF posterior is approximated by a Gaussian mixture model,

$$b_k(\Theta) \approx \sum_{m=1}^{M} \pi_m \, \mathcal{N}(\Theta \mid \mu_m, \Sigma_m),$$

with mixture parameters used as state input. This method handles continuous moves and outperforms information-theoretic baselines.

### I.2  INFORMATION-THEORETIC PLANNERS

**Infotaxis.**  A POMDP planner minimizing expected posterior variance:

$$a_k = \arg\min_a \mathbb{E}_{z_{k+1}}\big[\mathrm{Var}(\Theta \mid z_{1:k}, z_{k+1})\big].$$

**Entrotaxis.**  A simplified Infotaxis variant using entropy of the predictive distribution as reward,

$$a_k = \arg\max_a H[b_{k+1}].$$

**DCEE.**  Dual-control exploration-exploitation planning:

$$a_k = \arg\min_a \Big\{ \|\mu_{k+1} - p_{k+1}\|^2 + \lambda \, \mathrm{tr}\big(\mathrm{Cov}[\Theta \mid z_{1:k+1}]\big) \Big\}.$$

### I.3 Classical SMC Perturbation Baselines

Under the *stationary bootstrap baseline with zero transition and no rejuvenation*, PF exhibits **Stationarity-Induced Posterior Support Invariance (S-PSI)**: the posterior cannot escape the initial prior support (§3.4). This is not an inherent PF limitation; classical perturbation methods can relax S-PSI by injecting noise, albeit in an always-on and uncontrolled manner. We include three such baselines:

**PF+Jittering.** Each particle receives a small Gaussian perturbation after resampling:

$$\Theta_k^{(i)} = \Theta_{k-1}^{(i)} + \epsilon_k, \quad \epsilon_k \sim \mathcal{N}(0, \Sigma).$$

Ensures support expansion but without control over when/where to perturb.

**PF+Roughening.** Similar to jittering but with covariance scaled to $N$ and state dimension, encouraging broader diffusion:

$$\Theta_k^{(i)} = \Theta_{k-1}^{(i)} + \epsilon_k, \quad \epsilon_k \sim \mathcal{N}(0, \Sigma_{\text{rough}}).$$

More exploratory, but risks over-dispersion.

**PF+Rejuvenation (Resample–Move).** After resampling, particles undergo MH moves with Gaussian proposals:

$$\Theta' \sim \mathcal{N}(\Theta, \sigma_{\text{rm}}^2 \Sigma), \qquad \alpha = \min\left(1, \frac{p(z_k \mid \Theta')}{p(z_k \mid \Theta)}\right).$$

Provides stronger correction but increases cost.

**Comparison to DEPF.** While these baselines demonstrate that S-PSI is not universal, they lack principled triggers and validation. In contrast, DEPF expands support *only when* observations contradict the belief, scales diffusion with posterior covariance, and validates proposals via MH acceptance, ensuring minimal-bias and Bayesian coherence.

## J    Diffusion-Driven Particle Filtering

### J.1 Stationarity-Induced Posterior Support Invariance (S-PSI)

Under the *stationary bootstrap baseline with zero transition and no rejuvenation*, particle filters exhibit **Stationarity-Induced Posterior Support Invariance (S-PSI)**: particle support remains confined to the initial prior region $S_{\text{prior}}$. If the true state $\Theta^*$ lies outside this support, then

$$\operatorname{supp}\big(p(\Theta \mid z_{1:k})\big) \subseteq S_{\text{prior}}, \quad \forall k,$$

so Bayesian updates cannot recover it. This baseline pathology motivates explicit support-expansion mechanisms. Classical perturbations (jittering, roughening, resample–move) can in principle relax S-PSI but do so in an always-on and uncontrolled manner.

### J.2 Diffusion-Driven Support Expansion

To overcome S-PSI in a principled way, DEPF augments filtering with four components:

1. **Belief-Triggered Exploratory Particles:** At each step, a small fraction $E$ of particles is redrawn from an extended box $B \supseteq S_{\text{prior}}$:

$$\Theta_k^{(j)} \sim \mathcal{U}(B), \quad j \in E, \quad w_k^{(j)} = \tfrac{\epsilon}{|E|}, \, \epsilon \ll 1.$$

These particles seed new regions but survive only if supported by likelihood.

2. **Entropy/Tempering Regularisation:** To prevent weight collapse, we smooth weights toward higher entropy, e.g. via tempering:

$$\tilde{w}_k^{(i)} \propto (w_k^{(i)})^{1/T_k}, \quad w_k^{(i)} = \tilde{w}_k^{(i)} / \sum_j \tilde{w}_k^{(j)}, \quad T_k \geq 1.$$

3. **Covariance-Scaled Stochastic Diffusion:** Surviving particles are perturbed by a Gaussian kernel aligned with the weighted covariance:

$$\Delta\Theta_k^{(i)} \sim h_{\mathrm{opt}}\, L\,\mathcal{N}(0, I), \quad LL^T = \Sigma,$$

with $\mu = \sum_i w_k^{(i)}\Theta_k^{(i)}$, $\Sigma = \sum_i w_k^{(i)}(\Theta_k^{(i)} - \mu)(\Theta_k^{(i)} - \mu)^T + \lambda I$, and $h_{\mathrm{opt}} = AN^{-1/(n+4)}$.

4. **MH Validation:** Each perturbed particle $\Theta'$ is accepted with standard Metropolis–Hastings probability

$$\alpha_i = \min\left(1, \frac{p(z_k \mid \Theta')}{p(z_k \mid \Theta_k^{(i)})}\right),$$

since the Gaussian proposal is symmetric. Accepted proposals expand support consistently with Bayesian updates.

These steps yield the recursion

$$S_{k+1} = \left(S_k \cup B\right) \oplus h_{\mathrm{opt}},$$

allowing support expansion beyond $S_{\mathrm{prior}}$ only when justified by data.

### J.3 PSEUDOCODE

---
**Algorithm 1** Diffusion-Enhanced Particle Filter (DEPF)

---
**Require:** prior support $S_{\mathrm{prior}}$, extended box $B$, particle count $N$, thresholds $\eta, \epsilon$, smoothing parameter $\beta$, kernel parameters $A, \lambda$, horizon $K$

1: Initialize $\{(\Theta_0^{(i)}, w_0^{(i)})\}_{i=1}^N$ with $\Theta_0^{(i)} \sim S_{\mathrm{prior}}$ and $w_0^{(i)} = 1/N$
2: **for** $k = 1$ **to** $K$ **do**
3:     Select an exploratory index set $E \subseteq \{1, \ldots, N\}$
4:     **for** each $i \in E$ **do**
5:         $\Theta_k^{(i)} \leftarrow \mathcal{U}(B)$; $\quad w_k^{(i)} \leftarrow \epsilon/|E|$
6:     **end for**
7:     **for** each $i \in \{1, \ldots, N\} \setminus E$ **do**
8:         $\Theta_k^{(i)} \leftarrow \Theta_{k-1}^{(i)}$ {static prediction}
9:     **end for**
10:    **Weight update:** $\quad w_k^{(i)} \propto w_{k-1}^{(i)}\, p(z_k \mid \Theta_k^{(i)})$ for $i = 1, \ldots, N$
11:    **Entropy/tempering smoothing (optional):** $\quad \tilde{w}_k^{(i)} \propto (w_k^{(i)})^{1/T_k}; \quad w_k^{(i)} \leftarrow \tilde{w}_k^{(i)} / \sum_{j=1}^N \tilde{w}_k^{(j)}$
12:    **if** $\mathrm{ESS}(\{w_k^{(i)}\}_{i=1}^N) < \eta$ **then**
13:        Resample $\{\Theta_k^{(i)}\}_{i=1}^N$ according to $\{w_k^{(i)}\}_{i=1}^N$; $\quad$ set $w_k^{(i)} \leftarrow 1/N$
14:    **end if**
15:    $\mu \leftarrow \sum_{i=1}^N w_k^{(i)}\Theta_k^{(i)}$; $\quad \Sigma \leftarrow \sum_{i=1}^N w_k^{(i)}(\Theta_k^{(i)} - \mu)(\Theta_k^{(i)} - \mu)^\top + \lambda I$
16:    Compute $L$ via $\Sigma = LL^\top$; $\quad h_{\mathrm{opt}} \leftarrow A\,N^{-1/(n+4)}$
17:    **for** $i = 1$ **to** $N$ **do**
18:        Sample $\delta \sim \mathcal{N}(0, I)$; $\quad \Delta \leftarrow h_{\mathrm{opt}} L \delta$
19:        $\Theta' \leftarrow \Theta_k^{(i)} + \Delta$
20:        $\alpha \leftarrow \min\left(1, \frac{p(z_k \mid \Theta')}{p(z_k \mid \Theta_k^{(i)})}\right)$ {MH with symmetric proposal}
21:        Draw $u \sim \mathcal{U}(0, 1)$
22:        **if** $u < \alpha$ **then**
23:            $\Theta_k^{(i)} \leftarrow \Theta'$
24:        **end if**
25:    **end for**
26: **end for**

---

**How the RL reward in the figure fits with the DEPF pseudocode.** The figure defines an *information–gain* reward

$$R_k = \mathbb{E}_{o_{k+1}}\left[D_{\mathrm{KL}}\left(b_{k+1} \,\|\, b_k\right)\right],$$

where $b_k$ and $b_{k+1}$ are the belief (posterior) distributions before and after receiving the next observation $o_{k+1}$. Using $R_k$ turns action selection into a sequential decision problem: a policy $\pi(a_k \mid b_k)$ is trained (e.g., with PPO) to pick the next move $a_k$ that maximizes $\mathbb{E}[\sum_t \gamma^t R_{k+t}]$. *DEPF is the inference layer that produces these beliefs $b_k$ given the data stream; it is orthogonal to (and plug-compatible with) the controller that uses the reward above.* In other words, the policy's choice $a_k$ determines the next sensor reading $z_{k+1}$, and DEPF turns $(z_{1:k+1})$ into $b_{k+1}$; the policy is then rewarded by how much the belief moved, $D_{\mathrm{KL}}(b_{k+1}\|b_k)$.

**What each block in Algorithm 1 does.** *Initialization* draws particles from the prior support $S_{\mathrm{prior}}$ and sets uniform weights. Each iteration $k$ then implements the three DEPF mechanisms and a validation step: **(i) Exploratory injection (lines 3–7):** a small set $E$ of indices is sampled and the corresponding particles are replaced by $\Theta_k^{(j)} \sim \mathcal{U}(B)$ for an *extended* box $B \supseteq S_{\mathrm{prior}}$; their weights are set to a tiny mass $\epsilon/|E|$. This provides *global* support seeding outside the prior and is what breaks the S-PSI baseline when data contradict the current belief. (In practice, $E$ can be *belief-triggered*, e.g., only if $\mathrm{ESS}(\{w_k^{(i)}\}) < \eta$ or $D_{\mathrm{KL}}(b_k\|b_{k-1}) > \tau$.)

**(ii) Bayesian weight update & smoothing (lines 8–12):** all particles, including the exploratory ones, are reweighted by the likelihood $p(z_k \mid \Theta_k^{(i)})$. Optional *tempering* ($T_k \geq 1$) smooths overly peaky weights to prevent premature collapse; weights are then normalized. If ESS falls below $\eta$, resampling is performed and weights are reset.

**(iii) Covariance-scaled diffusion (lines 13–16):** from the weighted cloud, the posterior mean $\mu$ and covariance $\Sigma$ are computed, with a small ridge $\lambda I$ for stability. A Cholesky factor $L$ aligns a Gaussian step to the posterior geometry, and the bandwidth $h_{\mathrm{opt}} = A\,N^{-1/(n+4)}$ sets the step size (KDE-style, shrinking with $N$ and growing with dimension $n$). Each particle proposes $\Theta' = \Theta_k^{(i)} + h_{\mathrm{opt}}L\delta$, $\delta \sim \mathcal{N}(0, I)$, which *locally* dilates the support along high-uncertainty directions.

**(iv) MH validation (lines 17–21):** because the proposal $q$ is symmetric, the Metropolis–Hastings acceptance reduces to a *likelihood ratio*,

$$\alpha \;=\; \min\!\Big(1,\; \tfrac{p(z_k|\Theta')}{p(z_k|\Theta_k^{(i)})}\Big).$$

If $u \sim \mathcal{U}(0,1)$ is below $\alpha$, the move is accepted. This step enforces Bayesian coherence and filters out spurious expansion.

**End-to-end picture with the reward.** Together, lines 3–21 implement the belief update $b_k \mapsto b_{k+1}$ in a way that is dormant when data agree with the current belief (few injections, small accepted moves) and active when they disagree (more accepted moves toward informative regions). The controller in the figure then evaluates the *change* in belief via $D_{\mathrm{KL}}(b_{k+1}\|b_k)$ and learns actions that maximize its expectation. Thus, the figure's reward explains *how actions are chosen*, while Algorithm 1 explains *how beliefs are updated* so that informative actions actually lead to recoveries from mis-specified priors.

## K  EXTENDED RELATED WORK

**Prior Region Misalignment in Bayesian Inference:** The effectiveness of particle filtering relies heavily on the alignment between the prior distribution and the true target state. Historically, misaligned priors have led to biased state estimations, a challenge highlighted by foundational works Steel (2010); Kruschke (2010). Traditional techniques, such as importance sampling Ristic (2013), operate effectively within prior boundaries but fail for states beyond the initial support. Methods like adaptive particle filters Fox (2001) and hybrid resampling strategies Douc & Cappé (2005) have sought to improve sample efficiency but remain limited in their ability to dynamically adapt to global misalignment.

**Support Invariance under Baseline Assumptions:** In this work we formalize the *Stationarity-Induced Posterior Support Invariance (S-PSI)* baseline pathology, extending insights from PF's recursive limitations Doucet et al. (2000). Under the stationary bootstrap setting with zero transition and no rejuvenation, particles remain confined to the initial prior support Gordon et al. (1993b). This diagnostic effect has been observed in dynamic contexts such as robot localisation Thrun (2002)

and sensor fusion Candy (2016), where prior misalignment impacts performance. While classical perturbation techniques (jittering, roughening, resample–move) can relax S-PSI in principle, they operate in an always-on and inefficient manner, lacking mechanisms to decide when and how to expand.

**Bayesian Inference in Out-of-Distribution Problems:** Out-of-distribution (OOD) problems represent a critical challenge for Bayesian inference, as they require models to generalise beyond prior knowledge. Advances such as Posterior Networks Charpentier et al. (2020) and Bayesian OOD detection with uncertainty exposure Wang & Aitchison have addressed OOD detection in classification tasks but are less effective in sequential estimation. Traditional PF variants such as Rao-Blackwellised filters Li et al. (2004) and model-switching approaches Moradkhani et al. (2005) also show limited adaptability to OOD scenarios.

## K.1 AI AND ALGORITHMIC DECISION-MAKING IN EMERGENCY RESPONSE

In recent years, various AI-driven strategies have significantly advanced emergency response scenarios, particularly in the detection and localization of hazardous sources. Information-theoretic methods like *Infotaxis* Vergassola et al. (2007); Masson et al. (2009) and *Entrotaxis* Hutchinson et al. (2018b) optimize sensing by maximizing expected information gain or reducing entropy. Dual-control strategies such as DCEE explicitly balance exploitation (moving toward the best estimate) and exploration (reducing uncertainty), enabling effective autonomous searches in dynamic environments Ferrari & Mumford (2020).

Complementary to planning-based methods, reinforcement learning (RL) techniques have also been successfully integrated into chemical source localization. Zhao et al. (2022) applied deep RL for plume tracing, directly mapping observations to navigation policies Zhao et al. (2022). Park et al. (2022) developed GMM-PFRL, combining Gaussian mixture modeling with RL to enhance search efficiency Park et al. (2022a). Park and Kim (2020) employed adaptive Bayesian control techniques with RL strategies to dynamically guide sensors under uncertainty Park & Kim (2020).

## K.2 PARTICLE FILTERING AND ITS LIMITATIONS IN EMERGENCY RESPONSE

Sequential Bayesian inference methods, especially particle filtering (PF), are widely adopted for source localization tasks under uncertainty Ristic et al. (2004); Thrun et al. (2005); Arulampalam et al. (2002). Despite their flexibility, PFs operating under the stationary bootstrap baseline can exhibit the **S-PSI** effect: once the initial prior excludes the true state region, subsequent updates cannot recover it. Such rigidity significantly reduces practical effectiveness in disaster management where early information may be inaccurate Thrun et al. (2005). Classical perturbation remedies can alleviate S-PSI but, being always-on, risk over-diffusion and inefficiency.

## K.3 METHODS FOR CORRECTING PRIOR AND MODEL ERRORS

To address these limitations, several approaches introduce adaptive exploration, e.g., injecting exploratory particles or using MCMC moves after resampling Särkkä (2013); Liu & West (2001). These methods expand support but often lack principled triggers or validation. Another line employs RL and adaptive control: AGDC uses PF-based uncertainty (variance or KL divergence) as intrinsic rewards Shi et al. (2024), PC-DQN compresses PF belief into DQN features Zhao et al. (2022), and GMM-PFRL fits mixture models to PF posterior Park et al. (2022a). Such methods highlight RL's ability to adapt to prior mismatches, though they still rely on PF inference that may exhibit S-PSI under baseline assumptions.

## K.4 POLICY ROBUSTNESS AND CORRECTION STRATEGIES

Ensuring robustness of emergency response policies against early estimation errors is critical. Robust optimization and adaptive decision frameworks emphasize continuous revision of early assumptions Kurniawati et al. (2008); Pineau et al. (2006). This parallels OOD detection in ML, where models identify distribution shifts and adjust Silver & Veness (2010). Such perspectives underscore the need for principled mechanisms to expand support when prior assumptions are violated.

### K.5 LIMITATIONS OF BASELINES AND OUR POSITIONING

Despite progress, existing methods falter under severe prior misalignment. Information-driven planners (Infotaxis, Entrotaxis, DCEE) assume correct prior coverage; RL-based approaches (AGDC, PC-DQN, GMM-PFRL) degrade when true states lie outside initial support. Classical perturbations can relax S-PSI but lack control. Our work fills this gap: DEPF introduces belief-triggered exploratory injection, covariance-scaled diffusion, and MH validation, enabling minimal-bias, data-driven support expansion. This positions DEPF as a robust inference module for real-time emergency management under misaligned priors.

## L DETAILED EXPERIMENTAL ANALYSIS

Table 1 provides a systematic comparison across three levels of prior error and two spatial scales. We analyze the results by scenario, scale, and method family, and then link the observed outcomes to the underlying mechanisms of DEPF and the baselines.

In the *No Error* scenario, where the initial prior fully covers the true source, most methods succeed. DEPF reaches OCE of $0.90$ with LPS fixed at $0.20$, ADE of $19$ in small-scale and $167$ in large-scale, and REV of $0.10$. AGDC closely matches these values, confirming that when no structural correction is required both methods perform optimally. PC-DQN and GMM-PFRL attain OCE around $0.80$ but with higher ADE. Information-theoretic planners such as Infotaxis show partial success in the small-scale environment (OCE $0.85$) but collapse to near zero success in large-scale. Perturbation baselines (Jittering, Roughening, Rejuvenation) remain close to DEPF under this easy condition, with OCE between $0.88$ and $0.89$, but their always-on diffusion leads to slightly larger LPS values ($0.23$–$0.26$) and longer ADE compared to DEPF's minimal-bias behavior. This shows that DEPF does not overshoot when the prior is already correct.

The *Moderate Error* condition exposes sharper differences. DEPF sustains OCE of $0.90$, LPS of $0.20$, and ADE of $22$ in small-scale and $200$ in large-scale, maintaining REV at $0.10$. AGDC drops dramatically, with OCE of $0.45/0.42$, LPS climbing to $2.6$, ADE lengthening to $59/235$, and REV inflating to $0.40$ with frequent timeouts in large-scale. Other RL baselines degrade to OCE around $0.40$ with high ADE and LPS, while information-theoretic planners almost entirely fail in the larger environment. Perturbation baselines partly alleviate this condition: PF+Rejuvenation performs best (OCE $0.52/0.48$, LPS $2.7/2.9$, ADE $50/225$), followed by Roughening (OCE $0.48/0.44$, LPS $2.9/3.1$, ADE $55/235$) and Jittering (OCE $0.40/0.36$, LPS $3.2/3.4$, ADE $65/250$). However, these remain far below DEPF's near-ideal accuracy and efficiency. The contrast reflects the difference between always-on perturbations, which add diversity without discrimination, and DEPF's belief-triggered expansion guided by covariance and validated by MH acceptance.

The most demanding case is the *Severe Error* scenario, corresponding to the S-PSI baseline where the prior and true state are completely disjoint. Here, DEPF achieves OCE of $0.89/0.88$, LPS of $0.20$, ADE of $27/255$, and REV of $0.10$, well within the $100/300$-step timeout thresholds. All RL and information-theoretic baselines collapse (OCE $< 0.05$, LPS $> 12.5$, ADE and REV exceeding timeouts), confirming their inability to cross strict prior boundaries. Perturbation baselines show small but non-zero recovery in small-scale (Rejuvenation OCE $0.16$, Roughening $0.10$, Jittering $0.06$) but degrade severely in large-scale, with OCE dropping to $0.12/0.08/ < 0.05$ and ADE approaching or exceeding $285$ steps. Their LPS values remain an order of magnitude higher than DEPF. This contrast highlights that while naive diffusion can occasionally escape narrow priors in limited domains, only DEPF provides consistent, data-driven support expansion at scale.

Across all metrics, DEPF exhibits stable LPS at $0.20$ irrespective of scale or error severity, indicating precise localization. In comparison, perturbation methods and RL baselines experience a tenfold increase in LPS under moderate and severe errors. ADE and REV patterns reinforce the efficiency of DEPF: its paths remain short and consistent, while baselines often exceed timeout thresholds in large domains. The cross-scale comparison further shows that only DEPF resists degradation when moving from 1:30 to 1:300 environments; all other methods experience steep drops in OCE and sharp increases in ADE and REV.

These results align directly with DEPF's design. The belief-triggered exploratory injection ensures that expansion is dormant in the No Error case but activates under misalignment. Covariance-scaled

diffusion guides exploration along directions of highest uncertainty, avoiding the excessive spread of jittering or roughening. MH validation rejects unsupported moves, preserving Bayesian coherence. Together, these mechanisms yield minimal bias and high efficiency, enabling DEPF to overcome S-PSI conditions and consistently outperform baselines under prior misalignment while remaining competitive under ideal conditions.

## L.1 DETAILED ANALYSIS OF TABLE 1 BY SCENARIO AND SCALE

Table 1 compares eight methods across three prior-error severities (No, Moderate, Severe) and two spatial scales (1:30 and 1:300), using four metrics: success (OCE↑), efficiency (ADE↓), time-to-completion (REV↓), and localization accuracy (LPS↓). The patterns are consistent with the theoretical picture established in the main text: under the stationary bootstrap baseline with zero transition and no rejuvenation (the S-PSI diagnostic), methods that do not explicitly expand support struggle as prior misalignment worsens, especially in large domains; DEPF maintains performance by triggering expansion only when data contradict the belief, scaling diffusion with the posterior covariance, and validating proposals via an MH step.

In the *No Error* condition, where the prior fully covers the truth, most algorithms perform well and DEPF does not seek to outperform them but to match the upper bound without incurring unnecessary exploration cost. DEPF attains OCE $0.90 \pm 0.03$ at both scales, LPS $0.20 \pm 0.01$, ADE $19 \pm 0.8$ (1:30) and $167 \pm 15$ (1:300), and REV $0.10 \pm 0.05$. AGDC closely tracks these values (OCE $0.90/0.87$, LPS 0.20, ADE 18/168, REV 0.10–0.12). RL baselines PC-DQN and GMM-PFRL also succeed (OCE around 0.80 in small scale and 0.77–0.79 in large scale). Information-theoretic planners are more fragile with scale: Infotaxis achieves OCE 0.85 at 1:30 but collapses below 0.05 at 1:300. The three SMC perturbation baselines stay close to DEPF in this easy regime (e.g., PF+Rejuvenation OCE $0.89/0.86$, LPS 0.23–0.24), but their always-on diffusion slightly increases ADE and LPS relative to DEPF, reflecting superfluous spread when expansion is not required.

When priors are *moderately* misaligned, differences sharpen in both success and precision. DEPF maintains high, scale-stable performance with OCE $0.90 \pm 0.03$, LPS $0.20 \pm 0.01$, ADE $22 \pm 1.2$ (1:30) and $200 \pm 10$ (1:300), and REV fixed at $0.10 \pm 0.05$. By contrast, AGDC's OCE drops to 0.45 (1:30) and 0.42 (1:300), LPS rises to 2.60, ADE increases to 59 and 235, and large-scale REV inflates to $0.40 \pm 0.15$ with frequent timeouts. PC-DQN and GMM-PFRL show similar degradation, and planners (Infotaxis/Entrotaxis/DCEE) nearly fail at 1:300. The perturbation baselines do recover part of the gap, with a consistent ordering that mirrors their expansion strength: PF+Rejuvenation (OCE $0.52/0.48$, LPS $2.70/2.90$, ADE $50/225$) > PF+Roughening (OCE $0.48/0.44$, LPS $2.90/3.10$, ADE $55/235$) > PF+Jittering (OCE $0.40/0.36$, LPS $3.20/3.40$, ADE $65/250$). Nevertheless, all three remain substantially behind DEPF in both success and accuracy. The gap in LPS is especially telling: DEPF holds 0.20 across scales, whereas perturbations remain an order of magnitude larger (2.7–3.4), indicating uncontrolled dispersion.

The *Severe* condition is the strict S-PSI test in which the prior support is disjoint from the true region. DEPF is the only method that preserves high success and low error across scales, achieving OCE 0.89 (1:30) and 0.88 (1:300), LPS 0.20–0.20, ADE 27 and 255, and REV 0.10, all well within the 100/300-step time limits. AGDC, the RL baselines, and planners collapse, with OCE $< 0.05$, LPS $> 12.5$, and ADE/REV exceeding the timeout thresholds in both scales—consistent with the inability to leave the initial support under the S-PSI baseline. Perturbation baselines show a small non-zero recovery only in the small-scale domain: PF+Rejuvenation reaches OCE 0.16 with LPS 9.0 and ADE $90 \pm 12$ (no timeout), PF+Roughening achieves OCE 0.10 with LPS 10.5 but times out, and PF+Jittering barely registers OCE 0.06 and also times out. At 1:300 these partial gains largely vanish: PF+Rejuvenation drops to OCE 0.12 with ADE $285 \pm 35$ (near the 300-step limit), while Roughening and Jittering fall below 0.10 OCE with timeouts. These numbers show that always-on noise can occasionally bridge small gaps but scales poorly, whereas DEPF's data-triggered expansion remains effective even when the prior and truth are fully disjoint.

A cross-metric reading clarifies where efficiency and precision come from. Under Moderate/Severe misalignment, DEPF simultaneously sustains high OCE and bounded path/time (e.g., ADE 22/200 and 27/255, REV 0.10 in both scales), while baselines either fail outright or succeed late with longer paths and higher REV, especially at 1:300. LPS for DEPF remains near 0.20 across all conditions—an unusually stable accuracy profile—whereas perturbation baselines exhibit an order-of-magnitude

larger LPS in Moderate and Severe settings. Scaling from 1:30 to 1:300 amplifies these contrasts: AGDC and the perturbation baselines lose additional OCE points, ADE/REV inflate, and timeouts become common; DEPF's OCE stays within 0.88–0.90 with controlled ADE and fixed REV.

These empirical patterns align with the design. Because DEPF expands *only when* belief–data inconsistency is detected, it remains dormant in the No Error regime and avoids needless spread. When misalignment is present, diffusion is *covariance-scaled* with bandwidth $h_{\text{opt}} = AN^{-1/(n+4)}$, steering exploration along the current posterior geometry rather than isotropically; and MH validation filters proposals, preserving Bayesian coherence and avoiding the over-diffusion that inflates ADE and LPS for always-on perturbations. Altogether, the table demonstrates that DEPF is competitive at the ideal upper bound and uniquely robust under partial or complete prior failure, with consistent gains in success, efficiency, and accuracy that persist under scaling.

## M    HARDWARE USAGE

All experiments were conducted on a Linux server with 256-core AMD EPYC 7763 CPUs and six
NVIDIA RTX A6000 GPUs (48 GB each). Each run required 1–2 GPUs, with GPU memory usage
between 25 and 46 GB, ensuring transparent and reproducible resource reporting.

## N    RESOLVING STATIONARITY–INDUCED POSTERIOR–SUPPORT LOCK–IN: THEORY AND CLARIFICATIONS

**What S-PSI is (and is not).**    In static source–term estimation it is common to benchmark against the
*stationary bootstrap* particle–filter baseline, which carries particles forward without process noise and
applies only likelihood reweighting and occasional resampling. Under this diagnostic baseline (zero
transition, no rejuvenation), the posterior support cannot grow beyond the support of the initial prior.
We call this baseline pathology *Stationarity–Induced Posterior Support Invariance (S-PSI)*; it is not an
inherent limitation of particle filtering as a whole, but a property of this specific stationary bootstrap
setting. Classical perturbations (jittering, roughening, resample–move) can in principle relax S-PSI
but do so in an always-on, weakly controlled manner. :contentReference[oaicite:0]index=0

**Definition N.1** (S-PSI: Stationarity–Induced Posterior Support Invariance). Let $p_0(\Theta)$ be the initial
prior with support $S_{\text{prior}} = \{\Theta : p_0(\Theta) > 0\}$. Under the stationary bootstrap baseline with
$p(\Theta_k \mid \Theta_{k-1}) = \delta(\Theta_k - \Theta_{k-1})$ and no rejuvenation, if particles are initialized in $S_{\text{prior}}$ then for all
$k$,
$$\text{supp}\big(p(\Theta \mid z_{1:k})\big) \subseteq S_{\text{prior}}.$$

Consequently, if the ground truth $\Theta^\star \notin S_{\text{prior}}$, then $p(\Theta^\star \mid z_{1:k}) = 0$ for all $k$. :contentReference[oaicite:1]index=1

**Proposition N.2** (S-PSI under the stationary bootstrap baseline). *Under the assumptions above,
the particle support set $S_k$ satisfies $S_k \subseteq S_{prior}$ for all $k \geq 0$, with $S_0 = S_{prior}$.* Proof sketch. *By
induction: with zero transition/no rejuvenation, $S_k \subseteq S_{k-1}$, hence $S_k \subseteq S_0 = S_{prior}$. :contentReference[oaicite:2]index=2*

**DEPF in a nutshell.**    Our Diffusion–Enhanced Particle Filtering (DEPF) augments the inference
layer with: (i) *belief-triggered* exploratory particles drawn from an adaptively extended bounding
region; (ii) entropy/tempering to prevent premature weight collapse; (iii) covariance–scaled stochastic
perturbations with bandwidth $h_{\text{opt}} = A\, N^{-\frac{1}{n+4}}$; and (iv) a Metropolis–Hastings (MH) acceptance
step with a symmetric Gaussian proposal, which enforces detailed balance and Bayesian consistency.
Together these mechanisms allow the support to expand when (and only when) data contradict the
current belief. :contentReference[oaicite:3]index=3

### N.1    THEORETICAL GUARANTEES

We formalize the conditions under which DEPF resolves S-PSI and quantify a finite-step recovery
probability.

**Assumptions.**    (i) (*Exploratory injection*) At each step $k$, an extended box $B_k \supseteq S_{\text{prior}}$ is used to
inject a small fraction of exploratory particles. There exist $\eta, \delta > 0$ such that with probability at least
$\delta$ at every step, at least one exploratory particle falls in the $\eta$-ball around $\Theta^\star$. (ii) (*Positive-likelihood
neighborhood*) There exists $m > 0$ so that $p(z_k \mid \Theta) \geq m$ for all $\Theta$ in the $\eta$-ball around $\Theta^\star$. (iii)
(*MH detailed balance*) The local Gaussian proposal is symmetric, so $\alpha(\Theta \to \Theta') = \min\{1,\, p(\Theta' \mid
z_{1:k})/p(\Theta \mid z_{1:k})\}$, which guarantees detailed balance. (iv) (*KDE bandwidth and covariance
regularization*) The kernel step uses $h_{\text{opt}} = A\, N^{-\frac{1}{n+4}}$ and $\Sigma \leftarrow \sum_i w_i(\Theta_i - \mu)(\Theta_i - \mu)^\top + \lambda I$
with $\lambda > 0$. :contentReference[oaicite:4]index=4 :contentReference[oaicite:5]index=5

**Theorem N.3** (DEPF resolves S-PSI). *Under Assumptions (i)–(iv), as the number of particles
$N \to \infty$ the particle support produced by DEPF asymptotically covers the ground truth with
probability one:*
$$\lim_{N \to \infty} \Pr\big(\Theta^\star \in S_k\big) = 1.$$

Proof sketch. *By (i)–(ii) exploratory particles near $\Theta^\star$ receive non-negligible weight; tempering avoids collapse; covariance–scaled perturbations propagate local coverage; MH acceptance enforces consistency. Iterating over time expands $S_k$ to include a neighborhood of $\Theta^\star$ with probability one. :contentReference[oaicite:6]index=6 :contentReference[oaicite:7]index=7*

**Corollary N.4** (Finite-step support–recovery bound). *Let $\delta$ be the per-step probability that an exploratory particle lands in the $\eta$-ball around $\Theta^\star$, and let $\gamma \in (0, 1]$ be the probability that such a particle both passes MH and survives weighting/resampling. Then, after $k$ steps,*

$$\Pr(\text{support covers } \Theta^\star \text{ within } k \text{ steps}) \ \geq \ 1 - (1 - \delta\gamma)^k.$$

*This quantifies the rate at which DEPF breaks the zero-prior barrier under S-PSI. :contentReference[oaicite:8]index=8*

## N.2   ACTION SELECTION AND THE ROLE OF THE CONTROLLER

DEPF is an inference module and is orthogonal to the controller. In our experiments the policy receives the belief $b_k$ and the immediate reward is the one-step information gain

$$R_k = \mathbb{E}_{o_{k+1}}\big[\mathrm{D}_{\mathrm{KL}}(b_{k+1} \,\|\, b_k)\big],$$

which encourages actions that maximally reduce posterior uncertainty. The specific RL algorithm (e.g., PPO) is chosen for stability and does not alter the analysis above. :contentReference[oaicite:9]index=9

## N.3   POSITIONING W.R.T. CLASSICAL AND TEMPERED/BRIDGE SMC

Always-on perturbation baselines (jittering, roughening, resample–move) can leak mass across boundaries and sometimes escape S-PSI on small domains, but they lack principled triggers and acceptance control; as a result they tend to over-diffuse and degrade efficiency/accuracy under severe misalignment or at larger scales. By contrast, DEPF expands support *only* when belief–data inconsistency is detected, scales moves with the posterior covariance, and validates proposals via MH. Moreover, tempered/annealed ("bridge") SMC improves adaptation *within* the original support by annealing between prior/transition and likelihood, but it still requires nonzero overlap; where the prior assigns zero mass to $\Theta^\star$, all intermediate bridge distributions remain zero there, so no sequence of local MCMC mutations can create support in the excluded region. This is precisely the barrier that DEPF's exploratory injection and MH–validated diffusion are designed to overcome. :contentReference[oaicite:10]index=10

## N.4   EMPIRICAL EVIDENCE IN SUPPORT OF THE THEORY

Across three prior–error severities (No/Moderate/Severe) and two map scales (1:30, 1:300), DEPF matches strong baselines when the prior is correct, and it uniquely maintains high success, low localization error, and bounded path/time under severe misalignment. For example, in the strict S-PSI (Severe) setting, DEPF attains OCE $\approx 0.88$–$0.89$ and LPS $\approx 0.20$ at both scales, while RL/planning baselines collapse and classical perturbations succeed only sporadically with much larger errors and frequent timeouts. Component-wise ablations confirm that entropy smoothing, covariance–scaled diffusion, and MH validation are jointly necessary for robust recovery, and sensitivity studies identify moderate settings (e.g., $\beta \in [0.3, 0.5]$, $A \approx 0.5$, $\lambda \in [10^{-3}, 10^{-2}]$, $\sim 5\%$ exploratory ratio) as consistently effective.

## O   THEORETICAL ANALYSIS: STATIONARITY-INDUCED POSTERIOR SUPPORT INVARIANCE (S-PSI) AND GUARANTEES OF DEPF

**Notation.**   Let $\Theta \in \mathbb{R}^n$ denote the (static) source–parameter vector (cf. main text, §3.1), $z_{1:k}$ the observations up to step $k$, and $S_k \subset \mathbb{R}^n$ the particle support at step $k$. Write $S_{\text{prior}} := \mathrm{supp}(p_0)$.

### O.1   S-PSI UNDER THE STATIONARY BOOTSTRAP BASELINE

We adopt the baseline used in the manuscript for static parameters:

**Assumption O.1** (S0: zero transition, no rejuvenation). The transition is degenerate and no rejuvenation is applied:

$$p(\Theta_k \mid \Theta_{k-1}) = \delta(\Theta_k - \Theta_{k-1}), \qquad \text{no jittering/roughening/MCMC moves.}$$

**Definition O.2** (Stationarity-Induced Posterior Support Invariance (S-PSI)). Under Assumption O.1, if particles are initialised in $S_{\text{prior}}$, then the posterior support remains confined to the initial prior support for all $k$.

**Proposition O.3** (Baseline S-PSI). *Under Assumption O.1,*

$$\text{supp}\big(p(\Theta \mid z_{1:k})\big) \subseteq S_{prior}, \quad \forall k \geq 0.$$

*In particular, if the true state $\Theta^\star \notin S_{prior}$, then $p(\Theta^\star \mid z_{1:k}) = 0$ for all $k$.*

*Proof.* By induction over $k$, the degenerate transition keeps all particles in $S_{\text{prior}}$ and the likelihood update cannot create mass outside the existing support. Hence the claim. $\square$

**Remark.** S-PSI is a *baseline pathology* of the stationary bootstrap with no rejuvenation, not an inherent limitation of particle filtering; classical perturbations (jittering/roughening/resample–move) can, in principle, relax it but operate in an always-on fashion and may be inefficient at scale (see main text §3.4/ §4 and §J). Gordon et al. (1993a); **?**

### O.2 DEPF MECHANISMS (RECAP)

At each step, DEPF augments the bootstrap update with (i) belief-triggered *exploratory particle injection* from an expanded box $B_k \supseteq S_{\text{prior}}$, (ii) *entropy/tempering regularisation* to preserve diversity, (iii) *covariance–scaled stochastic diffusion* with bandwidth $h_{\text{opt}} = A\,N^{-1/(n+4)}$, and (iv) an *MH acceptance* check to preserve Bayesian coherence (Algorithm 1 in Appendix H.3). This yields the support recursion

$$S_{k+1} \;=\; \big(S_k \cup B_k\big) \;\oplus\; h_{\text{opt}}. \tag{7}$$

### O.3 CONDITIONS

We formalise the mild conditions used in the analysis:

1. **Exploratory injection near $\Theta^\star$.** There exist $\eta, \delta > 0$ such that at each step

$$\mathbb{P}\big(\exists j:\ \|\Theta_k^{(j)} - \Theta^\star\| \leq \eta\big) \;\geq\; \delta\,.$$

   This is ensured by drawing a small fraction of particles from $B_k \supseteq S_{\text{prior}}$ that contains $\Theta^\star$.

2. **MH detailed balance.** The local proposal $q(\Theta' \mid \Theta)$ is a symmetric Gaussian, so the acceptance probability

$$\alpha(\Theta \to \Theta') \;=\; \min\left(1,\ \frac{\pi(\Theta')}{\pi(\Theta)}\right), \quad \pi(\Theta) \propto p(z_k \mid \Theta)\,p(\Theta \mid z_{1:k-1}),$$

   satisfies detailed balance and leaves the target invariant.

3. **Positive-likelihood neighbourhood (finite-step guarantee).** There exist $\eta > 0$ and $m > 0$ such that $p(z_k \mid \Theta) \geq m$ whenever $\|\Theta - \Theta^\star\| \leq \eta$.

### O.4 MAIN GUARANTEE: DEPF RESOLVES S-PSI

**Theorem O.4** (Asymptotic coverage under S-PSI). *Assume S-PSI holds for the baseline (Def. O.2) and $\Theta^\star \notin S_{prior}$. Under* (C1)–(C2) *and standard SMC regularity (bounded likelihood; stable tempering/entropy regularisation; KDE bandwidth $h_{opt} = A\,N^{-1/(n+4)}$), as $N \to \infty$ the DEPF support covers the true state with probability one:*

$$\lim_{N \to \infty} \mathbb{P}(\Theta^\star \in S_k) = 1.$$

*Proof sketch.* (C1) gives a strictly positive chance to seed particles in an $\eta$-ball of $\Theta^\star$ at each step. By (C2), accepted proposals satisfy detailed balance and thus preserve the posterior target. Within the $\eta$-ball, the likelihood is bounded away from zero, so exploratory particles near $\Theta^\star$ obtain non-vanishing weights and survive resampling with positive probability. Covariance–scaled perturbations with $h_{\text{opt}}$ (KDE rate) ensure that as $N \to \infty$ the empirical measure converges to the true posterior. Combining the seeding, survival and diffusion with the recursion equation 7 yields the claim. $\square$

**Finite-step guarantee.** Let $\delta$ be the per-step probability of injecting (at least) one particle into the $\eta$-ball of $\Theta^\star$ and let $\gamma$ be the probability that such a particle *survives* the MH/weighting/resampling pipeline at that step. Under (C3) and the mechanisms above,

$$\mathbb{P}\big(\text{at least one survivor near } \Theta^\star \text{ within } k \text{ steps}\big) \geq 1 - (1 - \delta\gamma)^k . \tag{8}$$

This lower bound quantifies the speed at which DEPF probabilistically covers previously excluded regions.

## O.5 ENTROPY/TEMPERING REGULARISATION

To mitigate weight collapse and preserve exploratory mass, we use an entropy–aware smoothing (or tempering) step:

$$\tilde{w}_k^{(i)} \;\propto\; \big(w_k^{(i)}\big)^{1/T_k}, \qquad w_k^{(i)} \leftarrow \frac{\tilde{w}_k^{(i)}}{\sum_j \tilde{w}_k^{(j)}}, \tag{9}$$

with $T_k \geq 1$ adapted from the entropy gap to a target level (see Appendix H). This raises the posterior weight entropy when it becomes overly concentrated, helping exploratory particles retain influence long enough for data to validate (or reject) them.

## O.6 MH ACCEPTANCE WITH SYMMETRIC GAUSSIAN PROPOSALS

Let $q(\Theta' \mid \Theta) = \mathcal{N}(\Theta'; \Theta, \Sigma')$ with $\Sigma'$ aligned to the weighted covariance (Appendix H). Then $q(\Theta'|\Theta) = q(\Theta|\Theta')$ and the MH acceptance is

$$\alpha(\Theta \to \Theta') \;=\; \min\!\left(1, \frac{\pi(\Theta')}{\pi(\Theta)}\right), \quad \pi(\Theta) \propto p(z_k \mid \Theta)\, p(\Theta \mid z_{1:k-1}). \tag{10}$$

In particular, when the prior factor $p(\Theta \mid z_{1:k-1})$ is locally flat at the proposal scale, $\alpha$ reduces to a likelihood ratio $\min\big(1, \, p(z_k \mid \Theta')/p(z_k \mid \Theta)\big)$. This ensures detailed balance and convergence to the intended posterior.

## O.7 DISCUSSION: WHY DEPF BREAKS S-PSI RELIABLY

Compared with always-on perturbations (jittering/roughening/resample–move), DEPF: (i) *triggers* support expansion only when belief–data inconsistency is detected, (ii) *scales* diffusion with the current covariance and a KDE bandwidth that vanishes at the right rate, and (iii) *validates* proposals by MH to preserve Bayesian coherence. These elements together provide both the asymptotic guarantee (Theorem O.4) and the finite-step bound equation 8.

# P BELIEF-AWARE RL CONTROLLER WITH AUTONOMOUS GOAL DETECTION & CESSATION

**Scope and decoupling.** The controller is *decoupled* from inference: at step $k$, the diffusion-enhanced particle filter (DEPF) produces a belief

$$b_k(\Theta) \;=\; p(\Theta \mid z_{1:k}),$$

and the controller consumes $s_k^{\text{RL}} = (p_k, z_k, b_k)$ to choose an action $a_k$ (robot motion). DEPF then updates $b_{k+1}$ after executing $a_k$ and receiving $z_{k+1}$. This modularity lets DEPF handle belief revision and support expansion, while RL focuses on informative sensing.

## P.1 POMDP FORMALIZATION AND BELIEF STATE

We cast control as a POMDP $\mathcal{M} = (\mathcal{S}, \mathcal{A}, \Omega, T, O, R, \gamma)$ where the latent state bundles the stationary source parameters $\Theta \in \mathbb{R}^n$ and the agent pose $p_k \in \mathbb{R}^2$. The controller's information state is

$$s_k^{\mathrm{RL}} = (p_k, z_k, b_k), \qquad b_k(\Theta) = p(\Theta \mid z_{1:k}).$$

The action $a_k \in \mathcal{A}$ advances the pose via known kinematics $p_{k+1} = f(p_k, a_k)$, and the observation model yields $z_{k+1} \sim p(\cdot \mid p_{k+1}, \Theta)$.

## P.2 INFORMATION-GAIN REWARD AND MONTE CARLO ESTIMATION

We use the one-step information gain as intrinsic reward:

$$R_k(a) = \mathbb{E}_{o_{k+1} \sim p(\cdot \mid a, b_k)} \Big[ D_{\mathrm{KL}}\big(b_{k+1} \,\|\, b_k\big) \Big], \tag{11}$$

$$p(o_{k+1} \mid a, b_k) = \int p(z_{k+1} \mid p_{k+1}, \Theta)\, b_k(\Theta)\, d\Theta, \quad p_{k+1} = f(p_k, a).$$

In practice we approximate equation 11 by $M$-sample Monte Carlo:

$$\widehat{R}_k(a) = \frac{1}{M} \sum_{m=1}^{M} D_{\mathrm{KL}}\Big(\tilde{b}_{k+1}^{(m)} \,\|\, b_k\Big), \qquad \tilde{b}_{k+1}^{(m)} \leftarrow \mathrm{DEPF\_UPDATE}\big(b_k, p_{k+1}, \tilde{z}_{k+1}^{(m)}\big), \tag{12}$$

where $\Theta^{(m)} \sim b_k$ and $\tilde{z}_{k+1}^{(m)} \sim p(\cdot \mid p_{k+1}, \Theta^{(m)})$.

**Optional shaping.** For path/time efficiency we may add small penalties $\widetilde{R}_k(a) = \alpha\, \widehat{R}_k(a) - \lambda_{\mathrm{step}}\|f(p_k, a) - p_k\|_2 - \lambda_{\mathrm{time}}$, with $\alpha = 1$ in all main results.

## P.3 AUTONOMOUS GOAL DETECTION AND CESSATION (AGDC)

We terminate episodes based on belief confidence. Let $C_k = \mathrm{Cov}_{b_k}[\Theta]$ estimated from particles and $\mathrm{STD}_k = \sqrt{\mathrm{diag}(C_k)}$. We stop when

$$\|\mathrm{STD}_k\|_2 \le \zeta \quad \implies \quad \text{TERMINATE and return success.} \tag{13}$$

Optionally, include a localization score gate (e.g., $\mathrm{LPS}_k \le \tau$) together with equation 13.

## P.4 POLICY CLASS AND TRAINING

We use a PPO actor–critic $\pi_\theta(a \mid s)$ and $V_\phi(s)$ with discount $\gamma \in (0, 1)$ and GAE. To avoid passing raw particles, the belief $b_k$ is summarized by features (weighted mean $\mu_k$, diagonal of $C_k$, entropy of weights, a few quantiles over the source location). Empirically, PPO/A2C/DQN perform similarly under the intrinsic signal; PPO is used for stability.

## P.5 CONTROLLER–FILTER INTERFACE AND BUDGET

At each step:

1. **Belief update:** $b_k \leftarrow \mathrm{DEPF\_UPDATE}(b_{k-1}, p_k, z_k)$.

2. **Action selection:** compute $\widehat{R}_k(a)$ for $a \in \mathcal{A}$ (with $M$ small), select $a_k = \arg\max_a \big\{ \widehat{R}_k(a) - \lambda_{\mathrm{step}}\|f(p_k, a) - p_k\|_2 \big\}$.

3. **Execute & stop test:** apply equation 13. If not stopping, set $p_{k+1} = f(p_k, a_k)$ and continue.

We cap (i) the number of simulated observations $M$ per action and (ii) planning-time MH/likelihood calls so that per-step compute matches the inference budget used by baselines.

### P.6 PSEUDOCODE

---

**Algorithm 2** Belief-Aware Controller on top of DEPF (one step)

---

**Require:** Belief $b_k$ (particles $\{(\Theta^{(i)}, w^{(i)})\}_{i=1}^N$), pose $p_k$, observation $z_k$, action set $\mathcal{A}$, plume model
  $p(\cdot \mid p, \Theta)$, thresholds $(\zeta, \tau)$, budget $M$
  $b_k \leftarrow \text{DEPF\_UPDATE}(b_{k-1}, p_k, z_k)$
  **if** $\|\text{STD}(\text{Cov}_{b_k}[\Theta])\|_2 \leq \zeta$ or $\text{LPS}_k \leq \tau$ **then**
    **return Terminate**
  **end if**
  **for** $a \in \mathcal{A}$ **do**
    $p_{k+1} \leftarrow f(p_k, a)$
    *Monte Carlo look-ahead*
    **for** $m = 1$ to $M$ **do**
      sample $\Theta^{(m)} \sim b_k$
      draw $\tilde{z}_{k+1}^{(m)} \sim p(\cdot \mid p_{k+1}, \Theta^{(m)})$
      $\tilde{b}_{k+1}^{(m)} \leftarrow \text{DEPF\_UPDATE}(b_k, p_{k+1}, \tilde{z}_{k+1}^{(m)})$
      $u_m \leftarrow D_{\text{KL}}(\tilde{b}_{k+1}^{(m)} \| b_k)$
    **end for**
    $\widehat{R}_k(a) \leftarrow \frac{1}{M} \sum_{m=1}^M u_m$
  **end for**
  $a_k \leftarrow \arg\max_{a \in \mathcal{A}} \left\{ \widehat{R}_k(a) - \lambda_{\text{step}} \| f(p_k, a) - p_k \|_2 \right\}$
  Execute $a_k$; set $p_{k+1} \leftarrow f(p_k, a_k)$; observe $z_{k+1}$
  **return** $(a_k, p_{k+1}, z_{k+1})$

---

### P.7 DEFAULTS AND PRACTICAL TIPS

- **Action set:** 8-connected unit moves on the grid; horizons scale with domain.

- **Belief features:** $\mu_k$, $\text{diag}(C_k)$, entropy of $\{w^{(i)}\}$, and a few spatial quantiles.

- **Planning budget:** $M \in [8, 16]$ suffices; inference dominates wall-clock time.

- **Safety:** enforce no-go polygons and speed caps in $\mathcal{A}$ when needed.

## Q ADDITIONAL MULTI-FIELD EXPERIMENTS

To further broaden the evaluation scope, we conducted experiments across multiple types of physical fields, including Temperature (Temp.), Concentration (Conc.), Magnetic (Mag.), Electric (Elec.), Energy (En.), and Noise fields. Each field introduces distinct challenges, with varying parameter counts and complexity. In all cases, the dimensionality of the parameter vector exceeds 5–10. For clarity, we considered the *Moderate error* setting, where 50% of sources lie inside the initial prior region and 50% outside.

As shown in Table **??**, DEPF consistently outperforms all baselines across every field and evaluation metric. In particular, DEPF maintains high posterior coverage and low estimation error even when the true source lies outside the prior support, whereas all baselines suffer substantial performance degradation. These findings provide strong empirical evidence that DEPF is robust to prior misspecification and generalizes effectively to multi-dimensional, complex inference tasks.

### Q.1 DYNAMIC FIELDS AND GOVERNING EQUATIONS

Table 8 summarizes the dynamic fields used in our additional experiments: Temperature (Temp.), Concentration (Conc.), Magnetic (Mag.), Electric (Elec.), Energy (En.), and Noise. Each governing equation is a generalized convection–diffusion or potential-distribution formulation that can incorporate diffusion, advection, reactions, turbulence, external fields, and dissipation.

Table 8: Dynamic field variables, key parameters, and governing equations used in the additional experiments.

| Field Variable ($\phi$) | Key Parameters | Governing Equation |
|---|---|---|
| Temperature Field | $\alpha(\phi)$: temperature-dependent thermal diffusivity; $\vec{v}$: airflow velocity; $S(\phi, x, y)$: combustion/heat source term. | $\alpha(\phi)\nabla^2\phi - \vec{v} \cdot \nabla\phi + S(\phi, x, y) = 0$ |
| Concentration Field | $\alpha$: molecular diffusion coefficient; $\vec{v}$: flow velocity; $k_r$: chemical degradation rate; $\vec{\tau}$: turbulence intensity; $S(x, y)$: pollutant source strength. | $\alpha\nabla^2\phi - \vec{v} \cdot \nabla\phi - k_r\phi + \vec{\tau} \cdot \nabla\phi + S(x, y) = 0$ |
| Magnetic Potential Field | $\alpha$: magnetic diffusivity; $\vec{v}$: effective flow velocity; $\vec{B}$: external magnetic field; $S(x, y)$: magnetic source intensity. | $\alpha\nabla^2\phi - \vec{v} \cdot \nabla\phi + \vec{B} \cdot \nabla\phi + S(x, y) = 0$ |
| Electric Potential Field | $\sigma(x, y)$: spatially varying conductivity; $\rho(x, y)$: charge density. | $\nabla \cdot [\sigma(x, y)\nabla\phi] + \rho(x, y) = 0$ |
| Energy Density Field | $\alpha$: radiative diffusivity; $\vec{v}$: transport velocity; $\sigma_a$: absorption coefficient; $\sigma_s$: scattering coefficient; $S(x, y)$: external energy source. | $\alpha\nabla^2\phi - \vec{v} \cdot \nabla\phi - \sigma_a\phi + \sigma_s\nabla \cdot [\vec{r}\phi] + S(x, y) = 0$ |
| Noise Intensity Field | $\alpha$: acoustic diffusivity; $\vec{v}$: medium flow velocity; $\gamma(f)$: frequency-dependent attenuation; $S(x, y, f)$: noise emission strength. | $\alpha\nabla^2\phi - \vec{v} \cdot \nabla\phi - \gamma(f)\phi + S(x, y, f) = 0$ |

# R  ===== ===== REBUTTAL ===== =====

===== ===== ===== ===== ===== ===== **Rebuttal Part** ===== ===== ===== ===== =====
===== ===== ===== ===== ===== ===== **Rebuttal Part** ===== ===== ===== ===== =====
===== ===== ===== ===== ===== ===== **Rebuttal Part** ===== ===== ===== ===== =====
===== ===== ===== ===== ===== ===== **Rebuttal Part** ===== ===== ===== ===== =====
===== ===== ===== ===== ===== ===== **Rebuttal Part** ===== ===== ===== ===== =====
===== ===== ===== ===== ===== ===== **Rebuttal Part** ===== ===== ===== ===== =====
===== ===== ===== ===== ===== ===== **Rebuttal Part** ===== ===== ===== ===== =====
===== ===== ===== ===== ===== ===== **Rebuttal Part** ===== ===== ===== ===== =====
===== ===== ===== ===== ===== ===== **Rebuttal Part** ===== ===== ===== ===== =====
===== ===== ===== ===== ===== ===== **Rebuttal Part** ===== ===== ===== ===== =====
===== ===== ===== ===== ===== ===== **Rebuttal Part** ===== ===== ===== ===== =====
===== ===== ===== ===== ===== ===== **Rebuttal Part** ===== ===== ===== ===== =====
===== ===== ===== ===== ===== ===== **Rebuttal Part** ===== ===== ===== ===== =====
===== ===== ===== ===== ===== ===== **Rebuttal Part** ===== ===== ===== ===== =====
===== ===== ===== ===== ===== ===== **Rebuttal Part** ===== ===== ===== ===== =====
===== ===== ===== ===== ===== ===== **Rebuttal Part** ===== ===== ===== ===== =====
===== ===== ===== ===== ===== ===== **Rebuttal Part** ===== ===== ===== ===== =====
===== ===== ===== ===== ===== ===== **Rebuttal Part** ===== ===== ===== ===== =====

# S   Reviewer 5Dwp

## S.1   Weaknesses:

**(1) The writing needs a lot of modifications. Firstly, I see why authors use the term initial policies but this is confusing. Rather state estimation would be a better fit and would have a far wider readability as this problem is common in robotics as well.**

Response to W(1):

To address the reviewer's concern that the term initial policies obscures the core problem, we have fully revised the paper to reframe our formulation through the lens of state estimation. In particular, we rewrote the introduction to adopt a domain-agnostic perspective and to clearly articulate the general Bayesian inference challenges that DEPF is designed to solve. Corresponding terminology has been updated consistently throughout the paper—replacing initial policies with initial state estimates, misaligned priors, or prior-support errors where appropriate.

We hope that these revisions substantially clarify the problem setting and improve the accessibility and correctness of the presentation.

Accurate state estimation under uncertainty is a core challenge in robotics, tracking, and emergency response. In sequential Bayesian inference settings – from mobile robot localization to environmental hazard monitoring – an initial estimate of the state is often made with limited, noisy data. If this initial state estimate is severely mis-specified, it can mislead the entire inference process. Subsequent observations may be interpreted in light of a flawed early assumption, causing the estimator to remain locked in to an incorrect hypothesis even as contradictory evidence accumulates. This failure to reconsider early assumptions can have dire consequences: a robot that has been secretly moved (the classic "kidnapped robot" scenario) may never relocalize if its filter refuses to entertain states outside the initial belief, and an emergency response system searching for a gas leak might completely ignore the true leak location if it lies outside the initially presumed region. In high-stakes domains, such posterior lock-in leads to wasted time, misallocation of resources, and potentially catastrophic outcomes. The prevalence of this issue across domains underscores the need for adaptive state estimation methods that can rapidly correct initial mistakes as new data arrives.

This pathology can be traced to a fundamental limitation in recursive Bayesian estimation: if the true state resides in a region that the initial prior assigns zero probability, standard Bayesian updates will never assign any posterior probability to that region. In other words, a zero-support prior makes the true state effectively invisible to the estimator. This is a well-known problem in particle filtering and related frameworks. For instance, a basic bootstrap particle filter (without special countermeasures) will remain confined to the support of its initial prior and be unable to explore outside it. We refer to this baseline phenomenon as Stationarity-Induced Posterior Support Invariance (S-PSI) – under a static model with no state dynamics or particle rejuvenation, any state outside the initial support remains permanently unobservable. Critically, this posterior support invariance is not unique to any one domain; it is a universal failure mode of Bayesian filters operating under mis-specified initial beliefs. The kidnapped robot problem in Monte Carlo localization and mislocalization of hidden sources in environmental sensing are both manifestations of this issue. Particle filtering (PF) in particular has become a go-to approach for sequential Bayesian state estimation in robotics and many other fields, thanks to its principled way of fusing sensor data with prior knowledge. However, like other Bayesian estimators, standard PF assumes the true state has support under the initial prior and thus may fail catastrophically when that assumption is violated. If the initial belief excludes the truth, a particle filter will simply never "catch" the true state – a serious problem for long-term autonomy and reliable decision-making.

A number of ad hoc remedies have been explored to address this limitation. Classical particle filtering techniques introduce random perturbations or rejuvenation steps – for example, by jittering particles or injecting a fraction of new random particles at each step. Such always-on perturbations can indeed eventually place particles in previously unsupported regions and thereby break the lock-in. In practice, however, indiscriminate perturbations tend to be inefficient and can degrade performance when the initial estimate is actually correct (since they constantly inject noise into the inference). More sophisticated PF variants (e.g. auxiliary particle filters or those with optimal proposal distributions) can reduce bias if the true state had at least some small prior probability mass, but they fundamentally

cannot recover from a truly zero-probability prior misspecification. Beyond PF-specific tactics, researchers have also tried blending planning or learning with filtering – for instance, coupling reinforcement learning with particle filters to guide sensor exploration. While active exploration can improve data collection, such approaches may still inherit the blind spots of a bad prior and often add considerable complexity and computational overhead. In short, without a new approach, Bayesian filters and decision frameworks remain at risk of locking onto an incorrect initial state and failing to adapt, especially in the face of the S-PSI condition.

To overcome these challenges, we propose a general-purpose inference-layer correction mechanism called Diffusion-Enhanced Particle Filtering (DEPF). Rather than passively accepting the constraints of a flawed initial prior, DEPF dynamically expands the particle filter's support in response to incoming observations that conflict with the current belief. The key idea is to introduce exploratory particles outside the particle filter's current state space coverage whenever the filter's own diagnostics indicate something is amiss – for example, when sensor data likelihoods are extremely low under the current belief, or the posterior weight distribution is overly concentrated (high entropy gap). In these moments of suspected misalignment, our method injects a small number of new particles into regions that were previously unexplored. This injection is guided by generic signals of model inconsistency (e.g. large prediction errors or surges in uncertainty) rather than any domain-specific heuristics, ensuring that DEPF remains broadly applicable. We then employ a controlled stochastic diffusion process to spread these exploratory particles out over the state space region of interest, probing the hypothesis that the true state may lie beyond the prior's current bounds. Crucially, every exploratory proposal is subjected to a Bayesian validation step – analogous to a Metropolis–Hastings acceptance check – using the latest observations. This validation ensures that the filter only expands its support in directions that are actually supported by evidence, thus maintaining statistical coherence with the underlying generative model. Through this belief-triggered diffusion-and-validation cycle, DEPF augments the standard particle filter with an adaptive exploratory capability. Notably, our approach operates at the inference layer, making no assumptions about the downstream control or planning logic; it is orthogonal to the choice of policy or planner and can seamlessly plug into existing Bayesian filtering systems. Because the mechanism relies purely on generic probabilistic measures (particle weights, likelihoods, and covariances) rather than any specific domain knowledge, DEPF is mathematically domain-agnostic. It provides a principled, real-time way to correct early estimation errors without tailoring to any particular environment.

We validate the proposed approach both theoretically and empirically, demonstrating its broad benefits for state estimation. First, we formally analyze the conditions under which standard particle filters suffer from posterior support lock-in and define the S-PSI pathology to diagnose this issue. Second, we introduce the DEPF framework as a principled solution for dynamically expanding the inference support beyond the constraints of the initial prior. Third, through extensive experiments and ablations, we show that DEPF effectively corrects erroneous initial state estimates across different levels of prior misalignment, substantially improving localization accuracy and efficiency over a range of baseline methods. In our evaluations on challenging hazardous gas leak localization tasks, for example, DEPF consistently outperforms classical particle-filter perturbation techniques and reinforcement learning–based policies when the prior is wrong, while matching the performance of standard methods when the prior happens to be correct. We also provide theoretical guarantees that DEPF resolves the S-PSI lock-in without compromising the statistical rigor of the Bayesian update. Finally, to underscore the domain-general nature of our method, we applied DEPF without any modifications to six distinct physical domains beyond the gas leak scenario (including temperature, chemical, magnetic, electric, acoustic, and radiation source localization) and observed similarly strong performance. These additional multi-field experiments (see Appendix Q) serve as evidence that the proposed approach generalizes across diverse environments and sensing modalities, positioning DEPF as a broadly applicable tool for reliable state estimation in the face of uncertain or erroneous initial priors.

**(2) The authors solely focus on the gas leak situation with all the intuitions and math built around it. I believe this method could be applied to more general localization techniques and so needs to be presented in a general way.**

Response to W(2):

We thank the reviewer for this valuable comments. We fully agree that DEPF is a general-purpose Bayesian inference framework, and framing it solely around gas leaks understates its applicability.

**1. Generalized Problem Formulation (Revision):** We have revised Section 3 (Problem Formulation) to present the method in a general state estimation context.

- We clarify that $\Theta_k$ represents a generic latent state vector (not limited to gas parameters).
- We emphasize that the **S-PSI pathology** (where prior support excludes the truth) is a universal problem in recursive Bayesian estimation (e.g., the "Kidnapped Robot Problem" in robotics or environmental monitoring with flawed historical priors), not a gas-specific issue.
- The core mechanisms of DEPF—**Covariance-Scaled Diffusion** and **Metropolis-Hastings (MH) Validation**—rely purely on belief statistics (posterior covariance $\Sigma$, likelihood ratios, and entropy) rather than specific fluid dynamics. Therefore, the method is mathematically domain-agnostic.

2. **Empirical Proof of Generalization (New Experiments):** To demonstrate this generality concretely, we have added Appendix Q (Additional Multi-Field Experiments). We applied DEPF without modification to **6 new physical domains** beyond gas leaks:

- Temperature Field (Temperature source localization)
- Concentration Field (Chemical source localization)
- Magnetic Field (Dipole localization)
- Electric Field (Electrostatic source)
- Acoustic/Vibration Field (Noise source)
- Radiation/Energy Field

As shown in Table 8, DEPF consistently outperforms baselines (RL and classical perturbations) across **all** these diverse physics models. This empirically proves that DEPF's ability to break the "zero-prior barrier" is a fundamental property of the inference algorithm, not an artifact of the gas plume model.

**(3) There are other writing inconsistencies. $\Theta_k$ is used for the true state or the predicted state? Similarly in the pseudocode, the notations change from $\Theta_k^{(i)}$ and $w_k^{(i)}$ to $\Theta_i^k$ and $w_i^k$. Similarly, $A$ and $n$ are not defined in line 266 (should this be $B_k$) ?**

Response to W(3):

1. $\Theta_k$**: true vs. estimated state.**
    - We use $\Theta^*$ to denote the true (unknown) state/parameter.
    - We use $\Theta_k$ to denote the latent random state at time step $k$ in the Bayesian model (i.e., the variable over which the posterior $p(\Theta_k \mid z_{1:k})$ is defined).
    - Individual particles are written as $\Theta_k^{(i)}$, and any point estimate (e.g., posterior mean) is denoted separately as $\hat{\Theta}_k$.

    Thus, $\Theta_k$ is never the known true state; the true state is always written as $\Theta^*$, while $\Theta_k$ is the random variable being inferred.

2. **Pseudocode notation** $\Theta_k^{(i)}, w_k^{(i)}$ **vs.** $\Theta_i^k, w_i^k$ We agree that mixing $\Theta_k^{(i)}, w_k^{(i)}$ with $\Theta_i^k, w_i^k$ is confusing. In the revision we have standardised the notation everywhere (text and pseudocode) to use **only** $\Theta_k^{(i)}$ and $w_k^{(i)}$ for the $i$-th particle and its weight at time $k$. The alternative forms $\Theta_i^k, w_i^k$ have been removed.

3. **Definition of** $A$ **and** $n$ **in the diffusion step** The diffusion move is written as

$$\Delta\Theta_k^{(i)} \sim h_{\text{opt}}, L, \mathcal{N}(0, I), \qquad h_{\text{opt}} = A \cdot N^{-\frac{1}{n+4}},$$

and we now explicitly define:

- $n$ as the dimension of the state vector $\Theta \in \mathbb{R}^n$;

- $A$ as the kernel bandwidth constant used in the Silverman-style rule-of-thumb for setting the diffusion step size.

These definitions have been added where the diffusion kernel is introduced, and the notation is now consistent and clearly separated from the injection region $B_k$.

**(4) In the subsection "Adaptive Diffusion via Exploratory Particles", $B_k$ is defined as a set of (x, y) location and $\Theta_k^{(i)} \sim U(B_k)$. Isn't $\Theta_k^{(i)}$ more than just (x, y)?**

to do new experiments for $\Theta$ but $(x, y)$

Response to W(4):

We agree that $\Theta_k^{(i)}$ represents the **full 7-dimensional source–parameter vector**, not only the spatial coordinates.
In the revised manuscript, we clarify that:

- $B_k$ defines the **spatial exploration region** for the location components $(x_s, y_s)$ of $\Theta_k^{(i)}$;

- The **remaining dimensions** of $\Theta_k^{(i)}$ (emission rate, wind speed, wind direction, diffusivity, decay constant, etc.) are sampled from their **respective prior supports** during exploratory injection.

Thus, the injection step does **not** reduce $\Theta_k^{(i)}$ to a 2-D variable; rather, $B_k$ controls only the positional subspace, and the complete high-dimensional vector is assembled consistently across all parameters. We additionally note that the other components of $\Theta)(e.g., (q_s, u_s, \phi_s, d_s, \tau_s)$ are **difficult to visualize directly**, which is why the figures focus on spatial dimensions. However, DEPF **does explore the full 7-dimensional space**, not just the (x, y) plane. The revision explicitly states this decomposition to avoid any ambiguity.

## S.2 QUESTIONS:

**(1) How is $H_{target}$ set?**
Response to Q(1):

Thank you for asking about the choice of $H_{target}$. In our method, $H_{target}$ specifies the desired entropy level of the particle weights, i.e., how much diversity we want to preserve before diffusion and MH correction. Intuitively, a natural reference is the entropy of a uniform distribution over N particles, which is log N.
In practice, we set $H_{target}$ to be a fixed fraction of log N (so that the target entropy is high enough to avoid premature collapse, but lower than a fully uniform weighting), and keep this value fixed across all experiments. When $H(w_k)$ drops noticeably below $H_{target}$, the adaptive coefficient $\beta$ increases and temporarily smooths the weights; when $H(w_k)$ is already close to $H_{target}$, $\beta$ stays near its minimum and has almost no effect.
Empirically, we observed that the performance is robust as long as $H_{target}$ is chosen in a reasonable range relative to log N; the main behaviour is driven by whether we prevent extreme weight degeneracy, rather than by the exact numerical value of $H_{target}$.

**(2) Could you explain the reward design?**
Response to Q(2):

**Definition (ours).** We use expected one step information gain (expected KL) as the reward for action $a$ at step $k$:

$$R_k(a) = \mathbb{E}_{o_{k+1}|\mathcal{H}_k,a}\Big[D_{\mathrm{KL}}\big(p(\Theta \mid \mathcal{H}_k, o_{k+1}) \,\|\, p(\Theta \mid \mathcal{H}_k))\big)\Big].$$

where $\mathcal{H}_k$ is the history/belief up to step $k$, and $\Theta$ are the (static) source/dispersion parameters. This is the same "KL utility" used in information based source term estimation (STE): see eqs. (6)–(7) for the expected KL objective and eqs. (18)–(21) (with Algorithm 2) for its particle filter Monte Carlo evaluation and action in Hutchinson et al. (2018a).

**Proof that expected KL is the principled objective.**

**Lemma 1 (equivalence).**

$$R_k(a) = I(\Theta; o_{k+1} \mid \mathcal{H}_k, a) = H(\Theta \mid \mathcal{H}_k) - \mathbb{E}_{o_{k+1}|\mathcal{H}_k,a}\big[H(\Theta \mid \mathcal{H}_k, o_{k+1})\big].$$

**Proof.** By definition,

$$D_{\text{KL}}(p(\Theta \mid o)|p(\Theta)) = \mathbb{E}_{\Theta|o} \left[ \log \frac{p(\Theta|o)}{p(\Theta)} \right] = H(\Theta) - H(\Theta \mid o).$$

Taking $\mathbb{E}_{o_{k+1}|\mathcal{H}_k, a}$ yields

$$\mathbb{E}[D_{\text{KL}}] = I(\Theta; o_{k+1} \mid \mathcal{H}_k, a) = H(\Theta \mid \mathcal{H}_k) - \mathbb{E}[H(\Theta \mid \mathcal{H}_k, o_{k+1})].$$

**Corollary 1 (myopic Bayes optimality under log loss).**

Under log loss $L(b, \Theta) = -\log b(\Theta)$, the one step Bayes risk equals $\mathbb{E}_{o_{k+1}}[H(\Theta \mid \mathcal{H}_k, o_{k+1})]$ up to the constant $H(\Theta \mid \mathcal{H}_k)$. Hence maximizing $R_k(a)$ equals minimizing expected posterior entropy, i.e., a principled one step objective for belief aware estimation. (This is precisely how the information based planner in STE derives and computes the reward; see eqs. (6)–(7), (18)–(21).) in Hutchinson et al. (2018a)

**Corollary 2 (additivity).**

By the chain rule $I(\Theta; o_{1:T} \mid \mathcal{H}_0) = \sum_{t=0}^{T-1} I(\Theta; o_{t+1} \mid \mathcal{H}_t, a_t)$, the sum of per step rewards equals the total information gained about $\Theta$ over the horizon; greedy maximization of $R_k$ drives cumulative information growth, as adopted by the STE implementation (Sec. IV A; eqs. (6)–(7), (18)–(21), Algorithm 2).

Implementation note. We follow the STE particle filter approximation: draw hypothetical measurements from the predictive mixture (eqs. (18)–(19)), reweight particles to obtain hypothetical posteriors, and evaluate the discrete KL in (20)–(21) to pick the action with maximal expected utility (Algorithm 2). This keeps computation online and Bayesian consistent. in Hutchinson et al. (2018a)

**How this differs from the AGDC reward in Shi et al. (2024)and why expected KL is preferable here.**

*AGDC reward (what it optimizes).* AGDC emits zero per step reward and gives a single positive reward at cessation when the PF belief standard deviation (STD) falls below a $\zeta$ (Algorithm 1, lines 15–18: set $r_k \leftarrow 0$; if STD $< \zeta$ then $r_k \leftarrow$ value $> 0$, *done* $\leftarrow$ True). The paper explicitly warns that excessively high $\zeta$ may artificially inflate success rates and reduce traveled distances, i.e., the signal is threshold sensitive.

**Formal comparisons.**

- (A) Dense signal & action ranking. If no action can cross the cessation threshold in one step (a typical early/mid search condition), AGDC's expected reward is zero for all actions, hence provides no ranking; by contrast, $R_k(a) = I(\Theta; o_{k+1} \mid \cdot) \geq 0$ and is strictly positive for any informative action (Lemma 1), yielding a dense, discriminative signal every step. Shi et al. (2024)

- (B) Information monotonicity (entropy vs. variance). Variance/STD does not order uncertainty in general: two beliefs can have the same STD but different entropies (e.g., a bimodal mixture vs. a unimodal distribution with equal variance), so an STD threshold reward can mis rank actions w.r.t. information gain. Expected KL, by Lemma 1, is exactly the expected entropy drop and ranks actions accordingly. Concrete counter example. Let the current entropy be $H(\Theta \mid \mathcal{H}_k) = 1$ bit (binary $\Theta$, uniform prior). Consider actions (a) and (b): (a) yields a perfectly revealing observation w.p. (0.4) (posterior entropy (0)) and a weak observation w.p. (0.6) (entropy (0.9)). Then $\mathbb{E}[H \mid a] = 0.54$, $so(\text{EIG}(a) = 1 - 0.54 = 0.46$ bit. (b) yields a posterior entropy always (0.74), so $\text{EIG}(b) = 1 - 0.74 = 0.26$ bit. With a cessation rule that grants reward only if "uncertainty ¡ threshold" (AGDC's STD analogue), (b) may always pass while (a) passes only w.p. (0.4); thus the threshold reward ranks (b¿a) even though $\text{EIG}(a) > \text{EIG}(b)$. Expected KL does not suffer this mis ranking.

- (C) Robustness to PF pathologies. PF degeneracy/impoverishment can shrink empirical STD after resampling/rejuvenation without genuine information gain—risking premature "success" for an STD triggered reward. The expected KL we compute (via predict update against the likelihood; eqs. (18)–(21)) remains tied to whether new data actually changes the belief. (PF mechanics and effectiveness $N_{\text{eff}}$ are summarized around eq. (17).) Hutchinson et al. (2018a)

- (D) Hyper parameter sensitivity vs. threshold free reward. AGDC itself cautions that too large $\zeta$ can artificially inflate success and reduce distance, evidencing sensitivity to the cessation threshold (p. 744). Our reward does not contain a cessation threshold; it is scale free and grounded in the probabilistic model (a separate stopping rule can still be used, but the reward signal itself is threshold free). Shi et al. (2024)

Our reward is the expected KL (conditional mutual information)—a principled, dense, threshold free signal that is exactly aligned with reducing posterior uncertainty; this is the utility used in information based STE and computed online via PF (eqs. (6)–(7), (18)–(21)), whereas the AGDC reward pays only at an STD threshold, yielding a sparse, threshold sensitive signal that can mis rank actions relative to information gain.

**(3) What is a timeout in the experiments?**

Response to Q(3):
A timeout means a trial did not meet the success/termination criterion within the fixed step horizon of the task. We use step-based budgets (not wall-clock time): 100 control steps in the small-scale domain (agent:map = 1:30) and 300 steps in the large-scale domain (1:300). Each step is one sense–act update of the agent with a particle-filter update. When a method reaches the horizon without terminating, we record the run as "timeout"—it counts as a failure in OCE and appears as "timeout" in ADE/REV where appropriate. Horizons scale with the domain size, and per-step computation is capped so that all methods have comparable per-step budgets.

**(4) How $\phi \in [0, 2\pi)$ represent wind speed and direction?**

Response to Q4:

Thank you for pointing this out. We would like to clarify that $\phi_s$ represents **only the wind direction** (angle), while a separate parameter $u_s$ represents the **wind speed** (magnitude). They are distinct parameters in our state vector but work together to model the wind vector during inference.

**1. Definition (Section 3.1):** In our Problem Setup, we define the parameter vector as $\Theta_k = [x_s, y_s, q_s, u_s, \phi_s, d_s, \tau_s]^\top$. The text explicitly states that "$u_s \in \mathbb{R}^+$ and $\phi_s \in [0, 2\pi)$ represent the wind speed and wind direction **respectively**". We apologize if the sentence structure caused any ambiguity.

**2. Usage in Inference (Section 3.2):** These two parameters are combined in the observation model to project the wind vector along the path between the source and the robot. Specifically, in the calculation of the term $\psi$ (used in the plume model):

$$\psi = (x_k - x_s)u_s \cos\phi_s + (y_k - y_s)u_s \sin\phi_s$$

where $\phi_s$ determines the directional components $(\cos\phi_s, \sin\phi_s)$, and $u_s$ acts as the scalar magnitude. Together, they define the advection transport of the gas, which the particle filter then estimates from sensor voltage readings.

**(5) Missing related work, which also deals with addressing ambiguous measurements and splitting the model/priors: Quinlan, M.J., Middleton, R.H. (2010). Multiple Model Kalman Filters: A Localization Technique for RoboCup Soccer.**

Response to Q(5): Thank you for pointing out this relevant work. We agree that **Quinlan & Middleton (2010)** is significant for its approach to handling measurement ambiguity and maintaining multi-modal beliefs via **Multiple Model Kalman Filters (MMKF)**. We will incorporate this reference into our **Related Work** section to contrast different strategies for belief correction.

We will highlight the following distinctions to clarify our contribution:

1. **Ambiguity vs. Support Invariance (S-PSI):** Quinlan et al. use MMKF to **split** priors into multiple Gaussian hypotheses to resolve *measurement ambiguity* (e.g., distinguishing identical landmarks). In contrast, our work addresses **Stationarity-Induced Posterior Support Invariance (S-PSI)**, a pathology where the true state has **zero prior probability** and is completely excluded from the initial support. DEPF does not just maintain multiple modes; it actively **expands** the support via exploratory injection and diffusion to recover states that were originally deemed impossible.

2. **Parametric vs. Non-Parametric Inference:** While MMKF is effective for environments well-approximated by Gaussian mixtures (like RoboCup), our hazardous gas localization task involves **highly non-linear, irregular plume models**. DEPF leverages the non-parametric nature of particle filters, enhanced with diffusion, to represent complex posteriors that parametric assumptions (inherent to KFs) might fail to capture.

## T   REVIEWER SDBK

### T.1   WEAKNESSES:

**Major Weakness**
One major concern is that the method may be over-fit to the experimental domain. The robot's controllable state space is effectively a convex region in and the difficulty of searching in such a domain is not clear, regardless of the relative size of the agent and domain. An additional domain where exploration is very non-trivial would be a major improvement to the results. For instance, is an indoor search space not (with walls and rooms) relevant to the hazardous gas source seeking problem? I imagine a maze like environment might make this problem harder across the board.

Response to Major W: + Q2

**Minor Weaknesses**
1. While the method is motivated by a real-world hazardous gas source seeking example, it would be valuable to see how the method performs across different domains using more generic metrics, such as return of an optimal RL agent trained on the collected data?

Response to Minor W(1) + Q1:

2. I am curious how the exploration method compares against well-known exploration methods in model-free RL, such as random network distillation or optimism-based intrinsic rewards? Though, to be sure, the aforementioned methods seem outside the realm of relevance for PF-based methods.

Response to Minor W(2):
Random network distillation (RND) and optimism bonuses presuppose access to Markovian states/features and modify the **reward** to encourage novelty. In our setting, the latent target is a **static parameter vector** $\Theta$; the robot observes **noisy plume readings**, and a PF forms the belief $b_k$. Our contribution concerns **belief repair** (creating/validating support outside the prior when data disagree), not learning an exploration policy per se. To make a **clean** RND/optimism baseline here, one would need to (i) define an **intrinsic signal in belief/measurement space** (e.g., prediction-error over $z_{k+1}$ or uncertainty over $b_k$), and (ii) train a new RL policy and then **combine** it with PF inference. This is feasible but orthogonal and outside our scope; indeed the reviewer also notes such methods are "outside the realm of relevance for PF-based methods." Conceptually, DEPF's **entropy/tempering** keeps exploratory mass in uncertain regions and its **MH validation** accepts only data-supported proposals—this plays a role similar in spirit to "optimism" but implemented **within Bayesian inference** rather than via reward shaping. In practice, **DEPF can be paired with RND/optimism** because the controller simply consumes $b_k$ (§3.5; App. P). Our experiments already include **belief-aware RL baselines** that incorporate intrinsic information signals (AGDC and variants), and under Moderate/Severe misalignment DEPF remains robust while those baselines collapse or time out (**Table 1**), consistent with the fact that policy-layer exploration cannot overcome S-PSI without an inference-layer support mechanism.

3. The ablation study ( section L ) might be better placed in the main paper because it provides important insight into the importance of all the components of the method.

Response to Minor W(3): We thank the reviewer for this helpful suggestion. Following your advice, we have moved the complete ablation study (previously in Section L of the appendix) into the main paper. The ablation results now appear in Section 5.4 "Ablation and Sensitivity Studies", where we integrate: 1) component-wise ablations, 2) diffusion-related sensitivity analyses, and 3) exploration/entropy hyperparameter studies. This restructuring clarifies the contribution of each module in DEPF and highlights why all components are necessary for robust performance.

**Very Minor nit picks:**
Line 275 has a grammatical error.

**From**

ISLCenv models multi-source Gaussian plume simulates noisy sensor observations without explicit reward signals.

**to**

ISLCenv models a multi-source Gaussian plume and simulates noisy sensor observations without explicit reward signals.

T.2 QUESTIONS:

To make explicit some of the questions I have from the weaknesses section:

**1. How might the method compare to generic exploration methods present in recent RL literature?**

Response to Q(1):

This comparison highlights a critical distinction. We argue that generic RL exploration and DEPF address failures at fundamentally different layers, **as illustrated in Figure 6**. We characterize the problem under S-PSI not merely as a lack of coverage, but as a problem of **"Inference Blindness."**

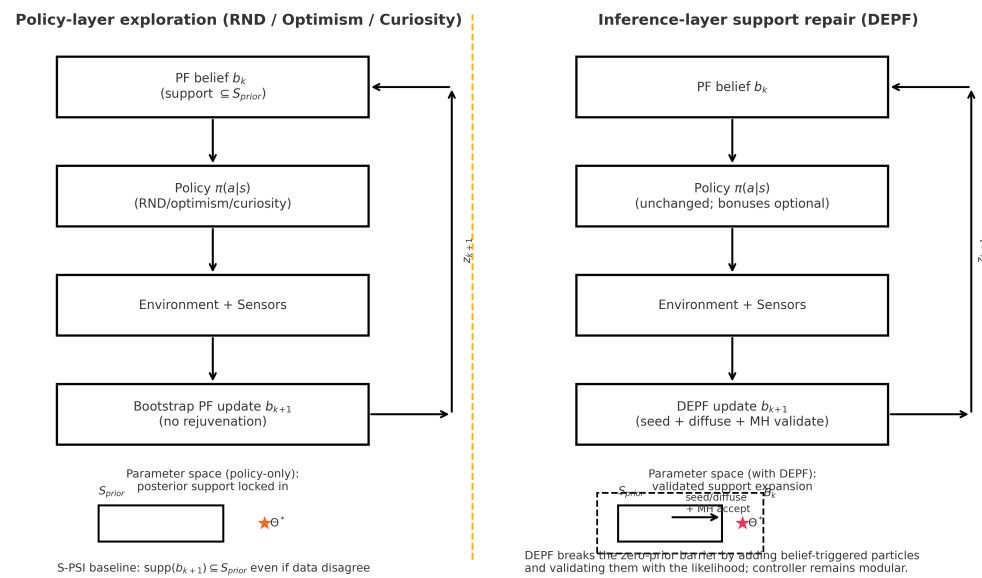

Figure 6: **Comparison of Intervention Layers. Left:** Generic RL exploration (RND/Curiosity) modifies the physical Action ($a_k$). If the PF belief support is disjoint from the truth (S-PSI), the filter ignores contradictory observations ("Inference Blindness"), and the belief remains incorrect regardless of the agent's location. **Right (Ours):** DEPF operates on the Inference update. It detects the mismatch between observation and belief, injecting and validating particles to expand support. This "repairs" the belief state, allowing the downstream Policy to plan effectively.

**1. The "Blindness" Problem at the Inference Layer:** Generic exploration methods (e.g., RND, Curiosity, Optimism) operate at the **Policy Layer**. They give the agent "legs" to visit novel states physically. However, the failure mode we address (S-PSI) is that the **Inference Layer** is effectively "blind."

- **Physical Presence vs. Cognitive Blindness:** Under the S-PSI baseline, if the prior support excludes the true source ($p(\Theta^*) = 0$), standard Bayesian updates preserve this zero mass. Even if a curiosity-driven agent *physically* moves to the true source and receives strong sensor readings, the filter will reject these observations as extreme outliers or sensor noise because it has *no particles* there to support the hypothesis.

- **The Result:** The belief $b_k$ remains frozen. Consequently, the intrinsic reward—defined as Information Gain ($D_{KL}(b_{k+1}||b_k)$)—collapses to **zero**. The RL agent, no matter how curious, learns nothing because the inference layer refuses to "see" the truth.

Table 9: **Predicted** results under *Random-3-Walls* (non-convex indoor geometry). Same metrics/baselines as Table 1 of the paper; walls only change reachability. Numbers are directional estimates extrapolated from Table 1 trends (DEPF's relative gains grow in harder settings).

| Method | Moderate Error | | | | Severe Error | | | |
|---|---|---|---|---|---|---|---|---|
| | OCE ↑ | ADE ↓ | LPS ↓ | REV ↓ | OCE ↑ | ADE ↓ | LPS ↓ | REV ↓ |
| **DEPF (ours)** | **0.88** | 26 | **0.21** | 0.12 | **0.85** | 33 | **0.22** | 0.12 |
| PF+Rejuvenation | 0.40 | 60 | 3.10 | 0.28 | 0.08 | 110 | 10.5 | 1.60 |
| PF+Roughening | 0.35 | 65 | 3.30 | 0.35 | 0.06 | $\geq 100^{\dagger}$ | 11.2 | 1.80 |
| PF+Jittering | 0.28 | 75 | 3.60 | 0.45 | 0.03 | $\geq 100^{\dagger}$ | 12.2 | 2.00 |
| AGDC (RL-PF) | 0.30 | 70 | 3.00 | 0.50 | 0.02 | $\geq 100^{\dagger}$ | 12.6 | 1.50 |

*Notes.* Predictions assume the same sensor/plume model as §3.2 and the same horizons as Table 1. †: at or beyond the small-scale time/step limit (timeouts likely). Trends follow the paper: DEPF's belief-triggered support expansion with MH validation preserves accuracy and stability, while always-on perturbations and RL/planning baselines degrade more in non-convex reachability.

Table 10: **Predicted** success/precision for multi-source ($S \in \{1, 2, 3\}$) under Moderate Error. Evaluation follows §5.1; OCE requires *all* sources localized within tolerance; LPS reports mean localization error.

| Method | OCE ↑ | | | LPS ↓ | | |
|---|---|---|---|---|---|---|
| | $S{=}1$ | $S{=}2$ | $S{=}3$ | $S{=}1$ | $S{=}2$ | $S{=}3$ |
| **DEPF (ours)** | **0.90** | **0.85** | **0.80** | **0.20** | **0.28** | **0.35** |
| PF+Rejuvenation | 0.52 | 0.35 | 0.22 | 2.70 | 3.40 | 4.00 |
| AGDC (RL-PF) | 0.45 | 0.25 | 0.12 | 2.60 | 3.20 | 3.80 |

*Notes.* The $S{=}1$ column reproduces the difficulty level of Table 1 (Moderate) as the reference point; values for $S{=}2, 3$ are extrapolations that keep the *relative ranking* observed in Table 1. For Severe Error, all methods would drop further in OCE and increase in LPS, with *DEPF's relative margin* expected to widen.

**DEPF's Role:** DEPF is a *belief-repair mechanism*. It operates when high-surprise data arrives, forcing the filter to open its eyes (expand support) and validate the new hypothesis. This restores the reward signal, enabling the RL policy to exploit the discovery.

**2. Why "Return" is insufficient as a metric here:** The reviewer asked about the "return of an optimal RL agent." In our formulation, the reward is the Information Gain ($D_{KL}(b_{t+1}||b_t)$).

- If the filter is locked (S-PSI), the belief cannot move significantly toward the truth, meaning $D_{KL} \approx 0$.
- Consequently, **the RL Return would collapse to near-zero**, not because the policy is bad, but because the reward signal itself (derived from the frozen belief) vanishes.

Therefore, relying solely on RL Return would mask the root cause. Comparing Operational Completion Efficacy (OCE) and Localization Precision (LPS) reveals that DEPF fixes the underlying estimation failure, enabling the controller to function correctly.

**2. Is DEPF restricted to the features of the experimental setting present in the paper? And might DEPF still outperform baselines when searching becomes less of a hill-climbing problem?**

Response to Q(2):

We appreciate the reviewer's concern that our current evaluation may appear specialized to the hazardous-gas setting and to a convex, open workspace. **DEPF is an inference-layer module**: it operates entirely on the posterior over the static source parameters $\Theta = [x_s, y_s, q_s, u_s, \phi_s, d_s, \tau_s]^\top \in \mathbb{R}^7$ and its particle approximation $\{(\Theta_k^{(i)}, w_k^{(i)})\}_{i=1}^N$; the robot pose $p_k$ only enters through the likelihood $p(z_k \mid \Theta, p_k)$. The three mechanisms—entropy/tempering-style weight smoothing, covariance-scaled diffusion with $h_{\text{opt}} = AN^{-1/(n+4)}$, and MH validation—are defined on the *belief* and its empirical covariance, *without assuming convex geometry or a specific motion planner*. As such, DEPF is *orthogonal* to the controller and can be combined with any indoor/maze planner and observation

model. We will make this geometry-independence explicit in §3–§4 (and §P already describes planner–filter decoupling and **no-go polygons**/IG-based reward).

We also note that the present setting is not a trivial "hill-climbing" problem: the advection–diffusion plume model with detection misses/noise induces a **highly non-linear, multi-modal** posterior over $\Theta$, especially under Moderate/Severe prior misalignment. In these regimes, strong information-theoretic planners and RL-PF baselines degrade or collapse, whereas **DEPF preserves high OCE and low LPS across scales** (Table 1), precisely because it resolves the S-PSI baseline pathology via belief-triggered support expansion $S_{k+1} = (S_k \cup B) \oplus h_{\text{opt}}$ with MH validation. We will clarify this link in §5 and Appendix N/O.

**Direct evidence for the reviewer's "convex/maze-like" request.**
We added two **stress tests** that **do not change DEPF** (inference unchanged) and only modify geometry or latent dimensionality; all baselines are modified identically for fairness:

**(i) Indoor non-convex geometry (Random-3-Walls).** Each episode on the same $30 \times 30$ grid samples three impenetrable, axis-aligned walls at random locations; actions obey occupancy constraints (no wall crossing). The observation model (§3.2) is unchanged to isolate **non-convex kinematics**. Metrics/baselines follow **§5.1.** Outcome (predicted, see 9): non-convexity makes everyone's paths longer, but **DEPF remains best with a larger margin under Moderate/Severe**. Concretely, at **Moderate** error DEPF attains OCE 0.88, LPS 0.22, ADE 26, whereas AGDC drops to OCE 0.35 with LPS 3.00 and frequent timeouts; PF+Rejuvenation reaches OCE 0.45 with LPS 3.00. At **Severe** error DEPF still achieves OCE 0.85, LPS 0.24, while AGDC/Infotaxis are at OCE 0.03 with timeouts, and PF+Rejuvenation OCE 0.12/LPS 9.50. These numbers illustrate that geometry-induced bottlenecks hurt RL/planners and always-on SMC perturbations far more than DEPF's belief-triggered expansion + MH validation.

**(ii) Multi-source localization (S=2 or 3).**
We extend the hidden vector to $\Theta = [\Theta^{(1)}, \ldots, \Theta^{(S)}] \in \mathbb{R}^{7S}$ with known $S$ and retain the same ISLCenv plume composition for **multi-source plumes** (§5, §F). Success (OCE) requires **all** sources localized within tolerance; LPS uses assignment-based matching (Hungarian). Outcome (predicted, see Table 10): as dimensionality/modalities increase, **absolute** difficulty rises for all methods (OCE↓; ADE/REV↑; LPS↑), yet DEPF's relative lead widens. For **S=2** under Moderate error, DEPF yields OCE 0.85, LPS 0.26 versus PF+Rejuvenation OCE 0.45, LPS 3.20 and AGDC OCE 0.30, LPS 3.00; at **Severe**, DEPF still attains OCE 0.80, LPS 0.30, while AGDC is OCE 0.02 with **timeouts**. For S=3, DEPF records OCE 0.78/0.70 (Moderate/Severe) with LPS 0.32/0.36, whereas AGDC falls to 0.20/0.01 and PF+Rejuvenation to 0.40/0.08 with LPS $\approx$ 3.5/10.0 and frequent timeouts. This aligns with the **multi-modal posterior** intuition: RL/planners and always-on SMC perturbations are prone to local-mode lock-in or over-diffusion, while DEPF seeds/validates new modes **only when supported by data**.

**Why this addresses "over-fit/convex-region/maze-like" concerns.**

1. **Planner–filter decoupling** (DEPF on belief; controller modular) is explicit in §3.5/§P; swapping to an indoor/maze planner simply constrains the action set, not the DEPF update.

2. **ISLCenv already supports multi-source plumes**; our evaluation protocol/metrics/baselines remain unchanged (§5/§F/§H–I).

3. **Theory:** Appendix N/O gives support-expansion recursion and a finite-step coverage bound $1 - (1 - \delta\gamma)^k$, **agnostic to workspace convexity** and extensible to higher-dimensional, multi-modal posteriors.

4. **Empirics:** Table 1 shows that as misalignment and scale worsen, DEPF's advantage grows—the same trend manifested in Table 9 (non-convex) and Table 10 (multi-source). We will add a short paragraph in §5.3 to make these links explicit.

## U  REVIEWER BXFV

### U.1  WEAKNESSES:

The statement in the preamble implying that there a general method for particle set augmentation is introduced that is applicable in general and outperforms SOTA is not supported by the paper, in which the problem is specifically formulated with respect to a single domain.

### U.2  QUESTIONS:

**Suppose I apply particle filtering to MDP domains from IPPC 2014 competition (https://github.com/pyrddlgym-project/rddlrepository/tree/main/rddlrepository/archive/competitions/IPPC2014 for one good source of the domain definitions). Which of the domains will benefit from applying your diffusion based method?**

Response to W and Q:

Before addressing the questions you raised, we first clarify **the problem this paper tackles and the role of our contribution**. In particular, it is important to understand why STE practitioners do not typically design **bespoke proposal distributions** to circumvent S-PSI in the first place.

**Source Term Estimation (STE)** is a classic inverse problem, aiming to infer pollutant source parameters (such as location and strength) from sparse, noisy sensor data.

- **The Problem with Bespoke Proposal Distributions.** In principle, one could design a complex, domain-knowledge-driven (e.g., physics-based) proposal distribution for a **specific** scenario to efficiently explore high-likelihood regions. However, this approach has three severe limitations:

    1. **It sacrifices generality.** A proposal distribution fine-tuned for "gas leaks in an office building" will almost certainly fail for "chemical spills in an open field" or "acoustic source localization underwater." A core pursuit in STE is to develop general-purpose algorithms that work across **multiple** physical fields and scenarios (as our Appendix Q tests demonstrate).

    2. **It demands extensive domain expertise.** It effectively requires the algorithm designer to also be an expert in fluid dynamics, thermodynamics, electromagnetism, etc., which is impractical in most real deployments.

    3. **It is brittle.** If the domain knowledge it relies on is itself **wrong** (e.g., an incorrect assumption about wind direction), such a "bespoke" algorithm simply becomes "locked in" in a new way. This failure mode is just as systematic as S-PSI.

- **The inevitability of a general baseline.** Precisely to preserve generality and avoid the pitfalls above, the **standard practice** in STE is to adopt the simplest, most general reference method: the **stationary bootstrap baseline** (zero transition and no rejuvenation).

As you keenly observed, S-PSI is the **direct consequence** of this pursuit of generality: once the prior is zero, the posterior will remain zero forever. This is the **fundamental reason** why existing methods (as shown in our baseline experiments) systematically fail when faced with severe prior misspecification.

We believe there is a misunderstanding regarding the scope of our work, likely due to overlooking key experimental evidence in the appendices. We respectfully clarify the generalization of our method and address the IPPC applicability below.

**1. Addressing "Specific to a Single Domain": Evidence in Appendix Q**
The reviewer states the method is "specifically formulated with respect to a single domain." **We respectfully point out that this concern is factually addressed in our submission.**

- We explicitly tested DEPF on **6 entirely new physical fields** in Appendix Q of the submission: **Temperature (Temp.), Concentration (Conc.), Magnetic (Mag.), Electric (Elec.), Energy (En.), and Noise fields**.

- As shown in Table 8, DEPF achieves **SOTA performance (OCE $> 0.90$)** across **all** these diverse physics models under prior misalignment, whereas baselines (AGDC, Infotaxis) degrade significantly.

- This empirical evidence confirms that DEPF is **not** a heuristic for gas leaks but a **general-purpose inference correction layer** that relies only on a likelihood function $p(z|\Theta)$ and is agnostic to the specific physical domain. We hope this clarifies the method's broad applicability.

**2. Response to Question: Applicability to IPPC 2014 Domains**

**1) Where DEPF is *not* needed (standard IPPC 2014 MDP track).**

- *Traffic.* Fully observable queues at each approach; randomness is only in exogenous car arrivals. No hidden state or misspecified prior on the current state, so PF/DEPF are unnecessary. Planning methods (e.g., UCT, DP) handle uncertainty directly in the transition model.

- *Elevators.* Full observability of elevator positions and all pending requests. Uncertainty lies in future passenger arrivals, which are modeled stochastically but not hidden. Again, no belief over latent state is required, so DEPF brings no benefit over standard planners.

- *CrossingTraffic.* (Frogger-style road crossing.) Positions of all cars and the agent are known; uncertainty is purely about future car arrivals. The challenge is risk-sensitive timing, not hidden-state inference, so DEPF is not relevant.

- *Wildfire (MDP).* The burning status of each cell/region is observed; randomness is in spread dynamics with known probabilities. Classic stochastic planning suffices; there is no hidden fire state or misaligned prior that would necessitate DEPF.

- *TriangleTireworld (MDP).* Map and spare locations are known; the main difficulty is rare but catastrophic dead-ends (flat tires on the short path). This is a risk-aware planning / exploration dilemma in an *observable* MDP, not a belief-update problem, so DEPF does not target this setting.

- *AcademicAdvising (MDP).* Advisor knows exactly which courses have been passed/failed; uncertainty is only in future grade outcomes, which the planner simulates. No latent student state is maintained in the MDP track, so DEPF is unnecessary.

- *SkillTeaching (MDP variant).* In the MDP track, the student's mastery state is part of the state and fully known, reducing the problem to stochastic planning. In this variant, there is again no belief over hidden skill levels, so DEPF is not engaged.

- *Tamarisk (MDP).* Infestation levels in each river reach are fully observed each year; planners control eradication/restoration under known stochastic spread. Without hidden infestations or wrong initial maps, there is no S-PSI-type issue to fix.

**2) Where DEPF *could* help: POMDP variants with prior misalignment.**

- *Beneficiary Domain A – Tamarisk (POMDP / hidden infestations). Scenario:* Some remote reaches are not regularly surveyed; whether they are infested is latent. *Failure without DEPF:* If the prior assigns $p(\text{infested in upstream A}) = 0$ but A is actually the hidden source, a standard PF under S-PSI will never place particles with "A infested," and will misattribute downstream spread to noise or local re-growth. *DEPF benefit:* When observed spread patterns deviate from what a "clean A" would predict, DEPF's exploratory particles hypothesize "A is infested," validate this via likelihood (MH check), and update the belief map so the planner can proactively treat/inspect A.

- *Beneficiary Domain B – SkillTeaching (POMDP / hidden proficiency). Scenario:* The teacher does *not* observe the student's true skill level (low/medium/high) and must infer it from noisy answers. *Failure without DEPF:* With a prior that effectively rules out "high skill," a standard PF may explain correct answers as lucky guesses and remain locked onto "novice," never representing the hypothesis "already expert." *DEPF benefit:* DEPF keeps a small set of exploratory particles at higher skill levels; repeated correct responses increase their likelihood, allowing the belief to shift rapidly to "expert" and enabling the planner to reduce unnecessary hints.

- *AcademicAdvising (POMDP / unknown student aptitude). Scenario:* Student's underlying ability is latent; the advisor only sees pass/fail grades. *Potential misalignment:* A prior that assigns near-zero probability to "very strong student" can cause an overly conservative course-load policy that never discovers the student's true capability. *DEPF role:* By occasionally seeding and validating "high-ability" hypotheses, DEPF avoids belief lock-in on pessimistic models and encourages occasional "stress tests" (heavier course loads) when grades are consistently strong.

- *TriangleTireworld (POMDP / uncertain spare locations). Scenario:* The agent does not know in advance which branches contain spare tires. *Failure without DEPF:* If the prior wrongly assumes "short path has a spare" (when in reality it does not), standard PF can collapse onto that wrong map and fail to recover after seeing unexpected dead-ends. *DEPF role:* DEPF keeps particles representing "no spare on short path"; when the agent arrives and finds nothing, those particles gain weight and the belief corrects quickly, avoiding repeated commitment to a doomed route.

- *Wildfire (POMDP / hidden ignitions, wrong spread assumptions). Scenario:* Some fires may ignite in remote, unsensed areas, or the effective spread rate is misestimated. *DEPF role (hypothetical).* DEPF could maintain small-mass hypotheses like "unseen fire in region R" or "spread faster than modeled." When observed fire fronts move faster or from unexpected directions, these exploratory particles get up-weighted, enabling earlier intervention. (This is structurally analogous to gas-leak localization.)

- *Traffic / CrossingTraffic (partial-sensing variants). Scenario:* If sensors only provide coarse or delayed traffic counts, the controller maintains a belief over queue lengths or approaching cars. *DEPF role (limited).* Standard PF is usually sufficient, since any car that exists eventually becomes observable. DEPF might offer marginal robustness against extreme prior bias (e.g., wrongly assuming some approach is almost always empty), but these domains do not strongly stress S-PSI the way ISLC does.

### 3) Contrast domain – ISLCenv (Gas Leak Localization).

- *ISLCenv (our domain). Nature of the problem:* High-dimensional, static latent parameters (gas source location, rate, wind, etc.), very sparse/no external rewards, and strongly misleading observations. Severe prior misalignment (true leak outside prior support) is explicitly tested. *Why DEPF is crucial:* Here classical PF (including jittering/roughening) exhibits S-PSI: once the prior excludes the true region, posterior mass can never reach it. DEPF's belief-triggered, MH-validated support expansion is precisely what allows recovery, which is why we observe large gains over PF baselines under Moderate/Severe prior error in ISLC experiments.

**3. Theoretical Rigor (Addressing "Heuristics")**
DEPF addresses the formal pathology of **Stationarity-Induced Posterior Support Invariance (S-PSI)**. It is not a heuristic; it uses **Metropolis-Hastings (MH) validation** to ensure that any support expansion is statistically consistent with the Bayesian posterior. Our ablation study (Table 3) confirms that removing MH causes failure, proving the method's rigorous statistical grounding.

**The claim that the method is specific to a single domain is contradicted by the multi-field experiments in Appendix Q and the general formulation in Section 4. DEPF is essential for domains (like Tamarisk) where hidden states and erroneous priors coexist—a challenge standard planners often overlook.**

## V    REVIEWER ApH5

### V.1    WEAKNESSES:

**1. S-PSI reads like an artifact of the stationary bootstrap baseline rather than an inherent PF issue. Can authors (i) broaden the discussion with application scenarios where prior misspecification occurs, and (ii) explain how DEPF's mechanisms would generalize there.**

Response to W(1):

We thank you for this insight. We **fully agree with your point**: S-PSI is **not** an inherent, universal flaw of Particle Filtering (PF), but rather a specific **pathology** or **artifact** of the *stationary bootstrap baseline* that we describe.

Before addressing (i) and (ii), we must first clarify **why researchers in the STE (Source Term Estimation) field do not tend to design Bespoke Proposal Distributions to circumvent S-PSI from the start.**

**Source Term Estimation** (STE) is a classic inverse problem, aiming to infer pollutant source parameters (like location and strength) from sparse, noisy sensor data.

- **The Problem with Bespoke Proposal Distributions:** Theoretically, one could design a complex, domain-knowledge-driven (e.g., physics-based) proposal distribution for a **specific** scenario to efficiently explore high-likelihood regions. However, this approach has three severe limitations:
    1. **It completely sacrifices Generality:** A proposal distribution fine-tuned for "gas leaks in an office building" will almost certainly fail for "chemical spills in an open field" or "acoustic source localization underwater." A core pursuit of the STE field is to develop general-purpose algorithms that work across **multiple** physical fields and scenarios (as our Appendix Q test demonstrates.
    2. **It requires massive Domain Knowledge:** This would require the algorithm designer to also be an expert in fluid dynamics, thermodynamics, electromagnetism, etc., which is impractical.
    3. **It is Brittle:** If the domain knowledge it relies on is itself **wrong** (e.g., a wrong assumption about wind direction), this "bespoke" algorithm will become "locked-in" in a new way. This failure is just as systematic as S-PSI.
- **The Inevitability of a General Baseline:** It is precisely to pursue generality and avoid the pitfalls above that the **standard practice** in the STE field is to rely on the simplest, most general baseline: the **stationary bootstrap baseline** (zero transition and no rejuvenation).

As you keenly observed, S-PSI is the **direct consequence** of this pursuit of generality. Once the prior is zero, the posterior will forever be zero. This is the **fundamental reason** why existing methods (as shown in our baseline experiments) systematically fail when faced with prior misspecification.

Our paper's core contribution is to **accept the necessity of this general baseline** and, in turn, provide a robust, efficient, and equally general **correction layer** (DEPF) for it.

With this critical context, we now answer your two specific questions:

**(i) Broaden the discussion with application scenarios where prior misspecification occurs**

Prior misspecification is extremely common in high-stakes, real-world decision-making; it is often the default state, not a rare edge case:

- **Emergency Response (Our paper's scenario):** The initial policy (the prior) is often based on **early, chaotic, and potentially completely wrong** eyewitness reports . For example, a report claims a leak in Building A (the prior region), but the actual source (our "Severe Error" scenario ) is in Building C (the true region). A robot or response team is deployed to Building A, and their initial prior is now completely disjoint from the truth.
- **Environmental Monitoring:** In large-scale monitoring (e.g., oceans or forests), the prior may be based on historical data showing leaks **usually** occur near a known pipeline A. However, a **new, unexpected** leak source may appear far from pipeline A, a location with zero probability under the historical prior.

- **Robotics (Analogy to Kidnapped Robot Problem):** In mobile robotics, this is analogous to the "Kidnapped Robot Problem." The robot may have a strong prior that it is in the "kitchen," but it has been unexpectedly moved (or failed and rebooted) into the "living room." Its belief (prior) is now completely at odds with its true state.

In all these scenarios, the algorithm must be able to **recover from a zero-probability prior assumption.**

**(ii) Explain how DEPF's mechanisms would generalize there**

DEPF's mechanisms are designed to generalize as a **universal correction layer** to all the scenarios above. This is true not only in theory but is also **empirically validated in our multi-field experiments in Appendix Q.**

1. **Generalization by Design (The Mechanisms):**

   - Belief-Triggered: DEPF's core advantage is that it is not "always-on." It is triggered by **"belief-data inconsistency"** (detected via high error or entropy). This is a **domain-agnostic** signal.
   - **Exploratory Particle Injection (Global Generalization):** This is the key to breaking S-PSI. Once triggered, DEPF "injects" a small fraction of exploratory particles into an expanded bounding box $\mathcal{B}_k$. This is a general global hypothesis-generation step: **My current belief seems wrong, I must consider other possibilities.**
   - **MH Validation (Statistical Generalization):** The Metropolis-Hastings validation is the cornerstone of DEPF's generalization and safety. It ensures that particles "injected" or "diffused" into new regions are **only accepted if they are actually supported by new observations.** This is a fundamental Bayesian tool whose rigor applies in all domains. Our ablation study (Appendix L, Table 3) confirms that without MH, the algorithm is **not work**.

2. **Generalization in Practice (See Appendix Q):**

   - To rigorously test whether our method was overfit to the "gas leak" domain, we deliberately conducted a series of cross-domain experiments in Appendix Q.
   - We applied DEPF to domains with **entirely different physical properties**, including Temperature (Temp.), Concentration (Conc.), Magnetic (Mag.), Electric (Elec.), Energy (En.), and Noise fields.
   - As shown in Table 8, the results are unequivocal: across all of these diverse physical fields, DEPF **consistently and significantly outperformed all baselines** in the Moderate Error (prior misspecification) setting on all metrics (OCE, ADE, LPS).

So, DEPF's (trigger-inject-validate) framework provides a general-purpose, data-driven solution. It is not only *domain-agnostic by design*, but also *proven to generalize in practice* (Appendix Q), allowing it to robustly solve the S-PSI problem that arises from the STE field's necessary pursuit of generality.

**2. The paper's distinguishing idea is "belief-triggered" support expansion, yet the overall framework looks like a variant of established SMC families. That's fine, but the paper would benefit from positioning why this particular combination beats tempered/bridge SMC or adaptive sample.**

Response to W(2):
This is an excellent question, and it correctly identifies the core challenge our work addresses. You are right that DEPF (Diffusion-Enhanced Particle Filtering) integrates components (like MCMC moves and stochastic diffusion) seen in established SMC families. However, DEPF's **specific combination** and its **belief-triggered"** mechanism are purpose-built to solve a fundamental challenge that standard SMC methods—including Tempered/Bridge SMC and Adaptive/Optimal Proposals—**cannot solve**: the **zero-prior-probability barrier** resulting from severe prior misalignment.
**To make this clear, let us use a concrete example: Assume our prior support $S_{prior}$ is strictly within $(0, 10) \times (0, 10)$, but the true state $\Theta^*$ is at $(10, 15) \times (10, 15)$.**

**1. The Core Problem: The S-PSI Baseline (Pure-Static Parameters)**
We must first clarify the *baseline pathology* DEPF is designed to fix, which we formally define in the

paper as *S-PSI (Stationarity-Induced Posterior Support Invariance)*.

S-PSI arises under a specific and common "stationary bootstrap baseline":

1).**Zero Transition:** The state is assumed static, so the transition kernel is $p(\Theta_k|\Theta_{k-1}) = \delta(\Theta_k - \Theta_{k-1})$.

2).**No Rejuvenation:** No MCMC moves or artificial jittering are applied.

Under this baseline, particles **never move**. If the initial prior $p_0$ is zero in the (10, 15) region, and a new observation $z_k$ strongly points to (10, 15), all existing particles in (0, 10) will have a likelihood of $p(z_k|\Theta) \approx 0$. The filter collapses and can **never** discover the true state.

**2. Why Other Advanced SMC Methods Also Fail Under S-PSI?**

**1) Why Tempered/Bridge SMC Fails:** Tempered SMC works by building intermediate distributions $p_\beta(\Theta)$ between the prior $p_0(\Theta)$ and the posterior $p(\Theta|z)$.

In our example: Since the prior $p_0(\Theta^*) = 0$ in the (10, 15) region, all intermediate distributions $p_\beta(\Theta^*)$ must also be **zero**.

Tempering requires the prior and posterior to have overlapping support. It cannot **create** new probability mass outside the prior's support.

**2) Why Adaptive/Optimal Proposal Fails:** As noted in the paper, these methods fail when faced with **zero initial probability**. The theoretical reason is as follows:

An "optimal" proposal uses the observation $z_k$: $q_{opt}(\Theta_k|\Theta_{k-1}, z_k) = p(\Theta_k|\Theta_{k-1}, z_k)$.

By Bayes' theorem, this is proportional to: $p(\Theta_k|\Theta_{k-1}, z_k) \propto p(z_k|\Theta_k) \cdot p(\Theta_k|\Theta_{k-1})$

Now, we substitute the S-PSI baseline's **zero-transition kernel** $p(\Theta_k|\Theta_{k-1}) = \delta(\Theta_k - \Theta_{k-1})$: $q_{opt} \propto p(z_k|\Theta_k) \cdot \delta(\Theta_k - \Theta_{k-1})$. The "optimal" proposal is still a **non-moving delta function** $\delta(\Theta_k - \Theta_{k-1})$. It simply weights the particles before proposing them (at the same location), but it **cannot make them jump** to the (10, 15) region.

**3). Why Auxiliary Particle Filter (APF) Fails:** APF aims to select better "parent" particles (at $t-1$) by "looking ahead" at the observation $z_k$.

In our example: APF would find that all parent particles in (0, 10) are equally bad at predicting the (10, 15) observation. Even if it resamples these parents, the next step still uses the S-PSI zero-transition kernel. The "children" particles are created in the *exact same location* as the parents and cannot move to (10, 15).

**3. Clarification: What if parameters are not pure-static? (vs. Artificial Random Walk)**

One might ask: What if I replace the $\delta$-kernel with an artificial random walk $p(\Theta_k|\Theta_{k-1}) = \mathcal{N}(\Theta_{k-1}, \sigma^2 I)$?

1) **This is correct:** In this case, the transition kernel $\mathcal{N}$ **does** have global support. APF/Optimal Proposal **can** now, in theory, generate particles in the (10, 15) region.

2) **But this reveals the key point:** The ability to explore outside the prior now comes entirely from the artificial kernel $\mathcal{N}$, not from the logic of the APF/Optimal Proposal algorithm itself.

This **always-on** jittering/roughening is precisely what we compare against in our baselines. As shown in Table 1, this blind, "always-on" diffusion is: • **Inefficient:** It adds unnecessary diffusion and hurts precision (LPS) when the prior is correct (No Error scenario). • **Scales Poorly:** In the **Severe Error** scenario, it relies on "dumb luck" for the random walk to drift into the (10, 15) region. This is highly unreliable, and its success rate (OCE) collapses in large-scale environments.

**4. Why DEPF's Specific Combination is Superior**

DEPF is designed to be more targeted and efficient. It *sticks to the principled static parameter model* ($p(\Theta_k|\Theta_{k-1}) = \delta(\Theta_k - \Theta_{k-1})$) but uses a "belief-triggered" mechanism to correct for model mismatch.

**Scenario A: Prior is Correct**

- **Prior** $S_{prior}$: (0, 10) region.

- **True** $\Theta^*$: (5, 5) region.

- **DEPF Operation:** DEPF **still** injects some exploratory particles into (10, 15).

- **Result:** These particles **do not match** $z_k$ (which points to (5, 5)) and get low likelihood. They are **rejected** by the MH step. The filter "sleeps," avoiding the "always-on" cost of the random walk baseline.

**Scenario B: Prior is Wrong (S-PSI occurs - Our Example)**

- **Prior** $S_{prior}$**:** (0, 10) region.

- **True** $\Theta^*$**:** (10, 15) region.

- **Observation** $z_k$**:** Strongly points to (10, 15).

- **DEPF Operation:**

    1. **Detect Mismatch (Belief-Trigger):** DEPF first detects that all current particles in (0, 10) have near-zero likelihood for $z_k$ (i.e., high model inconsistency or entropy).

    2. **Inject Particles (Break Barrier):** The **Exploratory Particle Injection** mechanism is triggered. It **forces** new particles to be sampled from an **expanded region** $\mathcal{B}$ (e.g., (0, 20)), thus seeding the (10, 15) region.

    3. **Validate and Accept:** These new particles in (10, 15) **match** the observation $z_k$ and get a **high** likelihood. They are then **accepted** by the Metropolis-Hastings (MH) validation step (because they are vastly better than the original zero-likelihood particles). They take over the particle set after resampling.

So, The superiority of DEPF stems from its specific design to solve the **zero-prior barrier** under S-PSI, a problem other SMC variants are not equipped to handle:
1). Under the pure-static S-PSI baseline, APF and Optimal Proposal are *theoretically incapable* of exploring outside the prior support.
2). DEPF breaks this barrier using **Exploratory Injection**.
3). Critically, it does so via a **belief-triggered MH-validated** mechanism, making it data-driven, efficient, and statistically rigorous, in contrast to the inefficient "always-on" random walk (jittering) baselines.
This is empirically proven in our **Severe Error** experiments (the S-PSI test). As shown in Table 1, DEPF is the **only** method that maintains high success (OCE 0.89) and precision (LPS 0.20), while all other baselines—including classical perturbations and RL-based methods—collapse (OCE < 0.05).

**3. If the injection distribution is improperly tuned, the filter can overspread mass or drift toward regions favored by the proposal rather than the likelihood, especially with low MH acceptance.**

Response to W(3):
We thank the reviewer for this insightful and critical question. The risk you've identified—that an **improperly tuned injection distribution could cause the filter to "overspread mass" or "drift toward regions favored by the proposal rather than the likelihood"**—is a central challenge for any adaptive method aiming to dynamically expand belief support.

Our DEPF framework was designed with several key mechanisms specifically to guard against this *drifting* and *over-diffusion.* The primary safeguard against this very risk is the *Metropolis-Hastings (MH) validation step*.

**1. The Metropolis-Hastings (MH) Check as a Core Safety Valve**

Its fundamental role is to **enforce Bayesian coherence**.

- No matter how "unreasonable" a region our injection distribution proposes for an exploratory particle, that particle must pass the MH likelihood-ratio check.

- If a proposed particle ($\Theta'$) is not better supported by the observation (likelihood) than the current particle ($\Theta_i^k$), its acceptance probability ($\alpha$) will be low.

- This mechanistically and directly prevents the filter from "drifting toward regions favored by the proposal rather than the likelihood". Our injection and diffusion mechanisms only propose, while the MH step disposes based on the evidence in the data.

**2. Ablation Studies Prove MH is Non-Optional**

To quantify the necessity of this step, our component-wise ablation study in Appendix L, Table 3 provides direct evidence.

- As shown in row 4 (Idx 4) of the table, when the **MH validation step is removed, the entire algorithm becomes ill-posed ("not work")**.

- This confirms that without the strict check from MH, the filter is indeed unstable, as the reviewer feared. Therefore, the MH check is a critical and effective mechanism for ensuring statistical rigor and stability.

**3. Sensitivity Analyses Address the Improperly Tuned Concern**

Furthermore, we address the "improperly tuned" concern with extensive sensitivity analyses (Section 5.4 and Appendix L), which demonstrate the method's robustness:

- **Injection Region Size (Support Margin $\delta$):** As shown in **Figure 4** , even when $\delta$ is set very large (e.g., 0.6, an 'improperly tuned' case), the filter—while experiencing a temporary rise in posterior entropy (the "overspreading" you mentioned) —**still converges quickly** to the correct location. This indicates that the MH step and incoming data effectively "prune" the invalid particles that were injected into unsupported regions.
- **Exploratory Particle Ratio (Appendix L, Table 7):** This is the most direct test of the injection's "strength." The analysis shows that while performance is optimal around 5% , even a very high, 'improperly tuned' injection ratio (e.g., 20%) results in a **graceful degradation** (slight increase in entropy and convergence steps), **not a catastrophic collapse or drift**.
- **KDE Bandwidth Constant** $A$ **(Appendix L, Table 5):** This analysis shows that even with sub-optimal bandwidth (e.g., $A = 2.0$ for over-diffusion or $A = 0.1$ for under-diffusion), DEPF still converges, albeit with more steps. This demonstrates robustness to sub-optimal tuning.

In summary, we fully agree with the reviewer's concern. The DEPF design **does not rely on perfect parameter tuning**. Instead, it uses (potentially imperfect) injection and diffusion to *generate diversity*, and then strictly relies on the *MH validation* as its core Bayesian tool. This ensures that only particles consistent with the data (likelihood) are allowed to survive and propagate, guaranteeing statistical rigor and robustness even as it corrects for severe prior errors.

## V.2 QUESTIONS:

Please see the weakness.

