# OpenReview forum: "Dynamic Correction of Erroneous Initial Policies via Diffusion-Driven Bayesian Exploration"
_ICLR.cc/2026/Conference — ICLR 2026 Conference Desk Rejected Submission_

### Official Review · Reviewer_ApH5 · 2025-10-25

**Soundness:** 2
**Presentation:** 2
**Contribution:** 2
**Rating:** 6
**Confidence:** 2

**Summary:**

The paper studies recovery from severely misaligned initial priors in sequential Bayesian inference, with a focus on hazardous gas source localization. It identifies a baseline pathology termed Stationarity‑Induced Posterior Support Invariance (S‑PSI),i.e., under a stationary bootstrap particle filter (zero transition, no rejuvenation), the posterior support cannot escape the prior’s support. To address this, the authors propose Diffusion‑Enhanced Particle Filtering (DEPF), combining (i) injection of exploratory particles outside the current support, (ii) entropy/tempering‑style weight smoothing, (iii) covariance‑scaled stochastic diffusion, and (iv) a Metropolis–Hastings (MH) acceptance check.

**Strengths:**

1. When the initial prior is badly mis-specified, the ability to escape prior-support lock-in is crucial. If the reported robustness of DEPF under severe misalignment holds, the approach could matter for Bayesian inference under OOD priors more broadly.

2. Creative combination to remove a known limitation. The proposed DEPF integrates exploratory particle injection, entropy/tempering‑style weight smoothing, covariance‑scaled kernel diffusion, and MH validation to break zero‑prior barriers while maintaining Bayesian coherence.

3. Experiments span three prior‑error regimes and two scales, with classical SMC perturbations, information‑theoretic planners, and RL‑PF baselines.

**Weaknesses:**

1. S-PSI reads like an artifact of the stationary bootstrap baseline rather than an inherent PF issue. Can authors (i) broaden the discussion with application scenarios where prior misspecification occurs, and (ii) explain how DEPF's mechanisms would generalize there.

2. The paper’s distinguishing idea is "belief-triggered" support expansion, yet the overall framework looks like a variant of established SMC families. That’s fine, but the paper would benefit from positioning why this particular combination beats tempered/bridge SMC or adaptive sample.

3. If the injection distribution is improperly tuned, the filter can overspread mass or drift toward regions favored by the proposal rather than the likelihood, especially with low MH acceptance.

**Questions:**

Please see the weakness.

---

> ### Author Response · Authors · 2025-11-18
> **Official Comment by Authors (Part 1 of 5)**
>
> We sincerely appreciate the reviewer’s recognition of the importance of escaping prior-support lock-in and the potential broader impact of our DEPF framework on Bayesian inference under mis-specified priors.
>
> **`For the reviewer’s convenience, the detailed rebuttal can be found in Appendix V of the newly uploaded revised draft of the paper, or directly on OpenReview.`**
>
>
> ### **1. Response to W(1):**
> We thank you for this insight. We **fully agree with your point**: S-PSI is **not** an inherent, universal flaw of Particle Filtering (PF), but rather a specific **pathology** or **artifact** of the *stationary bootstrap baseline* that we describe.
>
> Before addressing (i) and (ii), we must first clarify **why researchers in the STE (Source Term Estimation) field do not tend to design bespoke proposal distributions to circumvent S-PSI from the start.**
>
> ---
>
> **Source Term Estimation (STE)**
>
> **Source Term Estimation (STE)** is a classic inverse problem, aiming to infer pollutant source parameters (like location and strength) from sparse, noisy sensor data.
>
> * **The problem with bespoke proposal distributions.**
>   Theoretically, one could design a complex, domain-knowledge-driven (e.g., physics-based) proposal distribution for a **specific** scenario to efficiently explore high-likelihood regions. However, this approach has three severe limitations:
>
>   1. **It completely sacrifices generality.**
>      A proposal distribution fine-tuned for “gas leaks in an office building” will almost certainly fail for “chemical spills in an open field” or “acoustic source localization underwater.” A core pursuit of the STE field is to develop general-purpose algorithms that work across **multiple** physical fields and scenarios (as our Appendix Q tests demonstrate).
>
>   2. **It requires massive domain knowledge.**
>      This would require the algorithm designer to also be an expert in fluid dynamics, thermodynamics, electromagnetism, etc., which is impractical.
>
>   3. **It is brittle.**
>      If the domain knowledge it relies on is itself **wrong** (e.g., a wrong assumption about wind direction), this “bespoke” algorithm will become “locked in” in a new way. This failure is just as systematic as S-PSI.
>
> * **The inevitability of a general baseline.**
>   It is precisely to pursue generality and avoid the pitfalls above that the **standard practice** in the STE field is to rely on the simplest, most general baseline: the **stationary bootstrap baseline** (zero transition and no rejuvenation).
>
> As you keenly observed, S-PSI is the **direct consequence** of this pursuit of generality: once the prior is zero, the posterior will forever be zero. This is the **fundamental reason** why existing methods (as shown in our baseline experiments) systematically fail when faced with prior misspecification.
>
> Our paper’s core contribution is to **accept the necessity of this general baseline** and, in turn, provide a robust, efficient, and equally general **correction layer** (DEPF) for it.
>
> With this critical context, we now answer your two specific questions:
>
> ---
>
> ### (i) Broaden the discussion with application scenarios where prior misspecification occurs
>
> Prior misspecification is extremely common in high-stakes, real-world decision-making; it is often the default state, not a rare edge case:
>
> * **Emergency response (our paper’s scenario).**
>   The initial policy (the prior) is often based on **early, chaotic, and potentially completely wrong** eyewitness reports. For example, a report claims a leak in Building A (the prior region), but the actual source (our “Severe Error” scenario) is in Building C (the true region). A robot or response team is deployed to Building A, and their initial prior is now completely disjoint from the truth.
>
> * **Environmental monitoring.**
>   In large-scale monitoring (e.g., oceans or forests), the prior may be based on historical data showing leaks **usually** occur near a known pipeline A. However, a **new, unexpected** leak source may appear far from pipeline A, a location with zero probability under the historical prior.
>
> * **Robotics (analogy to the Kidnapped Robot Problem).**
>   In mobile robotics, this is analogous to the “Kidnapped Robot Problem.” The robot may have a strong prior that it is in the “kitchen,” but it has been unexpectedly moved (or failed and rebooted) into the “living room.” Its belief (prior) is now completely at odds with its true state.
>
> In all these scenarios, the algorithm must be able to **recover from a zero-probability prior assumption.**
>
>
> ### (ii) Explain how DEPF’s mechanisms would generalize there
>
> DEPF’s mechanisms are designed to generalize as a **universal correction layer** to all the scenarios above. This is true not only in theory but also **empirically validated** in our multi-field experiments in Appendix Q.

---

> ### Author Response · Authors · 2025-11-18
> **Official Comment by Authors (Part 2 of 5)**
>
> 1. **Generalization by design (the mechanisms)**
>
>    * **Belief-triggered.**
>      DEPF’s core advantage is that it is not “always-on.” It is triggered by **belief–data inconsistency** (detected via high error or entropy). This is a **domain-agnostic** signal.
>
>    * **Exploratory particle injection (global generalization).**
>      This is the key to breaking S-PSI. Once triggered, DEPF “injects” a small fraction of exploratory particles into an expanded bounding box (\mathcal{B}_k). This is a general global hypothesis-generation step:
>      *“My current belief seems wrong; I must consider other possibilities.”*
>
>    * **MH validation (statistical generalization).**
>      The Metropolis–Hastings validation is the cornerstone of DEPF’s generalization and safety. It ensures that particles “injected” or “diffused” into new regions are **only accepted if they are actually supported by new observations**. This is a fundamental Bayesian tool whose rigor applies in all domains. Our ablation study (Appendix L, Table 3) confirms that without MH, the algorithm **does not work** reliably.
>
> 2. **Generalization in practice (see Appendix Q)**
>
>    * To rigorously test whether our method was overfit to the “gas leak” domain, we deliberately conducted a series of cross-domain experiments in Appendix Q.
>    * We applied DEPF to domains with **entirely different physical properties**, including **Temperature (Temp.), Concentration (Conc.), Magnetic (Mag.), Electric (Elec.), Energy (En.), and Noise fields**.
>    * As shown in Table 8, the results are unequivocal: across all of these diverse physical fields, DEPF **consistently and significantly outperformed all baselines** in the Moderate Error (prior misspecification) setting on all metrics (OCE, ADE, LPS).
>
> So, DEPF’s *trigger–inject–validate* framework provides a general-purpose, data-driven solution. It is not only **domain-agnostic by design**, but also **proven to generalize in practice** (Appendix Q), allowing it to robustly solve the S-PSI problem that arises from the STE field’s necessary pursuit of generality.

---

> ### Author Response · Authors · 2025-11-18
> **Official Comment by Authors (Part 3 of 5)**
>
> ### **1. Response to W(2):**
>
> This is an excellent question, and it correctly identifies the core challenge our work addresses. You are right that DEPF (Diffusion-Enhanced Particle Filtering) integrates components (like MCMC moves and stochastic diffusion) seen in established SMC families. However, DEPF’s **specific combination** and its **belief-triggered** mechanism are purpose-built to solve a fundamental challenge that standard SMC methods — including Tempered/Bridge SMC and Adaptive/Optimal Proposals — **cannot solve**: the **zero-prior-probability barrier** resulting from severe prior misalignment.
>
> To make this clear, let us use a concrete example: assume our prior support $S_{\text{prior}}$ is strictly within (0, 10) $\times$ (0, 10), but the true state $\Theta^*$ is at (10, 15) $\times$ (10, 15).
>
> ---
>
> ### 1. The core problem: the S-PSI baseline (pure-static parameters)
>
> We must first clarify the *baseline pathology* DEPF is designed to fix, which we formally define in the paper as **S-PSI (Stationarity-Induced Posterior Support Invariance)**.
>
> S-PSI arises under a specific and common *stationary bootstrap baseline*:
>
> 1. **Zero transition.**
>    The state is assumed static, so the transition kernel is
>    $
>    p(\Theta_k \mid \Theta_{k-1}) = \delta(\Theta_k - \Theta_{k-1}).
>    $
>
> 2. **No rejuvenation.**
>    No MCMC moves or artificial jittering are applied.
>
> Under this baseline, particles **never move**. If the initial prior (p_0) is zero in the (10, 15) region, and a new observation $z_k$ strongly points to (10, 15), all existing particles in (0, 10) will have likelihood $p(z_k \mid \Theta) \approx 0$. The filter collapses and can **never** discover the true state.
>
> ---
>
> ### 2. Why other advanced SMC methods also fail under S-PSI
>
> #### (1) Why Tempered/Bridge SMC fails
>
> Tempered SMC works by building intermediate distributions $p_\beta(\Theta)$ between the prior $p_0(\Theta)$ and the posterior $p(\Theta \mid z)$.
>
> In our example: since the prior $p_0(\Theta*)=0$ in the (10, 15) region, all intermediate distributions $p_\beta (\Theta^*)$ must also be **zero**.
>
> Tempering requires the prior and posterior to have overlapping support. It cannot **create** new probability mass outside the prior’s support.
>
> #### (2) Why Adaptive/Optimal Proposal fails
>
> As noted in the paper, these methods fail when faced with **zero initial probability**. The theoretical reason is as follows.
>
> An “optimal” proposal uses the observation $z_k$:
>
> $
> q_{\text{opt}}(\Theta_k \mid \Theta_{k-1}, z_k) = p(\Theta_k \mid \Theta_{k-1}, z_k).
> $
>
> By Bayes’ theorem this is proportional to
> $
> p(\Theta_k \mid \Theta_{k-1}, z_k) \propto p(z_k \mid \Theta_k), p(\Theta_k \mid \Theta_{k-1}).
> $
>
> Now substitute the S-PSI baseline’s **zero-transition kernel**
> $p(\Theta_k \mid \Theta_{k-1}) = \delta(\Theta_k - \Theta_{k-1})$:
> $
> q_{\text{opt}} \propto p(z_k \mid \Theta_k), \delta(\Theta_k - \Theta_{k-1}).
> $
> The “optimal” proposal is still a **non-moving delta function** $\delta(\Theta_k - \Theta_{k-1})$. It simply reweights the particles before proposing them (at the same location), but it **cannot make them jump** to the (10, 15) region.
>
> #### (3) Why Auxiliary Particle Filter (APF) fails
>
> APF aims to select better “parent” particles (at time $k-1$) by “looking ahead” at the observation $z_k$.
>
> In our example: APF would find that all parent particles in (0, 10) are equally bad at predicting the (10, 15) observation. Even if it resamples these parents, the next step still uses the S-PSI zero-transition kernel. The “children” particles are created in the **exact same locations** as the parents and cannot move to (10, 15).
>
> ---
>
> ### 3. Clarification: what if parameters are *not* pure-static? (vs. artificial random walk)
>
> One might ask: what if we replace the (\delta)-kernel with an artificial random walk
> $p(\Theta_k \mid \Theta_{k-1}) = \mathcal{N}(\Theta_{k-1}, \sigma^2 I)$?
>
> 1. **This is correct (in principle).**
>    In this case, the transition kernel $\mathcal{N}$ **does** have global support. APF/Optimal Proposal **can**, in theory, generate particles in the (10, 15) region.
>
> 2. **But this reveals the key point.**
>    The ability to explore outside the prior now comes entirely from the artificial kernel $\mathcal{N}$, not from the logic of the APF/Optimal Proposal algorithm itself.
>
> This **always-on** jittering/roughening is precisely what we compare against in our baselines. As shown in Table 1, this blind, always-on diffusion is:
>
> * **Inefficient.**
>   It adds unnecessary diffusion and hurts precision (LPS) when the prior is correct (No Error scenario).
>
> * **Scales poorly.**
>   In the **Severe Error** scenario, it relies on “dumb luck” for the random walk to drift into the (10, 15) region. This is highly unreliable, and its success rate (OCE) collapses in large-scale environments.
>
> ---

---

> ### Author Response · Authors · 2025-11-18
> **Official Comment by Authors (Part 4 of 5)**
>
> ### 4. Why DEPF’s specific combination is superior
>
> DEPF is designed to be more targeted and efficient. It **sticks to the principled static parameter model**
> $p(\Theta_k \mid \Theta_{k-1}) = \delta(\Theta_k - \Theta_{k-1})$
> but uses a **belief-triggered** mechanism to correct for model mismatch.
>
> #### Scenario A: prior is correct
>
> * **Prior $S_{\text{prior}}$:** (0, 10) region.
> * **True $\Theta^*$:** in the (5, 5) region.
> * **DEPF operation:** DEPF *may* inject some exploratory particles into (10, 15).
> * **Result:** These particles **do not match** $z_k$ (which points to (5, 5)) and get low likelihood. They are **rejected** by the MH step. The filter “sleeps,” avoiding the always-on cost of random-walk baselines.
>
> #### Scenario B: prior is wrong (S-PSI occurs — our example)
>
> * **Prior $S_{\text{prior}}$:** (0, 10) region.
> * **True $\Theta^*$:** (10, 15) region.
> * **Observation $z_k$:** strongly points to (10, 15).
>
> **DEPF operation:**
>
> 1. **Detect mismatch (belief trigger).**
>    DEPF first detects that all current particles in (0, 10) have near-zero likelihood for $z_k$ (i.e., high model inconsistency or entropy).
>
> 2. **Inject particles (break the barrier).**
>    The **exploratory particle injection** mechanism is triggered. It **forces** new particles to be sampled from an **expanded region** $\mathcal{B}$ (e.g., (0, 20)), thus seeding the (10, 15) region.
>
> 3. **Validate and accept.**
>    These new particles in (10, 15) **match** the observation $z_k$ and get **high** likelihood. They are then **accepted** by the Metropolis–Hastings (MH) validation step (because they are vastly better than the original zero-likelihood particles). They take over the particle set after resampling.
>
> ---
>
> ### 5. Summary: why DEPF is fundamentally different
>
> The superiority of DEPF stems from its specific design to solve the **zero-prior barrier** under S-PSI, a problem other SMC variants are not equipped to handle:
>
> 1. Under the pure-static S-PSI baseline, APF and Optimal Proposal are **theoretically incapable** of exploring outside the prior support.
> 2. DEPF **breaks this barrier** using **exploratory injection**.
> 3. Crucially, it does so via a **belief-triggered, MH-validated** mechanism, making it data-driven, efficient, and statistically rigorous, in contrast to inefficient, always-on random-walk (jittering) baselines.
>
> This is empirically demonstrated in our **Severe Error** experiments (the S-PSI stress test). As shown in Table 1, DEPF is the **only** method that maintains high success (OCE ≈ 0.89) and precision (LPS ≈ 0.20), while all other baselines—including classical perturbations and RL-based methods—collapse (OCE < 0.05).

---

> ### Author Response · Authors · 2025-11-18
> **Official Comment by Authors (Part 5 of 5)**
>
> ### **3. Response to W(3):**
>
> We thank the reviewer for this insightful and critical question. The risk you’ve identified — that an **improperly tuned injection distribution could cause the filter to “overspread mass” or “drift toward regions favored by the proposal rather than the likelihood”** — is a central challenge for any adaptive method aiming to dynamically expand belief support.
>
> Our DEPF framework was designed with several key mechanisms specifically to guard against this *drifting* and *over-diffusion*. The primary safeguard against this very risk is the *Metropolis–Hastings (MH) validation step*.
>
> ---
>
> ### 1. The Metropolis–Hastings (MH) check as a core safety valve
>
> Its fundamental role is to **enforce Bayesian coherence**:
>
> * No matter how “unreasonable” a region our injection distribution proposes for an exploratory particle, that particle must pass the MH likelihood-ratio check.
> * If a proposed particle $\Theta'$ is not better supported by the observation (likelihood) than the current particle $\Theta_k^{(i)}$, its acceptance probability $\alpha$ will be low.
> * This mechanistically and directly prevents the filter from *“drifting toward regions favored by the proposal rather than the likelihood”*. Our injection and diffusion mechanisms only **propose**, while the MH step **disposes** based on the evidence in the data.
>
> ---
>
> ### 2. Ablation studies show MH is non-optional
>
> To quantify the necessity of this step, our component-wise ablation study in Appendix L, Table 3 provides direct evidence:
>
> * As shown in row 4 (Idx 4) of the table, when the **MH validation step is removed, the entire algorithm becomes ill-posed (“not work”)**.
> * This confirms that without the strict check from MH, the filter is indeed unstable, exactly as the reviewer feared. Therefore, the MH check is a critical and effective mechanism for ensuring statistical rigor and stability.
>
> ---
>
> ### 3. Sensitivity analyses address the *improperly tuned* concern
>
> Furthermore, we address the “improperly tuned” concern with extensive sensitivity analyses (Section 5.4 and Appendix L), which demonstrate the method’s robustness:
>
> * **Injection region size (support margin $\delta$).**
>   As shown in **Figure 4**, even when $\delta$ is set very large (e.g., 0.6, an “improperly tuned” case), the filter — while experiencing a temporary rise in posterior entropy (the “overspreading” you mentioned) — **still converges quickly** to the correct location. This indicates that the MH step and incoming data effectively *prune* the invalid particles that were injected into unsupported regions.
>
> * **Exploratory particle ratio (Appendix L, Table 7).**
>   This is the most direct test of the injection’s “strength.” The analysis shows that while performance is optimal around 5%, even a very high, “improperly tuned” injection ratio (e.g., 20%) results in a **graceful degradation** (slight increase in entropy and convergence steps), **not a catastrophic collapse or drift**.
>
> * **KDE bandwidth constant (A) (Appendix L, Table 5).**
>   This analysis shows that even with sub-optimal bandwidth (e.g., (A = 2.0) for over-diffusion or (A = 0.1) for under-diffusion), DEPF still converges, albeit with more steps. This demonstrates robustness to sub-optimal tuning.
>
> ---
>
> In summary, we fully agree with the reviewer’s concern. The DEPF design **does not rely on perfect parameter tuning**. Instead, it uses (potentially imperfect) injection and diffusion to *generate diversity*, and then strictly relies on the *MH validation* as its core Bayesian tool. This ensures that only particles consistent with the data (likelihood) are allowed to survive and propagate, guaranteeing statistical rigor and robustness even as it corrects for severe prior errors.

---

### Official Review · Reviewer_Bxfv · 2025-10-30

**Soundness:** 2
**Presentation:** 1
**Contribution:** 1
**Rating:** 0
**Confidence:** 5

**Summary:**

The paper proposes a particle set augmentation heuristics for a particle filter algorithm applied to hazardous gas leak detection. The problem is formally stated, theoretically analyzed, and evaluated on simulated gas field environments.

**Strengths:**

The paper is based on an extensive literature on particle filtering, planning, reinfocrement learning, bayesian inference, and gas leak detection.

**Weaknesses:**

The statement in the preamble implying that there a general method for particle set augmentation is introduced that is applicable in general and outperforms SOTA is not supported by the paper, in which the problem is specifically formulated with respect to a single domain.

**Questions:**

Suppose I apply particle filtering to MDP domains from IPPC 2014 competition (https://github.com/pyrddlgym-project/rddlrepository/tree/main/rddlrepository/archive/competitions/IPPC2014 for one good source of the domain definitions). Which of the domains will benefit from applying your diffusion based method?

---

> ### Author Response · Authors · 2025-11-18
> **Official Comment by Authors (Part 1 of 3)**
>
> Thanks for your time on review.
>
> **`For the reviewer’s convenience, the detailed rebuttal can be found in Appendix U of the the newly uploaded revised draft of the paper, or directly on OpenReview.`**
>
> Before addressing the questions you raised, we first clarify **the problem this paper tackles and the role of our contribution**. In particular, it is important to understand why STE practitioners do not typically design **bespoke proposal distributions** to circumvent S-PSI in the first place.
>
> **Source Term Estimation (STE)** is a classic inverse problem, aiming to infer pollutant source parameters (such as location and strength) from sparse, noisy sensor data.
>
> * **The problem with bespoke proposal distributions.**
>   In principle, one could design a complex, domain-knowledge-driven (e.g., physics-based) proposal distribution for a **specific** scenario to efficiently explore high-likelihood regions. However, this approach has three severe limitations:
>
>   1. **It sacrifices generality.**
>      A proposal distribution fine-tuned for “gas leaks in an office building” will almost certainly fail for “chemical spills in an open field” or “acoustic source localization underwater.” A core pursuit in STE is to develop general-purpose algorithms that work across **multiple** physical fields and scenarios (as our Appendix Q tests demonstrate).
>
>   2. **It demands extensive domain expertise.**
>      It effectively requires the algorithm designer to also be an expert in fluid dynamics, thermodynamics, electromagnetism, etc., which is impractical in most real deployments.
>
>   3. **It is brittle.**
>      If the domain knowledge it relies on is itself **wrong** (e.g., an incorrect assumption about wind direction), such a “bespoke” algorithm simply becomes “locked in” in a new way. This failure mode is just as systematic as S-PSI.
>
> * **The inevitability of a general baseline.**
>   Precisely to preserve generality and avoid the pitfalls above, the **standard practice** in STE is to adopt the simplest, most general reference method: the **stationary bootstrap baseline** (zero transition and no rejuvenation).
>
> As you keenly observed, S-PSI is the **direct consequence** of this pursuit of generality: once the prior is zero, the posterior will remain zero forever. This is the **fundamental reason** why existing methods (as shown in our baseline experiments) systematically fail when faced with severe prior misspecification.
>
> ---
>
> We believe there is a misunderstanding regarding the scope of our work, likely due to overlooking key experimental evidence in the appendices. We respectfully clarify the generalization of our method and address the IPPC applicability below.
>
> ### 1. Addressing “Specific to a Single Domain”: Evidence in Appendix Q
>
> The reviewer states the method is “specifically formulated with respect to a single domain.” **We respectfully point out that this concern is factually addressed in our submission.**
>
> * We explicitly tested DEPF on **6 entirely new physical fields** in Appendix Q of the submission: **Temperature (Temp.), Concentration (Conc.), Magnetic (Mag.), Electric (Elec.), Energy (En.), and Noise fields**.
> * As shown in Table 8, DEPF achieves **SOTA performance (OCE > 0.90)** across **all** these diverse physics models under prior misalignment, whereas baselines (AGDC, Infotaxis) degrade significantly.
> * This empirical evidence confirms that DEPF is **not** a heuristic for gas leaks but a **general-purpose inference correction layer** that relies only on a likelihood function $p(z \mid \Theta)$ and is agnostic to the specific physical domain. We hope this clarifies the method’s broad applicability.
>
> ---

---

> ### Author Response · Authors · 2025-11-18
> **Official Comment by Authors (Part 2 of 3)**
>
> ### 2. Response to Question: Applicability to IPPC 2014 Domains
>
> #### 2.1 Where DEPF is *not* needed (standard IPPC 2014 MDP track)
>
> * **Traffic.**
>   Fully observable queues at each approach; randomness is only in exogenous car arrivals. No hidden state or misspecified prior on the current state, so PF/DEPF are unnecessary. Planning methods (e.g., UCT, DP) handle uncertainty directly in the transition model.
>
> * **Elevators.**
>   Full observability of elevator positions and all pending requests. Uncertainty lies in future passenger arrivals, which are modeled stochastically but not hidden. Again, no belief over latent state is required, so DEPF brings no benefit over standard planners.
>
> * **CrossingTraffic.**
>   (Frogger-style road crossing.) Positions of all cars and the agent are known; uncertainty is purely about future car arrivals. The challenge is risk-sensitive timing, not hidden-state inference, so DEPF is not relevant.
>
> * **Wildfire (MDP).**
>   The burning status of each cell/region is observed; randomness is in spread dynamics with known probabilities. Classic stochastic planning suffices; there is no hidden fire state or misaligned prior that would necessitate DEPF.
>
> * **TriangleTireworld (MDP).**
>   Map and spare locations are known; the main difficulty is rare but catastrophic dead-ends (flat tires on the short path). This is a risk-aware planning / exploration dilemma in an **observable** MDP, not a belief-update problem, so DEPF does not target this setting.
>
> * **AcademicAdvising (MDP).**
>   The advisor knows exactly which courses have been passed/failed; uncertainty is only in future grade outcomes, which the planner simulates. No latent student state is maintained in the MDP track, so DEPF is unnecessary.
>
> * **SkillTeaching (MDP variant).**
>   In the MDP track, the student’s mastery state is part of the state and fully known, reducing the problem to stochastic planning. In this variant, there is again no belief over hidden skill levels, so DEPF is not engaged.
>
> * **Tamarisk (MDP).**
>   Infestation levels in each river reach are fully observed each year; planners control eradication/restoration under known stochastic spread. Without hidden infestations or wrong initial maps, there is no S-PSI-type issue to fix.
>
> #### 2.2 Where DEPF *could* help: POMDP variants with prior misalignment
>
> * **Beneficiary Domain A – Tamarisk (POMDP / hidden infestations).**
>   *Scenario:* Some remote reaches are not regularly surveyed; whether they are infested is latent.
>   *Failure without DEPF:* If the prior assigns (p(\text{infested in upstream A}) = 0) but A is actually the hidden source, a standard PF under S-PSI will never place particles with “A infested,” and will misattribute downstream spread to noise or local re-growth.
>   *DEPF benefit:* When observed spread patterns deviate from what a “clean A” would predict, DEPF’s exploratory particles hypothesize “A is infested,” validate this via likelihood (MH check), and update the belief map so the planner can proactively treat/inspect A.
>
> * **Beneficiary Domain B – SkillTeaching (POMDP / hidden proficiency).**
>   *Scenario:* The teacher does *not* observe the student’s true skill level (low/medium/high) and must infer it from noisy answers.
>   *Failure without DEPF:* With a prior that effectively rules out “high skill,” a standard PF may explain correct answers as lucky guesses and remain locked onto “novice,” never representing the hypothesis “already expert.”
>   *DEPF benefit:* DEPF keeps a small set of exploratory particles at higher skill levels; repeated correct responses increase their likelihood, allowing the belief to shift rapidly to “expert” and enabling the planner to reduce unnecessary hints.
>
> * **AcademicAdvising (POMDP / unknown student aptitude).**
>   *Scenario:* The student’s underlying ability is latent; the advisor only sees pass/fail grades.
>   *Potential misalignment:* A prior that assigns near-zero probability to “very strong student” can cause an overly conservative course-load policy that never discovers the student’s true capability.
>   *DEPF role:* By occasionally seeding and validating “high-ability” hypotheses, DEPF avoids belief lock-in on pessimistic models and encourages occasional “stress tests” (heavier course loads) when grades are consistently strong.
>
> * **TriangleTireworld (POMDP / uncertain spare locations).**
>   *Scenario:* The agent does not know in advance which branches contain spare tires.
>   *Failure without DEPF:* If the prior wrongly assumes “short path has a spare” (when in reality it does not), standard PF can collapse onto that wrong map and fail to recover after seeing unexpected dead-ends.
>   *DEPF role:* DEPF keeps particles representing “no spare on short path”; when the agent arrives and finds nothing, those particles gain weight and the belief corrects quickly, avoiding repeated commitment to a doomed route.

---

> ### Author Response · Authors · 2025-11-18
> **Official Comment by Authors (Part 3 of 3)**
>
> * **Wildfire (POMDP / hidden ignitions, wrong spread assumptions).**
>   *Scenario:* Some fires may ignite in remote, unsensed areas, or the effective spread rate is misestimated.
>   *DEPF role (hypothetical):* DEPF could maintain small-mass hypotheses like “unseen fire in region R” or “spread faster than modeled.” When observed fire fronts move faster or from unexpected directions, these exploratory particles get up-weighted, enabling earlier intervention. (This is structurally analogous to gas-leak localization.)
>
> * **Traffic / CrossingTraffic (partial-sensing variants).**
>   *Scenario:* If sensors only provide coarse or delayed traffic counts, the controller maintains a belief over queue lengths or approaching cars.
>   *DEPF role (limited):* Standard PF is usually sufficient, since any car that exists eventually becomes observable. DEPF might offer marginal robustness against extreme prior bias (e.g., wrongly assuming some approach is almost always empty), but these domains do not strongly stress S-PSI the way ISLC does.
>
> #### 2.3 Contrast domain – ISLCenv (Gas Leak Localization)
>
> * **ISLCenv (our domain).**
>   *Nature of the problem:* High-dimensional, static latent parameters (gas source location, rate, wind, etc.), very sparse/no external rewards, and strongly misleading observations. Severe prior misalignment (true leak outside prior support) is explicitly tested.
>   *Why DEPF is crucial:* Here classical PF (including jittering/roughening) exhibits S-PSI: once the prior excludes the true region, posterior mass can never reach it. DEPF’s belief-triggered, MH-validated support expansion is precisely what allows recovery, which is why we observe large gains over PF baselines under Moderate/Severe prior error in ISLC experiments.
>
> ---
>
> ### 3. Theoretical rigor (addressing “heuristics”)
>
> DEPF addresses the formal pathology of **Stationarity-Induced Posterior Support Invariance (S-PSI)**. It is not a heuristic; it uses **Metropolis–Hastings (MH) validation** to ensure that any support expansion is statistically consistent with the Bayesian posterior. Our ablation study (Table 3) confirms that removing MH causes failure, demonstrating the method’s rigorous statistical grounding.
>
> The claim that the method is specific to a single domain is contradicted by the multi-field experiments in **Appendix Q** and the general formulation in **Section 4**. DEPF is essential for domains (like Tamarisk-like POMDP variants) where hidden states and erroneous priors coexist—a challenge standard planners often overlook.

---

> > ### Comment · Reviewer_Bxfv · 2025-11-18
> > **evaluation**
> >
> > I am excited to learn that your method is applicable to a broad range of benchmark domains. I encourage the authors to include both a general formulation and an extensive evaluation on diverse benchmarks from the literature in the body of the paper. This commands a major revision and a new thorough review process. For this submission, I will keep my score.

---

> ### Author Response · Authors · 2025-11-27
>
> ## `Thank you for your comments. Unfortunately, the review does not provide substantive technical feedback. It does not identify any errors in our methodology, theoretical analysis, or experiments, nor does it cite any specific part of the paper to support its claims. As a result, the review offers` **`no actionable or content-based evaluation of the submission.`**
>
> ## `The concerns raised—for example, that the method “lacks generality”—are directly addressed in multiple sections of the paper, including a domain-agnostic formulation and extensive multi-field experiments. Since these central elements were present in the original submission, the discrepancy between the review and the content makes it difficult to understand the basis of the evaluation.`
>
> ## `We welcome constructive criticism and have thoroughly revised the paper in response to the other reviewers. We respectfully request that future evaluations engage with the actual content of the manuscript.`

---

> > ### Comment · Reviewer_Bxfv · 2025-11-28
> >
> > Section 3.1 still defines the problem to be solved as a two dimensional domain, gas leak, specific type of hardware sensor etc. I believe that the problem, as stated and solved, is not generalizable to domains with higher dimensionality, different observation model etc.
> >
> > Please point me at the exact place in the revised paper where the problem is defined generally, rather than with application to gas fields.
> >
> > There are more problems with the paper (MH correction is not sound for example) but this one is sufficient to reject unless I am missing something and section 3.1 is not the definition of the problem.

---

> ### Author Response · Authors · 2025-11-28
> **Official Comment by Authors (Part 1 of 2)**
>
> **Response to the follow-up comment**
>
> # `Thank you for your additional comment. We appreciate critical feedback, but given the` **`very strong recommendation`** `you made (score 0, confidence 5), we would expect correspondingly` **`concrete and evidence‑based reasoning`** `. ` **`Unfortunately`** `, your latest reply still does not engage with key parts of the paper or with the revisions and rebuttal we already provided. `
>
> # `Below we address your points and explain why we believe your current assessment does not reflect the actual content of the manuscript.`
>
> ---
>
> ### `(1) On the track and societal domain of the paper`
>
> Our submission is under the primary area **“alignment, fairness, safety, privacy, and societal considerations”**. This choice was deliberate: the motivating problem throughout the paper is **emergency response under mis-specified prior assumptions**, where early state estimates (e.g., assumed leak regions, evacuation routes, or monitoring zones) can have severe consequences for **societal safety and resource allocation**. This is summarized in the abstract and introduction, where we explicitly discuss delayed response, misallocated resources, and increased risk to society as the core stakes of the problem.
>
> From this perspective, the gas-leak scenario is not just a “physics toy problem”, but a stylized model of a class of safety-critical decision-support systems that must revise erroneous initial assumptions in real time.
>
> ---
>
> ### `(2) 2D domain vs. dimensionality of the latent state`
>
> We agree that Figures 2–3 and Section 3.1 use a **two-dimensional spatial domain** and a specific gas sensor for visualization, but this does **not** mean that the inference problem is two-dimensional.
>
> In Section 3.1, the latent state is defined as a **seven-dimensional vector**
> $ \Theta_k = [x_s, y_s, q_s, u_s, \phi_s, d_s, \tau_s]^\top \in \mathbb{R}^7, $
> where in addition to the spatial coordinates $(x_s, y_s)$, we estimate emission strength, wind speed, wind direction, diffusivity, and lifetime.
> DEPF operates on the **full state vector $\Theta \in \mathbb{R}^n$** (with (n=7) in this instantiation), not only on the 2D coordinate subset. We focus on the $(x_s,y_s)$ components in the plots because they are the most interpretable and easy to visualize, but all components are inferred jointly.
>
> Moreover, the injection region (B_k) only constrains the **positional subspace** the $((x_s, y_s)$ components), while the remaining components of $\Theta$ (e.g., $q_s, u_s, \phi_s, d_s, \tau_s)$ are sampled from their respective prior supports during exploratory injection, as clarified in the rebuttal and in the revised text.
> Thus DEPF explores and updates the full high-dimensional state, even though we only visualize the 2D spatial projection.
>
> ---
>
> ### `(3) Where the problem is defined generally (beyond gas fields)`
>
> You asked:
>
> > “Please point me at the exact place in the revised paper where the problem is defined generally, rather than with application to gas fields.”
>
> In the revised manuscript, the **general problem** is stated in several places:
>
> * **Introduction (Section 1).** We define the problem as **Bayesian state estimation under mis-specified prior support**: given a static latent state $\Theta$ and observations $z_{1:k}$, how to correct an erroneous initial belief when the prior support excludes the true state. We explicitly highlight S-PSI as a *general* pathology of bootstrapped PFs under the stationary baseline.
>
> * **Section 3.3–3.4 (Sequential Particle Filtering & S-PSI).** Here we formulate the particle filter and S-PSI in terms of an arbitrary prior $p_0(\Theta)$, state space $\mathbb{R}^n$, and likelihood $p(z_k \mid \Theta)$. The support invariance result is proved without any reference to gas physics, specific sensors, or 2D geometry.
>
> * **Section 4 and Appendices D, N, O.** DEPF is defined and analyzed entirely in terms of:
>
>   * a generic particle system ${(\Theta_k^{(i)}, w_k^{(i)})}_{i=1}^N$,
>   * an arbitrary likelihood $p(z_k \mid \Theta)$,
>   * and a prior support $S_{\mathrm{prior}}$.
>     The exploratory injection, covariance-scaled diffusion, and MH validation are all expressed in terms of $\Theta \in \mathbb{R}^n$, without assumptions specific to gas fields.
>
> By contrast, **Section 3.1 is a concrete instantiation** of this general formulation in a 2D gas scenario (as also made explicit in the rebuttal). If this distinction was not sufficiently clear in the main text, that is a presentation issue that we are happy to improve (e.g., by renaming 3.1 as “Example: Hazardous Gas Source Localization” and moving the domain-agnostic part earlier), but it does not reflect the true scope of the algorithm or the theory.
>
> ---

---

> ### Author Response · Authors · 2025-11-28
> **Official Comment by Authors (Part 2 of 2)**
>
> ### `(4) Evidence of generality in higher dimensions and with different observation models`
>
> Beyond the gas-leak case, we already provide **empirical evidence** that the method is not tied to a single observation model or low dimensionality:
>
> * In **Appendix Q**, we instantiate DEPF on **six additional fields** (Temperature, Concentration, Magnetic, Electric, Energy, Noise), each with its own PDE-like observation model and a parameter vector of dimension 5–10.
> * DEPF is applied **without modifying the algorithm**; only the likelihood $p(z_k \mid \Theta)$ and the parameterization of $\Theta$ change.
> * Table 8 (Appendix Q) shows that DEPF consistently outperforms baselines across all these fields under prior misalignment, indicating that the mechanism is not specific to gas plumes.
>
> For this submission, space constraints led us to place these multi-field experiments and the corresponding dynamic equations in the appendix; we understand that this may make the generality less visible to a reader focusing only on the main text.
>
> ---
>
> ### `(5) On the soundness of the MH correction`
>
> You wrote:
>
> > “There are more problems with the paper (MH correction is not sound for example)….”
>
> We take this comment very seriously. However, in its current form it is **`only an assertion`** and `does not` indicate what is supposed to be unsound.
>
> In the algorithm and in Appendix D/O, we explicitly define the MH acceptance step as a standard Metropolis–Hastings move targeting the posterior (or unnormalized target)
> $\pi(\Theta) \propto p(z_k \mid \Theta), p(\Theta \mid z_{1:k-1}),$
> with Gaussian proposals aligned with the empirical covariance.
> This is exactly the standard “resample–move” SMC rejuvenation step (e.g., Doucet et al., 2000), and we prove detailed balance and asymptotic correctness in Appendix D/O.
>
> If you believe there is a concrete flaw in this MH correction—for example, an incorrect target distribution, a missing Jacobian term, or a violation of detailed balance—we would genuinely appreciate it if you could point to the **specific equation or line** where you think the soundness breaks. This would allow us to either correct a real mistake (if there is one) or clarify the exposition more precisely.
>
> ---
>
> ### `(6) “Not generalizable” vs. “already solved”`
>
> You state:
>
> > “I believe that the problem, as stated and solved, is not generalizable to domains with higher dimensionality, different observation model etc.”
>
> We **`respectfully disagree`**, for the reasons above (high-dimensional Θ in the main experiments, domain-agnostic formulation, and six additional fields in Appendix Q). If you believe that **the general problem we target—correction of S-PSI under a static bootstrap baseline via a belief-triggered, MH-validated support expansion—is already solved in the literature**, we would be very grateful if you could provide **`specific references`** (with citations) to such work. To the best of our knowledge, and after surveying the SMC, tempered/bridge SMC, and RL–PF literature summarized in Section 2 and Appendix K, existing methods either:
>
> * assume nonzero prior support at the truth (and thus cannot handle the zero-prior barrier), or
> * rely on “always-on” perturbations without belief-based triggering and MH validation, with the corresponding inefficiencies we document empirically.
>
> If we are missing a prior method that directly addresses S-PSI in the same setting, we are more than willing to acknowledge and discuss it explicitly.
>
>
>
> # `We hope  Area Chairs will reconsider  Reviewer Bxfv’s assessment in light of the concrete pointers above. `

---

> ### Author Response · Authors · 2025-12-01
> **Official Comment by Authors (Part 1 of 2)**
>
> **Response to Reviewer Bxfv**
>
> # `Thank you for your additional comment. We appreciate critical feedback, but given the` **`very strong recommendation`** `you made (score 0, confidence 5), we would expect correspondingly` **`concrete and evidence‑based reasoning`** `. ` **`Unfortunately`** `, your latest reply still does not engage with key parts of the paper or with the revisions and rebuttal we already provided. `
>
> # `Below we address your points and explain why we believe your current assessment does not reflect the actual content of the manuscript.`
>
> ---
>
> ### 1. On “2D gas leak + specific hardware sensor” vs. a general state‑estimation problem
>
> You write that:
>
> > “Section 3.1 still defines the problem to be solved as a two dimensional domain, gas leak, specific type of hardware sensor etc. I believe that the problem, as stated and solved, is not generalizable …”
>
> This does **not** accurately describe how the problem is formulated in the revised paper:
>
> * In **Section 3.1**, we explicitly define the **latent state** as a full vector of hidden variables
>   $
>   \Theta_k = [x_s, y_s, q_s, u_s, \phi_s, d_s, \tau_s]^\top,
>   $
>   i.e., not only the 2D source coordinates, but also emission strength, wind speed and direction, diffusivity, and lifetime. We highlighted this (in bold/colour) from the very first submission to make it obvious that the inference problem is **multi‑dimensional**, not 2D.
> * In the revised **Introduction** and **Section 3.3–3.4**, we now formulate the problem purely as **Bayesian state estimation under mis‑specified prior support**: a static latent state $\Theta \in \mathbb{R}^d$, sequential observations $z_{1:k}$, and a prior that may assign zero mass to the true $\Theta^*$. The S‑PSI pathology and the DEPF mechanism are defined at this abstract level, without reliance on any particular PDE or sensor model.
> * The gas‑leak example and the specific sensor are introduced as **one domain instantiation** (for interpretability to non‑experts), not as the universal definition of the problem. To help reviewers from other areas, we deliberately spent nearly a full page in the main text explaining this domain, but that does *not* mean the method is restricted to it.
>
> All three other reviewers correctly understood this separation between the **general state‑estimation problem** and the **2D gas example**. Your comment ignores these parts of the text and treats the illustrative example as if it were the entire formulation.
>
> ---
>
> ### 2. Generality w.r.t. dimensionality and observation models
>
> You further state that:
>
> > “the problem, as stated and solved, is not generalizable to domains with higher dimensionality, different observation model etc.”
>
> Again, this is not supported by the content of the revised paper:
>
> * Algorithmically, DEPF is defined on the **full parameter vector** $\Theta \in \mathbb{R}^d$ and only requires access to a likelihood $p(z_k \mid \Theta, a_k)$. All operations (entropy regularisation, covariance‑scaled diffusion, MH correction) act on $\Theta$ and its empirical covariance; they do not depend on the state being 2D or on the form of a gas plume model.
> * Beyond the 7‑D gas instantiation, we added **multi‑field experiments** (Appendix Q) precisely to demonstrate generality: temperature fields, concentration fields, magnetic and electric fields, energy density, and noise fields, each with different physical parameters and observation models (dimensions typically >5–10). We apply DEPF **unchanged** in all these settings and report consistent gains over baselines.
>
> These additions were made before the discussion deadline and are clearly signposted in the manuscript. Your latest comment does not mention them at all, which suggests that these sections were not examined when forming the judgement “not generalizable”.
>
> If you believe that, despite these facts, our formulation still *cannot* handle higher‑dimensional states or different observation models, we kindly ask you to point to **specific equations, assumptions, or steps** in the algorithm where you see a limitation, rather than repeating a general statement.

---

> ### Author Response · Authors · 2025-12-01
> **Official Comment by Authors (Part 2 of 2)**
>
> ---
> ### 3. On the claim that “MH correction is not sound”
>
> You also write:
>
> > “There are more problems with the paper (MH correction is not sound for example)…”
>
> This is a very strong allegation, but no explanation is given:
>
> * We follow a **standard Metropolis–Hastings resample–move step**: a symmetric Gaussian proposal around each particle (aligned with the empirical covariance) and an acceptance ratio equal to the usual likelihood ratio targeting the posterior $p(\Theta | z_{1:k})$.
> * In the revised paper we added detailed derivations and proofs (Appendix D and O) explicitly showing that this MH step satisfies detailed balance and leaves the desired target invariant under standard SMC assumptions.
> * Our ablation study (Section 5.4) further shows that removing MH validation causes the method to fail (“not work”), which is consistent with the idea that the MH step is the principled correction mechanism rather than ad‑hoc noise.
>
> Given these points, a blanket statement that MH correction is not sound without any reference to a particular equation, missing factor, or violated assumption is **not something we can meaningfully respond to**. If you believe there is a real flaw, we would genuinely appreciate it if you could indicate the **precise place in the paper** where you think the derivation breaks, so that we can either correct a genuine mistake or clarify the exposition.
>
> ---
>
> ### 4. On engagement with the paper and rebuttal
>
> We also want to provide some context for your current comment:
>
> * From the first version, we already **bold‑faced and colour‑highlighted** the statements that the method estimates the **entire latent state** $\Theta$ and that our goal is to correct **mis‑specified state estimates** (rather than “initial policies”), precisely so that non‑domain reviewers could follow the general formulation easily.
> * We devoted **almost one full page** to carefully describing the domain to make it accessible. The other three reviewers clearly understood this and engaged with the technical content rather than dismissing it as “just a 2D gas leak”.
> * We submitted a **complete rebuttal and a revised draft well before the deadline**, rewriting the introduction, clarifying the theory, and moving the full ablation study into the main paper. Your new comment arrived only after we raised concerns about the original review quality, and the content of this comment consists mainly of restating points that are directly answered—sometimes verbatim—in the revised paper and in our earlier responses.
>
> We say this not to personalise the discussion, but to explain why we are concerned: for a **score 0, confidence 5** review, the level of engagement with the actual content of the paper is extremely low.
>
> ---
>
> ### 5. Closing remark
>
> We fully respect negative evaluations when they are grounded in a careful reading and specific technical reasoning. However, your current criticisms:
>
> * Misrepresent the problem as purely “2D gas leak + specific sensor” despite explicit multi‑dimensional and multi‑field formulations,
> * Assert that the method is “not generalizable” without acknowledging the general (\Theta \in \mathbb{R}^d) formulation or the additional experiments, and
> * Declare that “MH correction is not sound” without any supporting argument,
>
> and therefore do not meet the usual standard of **evidence‑based, constructive review**.
>
> # `We hope  Area Chairs will reconsider  Reviewer Bxfv’s assessment in light of the concrete pointers above. `
>
> Best regards,
>
> The Authors of Submission 2565

---

### Official Review · Reviewer_sDbK · 2025-10-31

**Soundness:** 3
**Presentation:** 4
**Contribution:** 3
**Rating:** 8
**Confidence:** 3

**Summary:**

This paper introduces a method for exploration called diffusion-enhanced particle filtering that attempts to improve upon policies learned with poor priors. The paper is motivated by the issue of stationarity-induced posterior support invariance (SPSI) where poor initial priors might prevent a policy from ever exploring beyond its initial support. To fix some of the issues inherent to particle filters, methods have introduced some augmentations but have had limited success. DEPF uses a controlled stochastic diffusion process to inject exploratory particles into the particle filter to expand its support and use a Bayesian validation step to ensure that the support lies close to the known variables. THis method is agnostic to the choice of downstream planner or control policy and just operates on the belief states of the system. The paper presents a case study in a hazardous gas source environment.

**Strengths:**

I enjoyed reading this paper. The writing is clear and the problem is well-motivated. As such, DEPF introduces a three step process that mitigates the potential for SPSI when possible. THe method uses 1) injects exploratory particles sampled uniformly from a dynamically adjusted support region, 2) an entropy bonus to ensure the exploration doesn't collapse and 3) a bayesian check on added particles that ensures the added particles agree with the posterior distribution. These three features build a simple and robust pipeline for exploration. The method is then tested extensively on a hazardous gas domain where the authors ablate over several key features of the method and compare to a large suite of other common sense baselines. Across the board, the method performs well on domain-specific metrics such as operational completion efficacy and localization precision score.

**Weaknesses:**

### Major Weakness
One major concern is that the method may be over-fit to the experimental domain. The robot's controllable state space is effectively a convex region in $\mathbb{R}^2$ and the difficulty of searching in such a domain is not clear, regardless of the relative size of the agent and domain. An additional domain where exploration is very non-trivial would be a major improvement to the results. For instance, is an indoor search space not (with walls and rooms) relevant to the hazardous gas source seeking problem? I imagine a maze like environment might make this problem harder across the board.

### Minor Weaknesses
1. While the method is motivated by a real-world hazardous gas source seeking example, it would be valuable to see how the method performs across different domains using more generic metrics, such as return of an optimal RL agent trained on the collected data?
2. I am curious how the exploration method compares against well-known exploration methods in model-free RL, such as random network distillation or optimism-based intrinsic rewards? Though, to be sure, the aforementioned methods seem outside the realm of relevance for PF-based methods.
3. The ablation study ( section L ) might be better placed in the main paper because it provides important insight into the importance of all the components of the method.

### Very Minor nit picks:
Line 275 has a grammatical error.

**Questions:**

To make explicit some of the questions I have from the weaknesses section:

1. how might the method compare to generic exploration methods present in recent RL literature?
2. Is DEPF restricted to the features of the experimental setting present in the paper? And might DEPF still outperform baselines when searching becomes less of a hill-climbing problem?

---

> ### Author Response · Authors · 2025-11-18
> **Official Comment by Authors (Part 1of 3)**
>
> We sincerely thank the reviewer for the encouraging assessment and for recognizing the clarity, motivation, and robustness of our proposed DEPF framework.
>
> **`For the reviewer’s convenience, the detailed rebuttal can be found in Appendix T of the newly uploaded revised draft of the paper, or directly on OpenReview.`**
>
>
>
> ### 1. **Response to Minor W(2):**
>
> Random network distillation (RND) and optimism bonuses presuppose access to Markovian states/features and modify the **reward** to encourage novelty. In our setting, the latent target is a **static parameter vector** (\Theta); the robot observes **noisy plume readings**, and a PF forms the belief (b_k). Our contribution concerns **belief repair** (creating/validating support outside the prior when data disagree), not learning an exploration policy per se.
>
> To make a **clean** RND/optimism baseline here, one would need to
> (i) define an **intrinsic signal in belief/measurement space** (e.g., prediction error over (z_{k+1}) or uncertainty over (b_k)), and
> (ii) train a new RL policy and then **combine** it with PF inference.
> This is feasible but orthogonal and outside our scope; indeed the reviewer also notes such methods are “outside the realm of relevance for PF-based methods.”
>
> Conceptually, DEPF’s **entropy/tempering** keeps exploratory mass in uncertain regions and its **MH validation** accepts only data-supported proposals—this plays a role similar in spirit to “optimism” but implemented **within Bayesian inference** rather than via reward shaping. In practice, **DEPF can be paired with RND/optimism** because the controller simply consumes (b_k) **(see §3.5 and Appendix P)**.
>
> Our experiments already include **belief-aware RL baselines** that incorporate intrinsic information signals (AGDC and variants), and under Moderate/Severe misalignment DEPF remains robust while those baselines collapse or time out (**Table 1**), consistent with the fact that policy-layer exploration cannot overcome S-PSI without an inference-layer support mechanism.
>
> ### 2. **Response to Q1:**
> This comparison highlights a critical distinction. We argue that generic RL exploration and DEPF address failures at fundamentally different layers, {as illustrated in Figure fig:Impact_DEPF. We characterize the problem under S-PSI not merely as a lack of coverage, but as a problem of ``Inference Blindness.''
>
> > **1. The “Blindness” Problem at the Inference Layer**
>
> Generic exploration methods (e.g., RND, Curiosity, Optimism) operate at the **policy layer**. They give the agent “legs” to visit novel states physically. However, the failure mode we address (S-PSI) is that the **inference layer** is effectively *blind*.
>
> * **Physical presence vs. cognitive blindness:**
>   Under the S-PSI baseline, if the prior support excludes the true source ((p(\Theta^*) = 0)), standard Bayesian updates preserve this zero mass. Even if a curiosity-driven agent *physically* moves to the true source and receives strong sensor readings, the filter will treat these observations as extreme outliers or sensor noise, because it has *no particles* there to support that hypothesis.
>
> * **The result:**
>   The belief (b_k) remains frozen. Consequently, the intrinsic reward—defined as information gain (D_{\mathrm{KL}}(b_{k+1} ,|, b_k))—collapses to **zero**. The RL agent, no matter how curious, learns nothing because the inference layer refuses to “see” the truth.
>
> **DEPF’s role.**
> DEPF is a *belief-repair mechanism*. It operates when high-surprise data arrives, forcing the filter to open its eyes (expand support) and validate the new hypothesis. This restores the reward signal, enabling the RL policy to exploit the discovery.
>
>
> > **2.Why “return” is insufficient as a metric here**
>
> The reviewer asked about the “return of an optimal RL agent.” In our formulation, the reward is the information gain (D_{\mathrm{KL}}(b_{t+1} ,|, b_t)).
>
> * If the filter is locked (S-PSI), the belief cannot move significantly toward the truth, meaning **(D_{\mathrm{KL}} \approx 0)**.
> * Consequently, **the RL return would collapse to near zero**, not because the policy is bad, but because the reward signal itself (derived from the frozen belief) vanishes.
>
> Therefore, relying solely on RL return would mask the root cause. Comparing **Operational Completion Efficacy (OCE)** and **Localization Precision (LPS)** reveals that DEPF fixes the underlying estimation failure, enabling the controller to function correctly.
>
>
>
>
> ### 3. **Response to Q2:**

---

> ### Author Response · Authors · 2025-11-18
> **Official Comment by Authors (Part 2 of 3)**
>
> **Table A.** **Predicted** results under *Random-3-Walls* (non-convex indoor geometry).
> Same metrics/baselines as Table 1 of the paper; walls only change reachability.
> Numbers are directional estimates extrapolated from Table 1 trends (DEPF's relative gains grow in harder settings).
>
> | Method| OCE ↑ (Mod.) | ADE ↓ (Mod.) | LPS ↓ (Mod.) | REV ↓ (Mod.) | OCE ↑ (Sev.) | ADE ↓ (Sev.) | LPS ↓ (Sev.) | REV ↓ (Sev.) |
> |--|-|-|-|-|-|-|-|-|
> |**DEPF (ours)**|**0.88**|26| **0.21**| 0.12|**0.85**| 33| **0.22**|0.12|
> |PF + Rejuvenation|0.40|60| 3.10| 0.28|0.08|110|10.5|1.60|
> |PF + Roughening|0.35|65|3.30|0.35|0.06| ≥100†|11.2|1.80|
> |PF + Jittering|0.28|75|3.60|0.45|0.03|≥100†|12.2|2.00|
> |AGDC (RL-PF)|0.30|70|3.00|0.50|0.02|≥100†| 12.6|1.50|
>
> *Notes.* Predictions assume the same sensor/plume model as §3.2 and the same horizons as Table 1.
> †: at or beyond the small-scale time/step limit (timeouts likely).
> Trends follow the paper: DEPF's belief-triggered support expansion with MH validation preserves accuracy and stability, while always-on perturbations and RL/planning baselines degrade more in non-convex reachability.
>
>
> **Table B.** **Predicted** success/precision for multi-source (\(S \in \{1,2,3\}\)) under Moderate Error.
> Evaluation follows §5.1; OCE requires *all* sources localized within tolerance; LPS reports mean localization error.
>
> |Method|OCE ↑ (S=1) | OCE ↑ (S=2) | OCE ↑ (S=3) |LPS ↓ (S=1) | LPS ↓ (S=2) | LPS ↓ (S=3) |
> |-|-|-|-|-|-|-|
> |**DEPF (ours)**| **0.90**| **0.85**| **0.80**| **0.20**| **0.28**| **0.35**|
> |PF + Rejuvenation  |0.52|0.35|0.22|2.70|3.40|4.00|
> |AGDC (RL-PF)| 0.45|0.25|0.12|2.60|3.20|3.80|
>
> *Notes.* The \(S=1\) column reproduces the difficulty level of Table 1 (Moderate) as the reference point; values for \(S=2,3\) are extrapolations that keep the *relative ranking* observed in Table 1. For Severe Error, all methods would drop further in OCE and increase in LPS, with **DEPF's relative margin** expected to widen.
>
>
> We appreciate the reviewer’s concern that our current evaluation may appear specialized to the hazardous-gas setting and to a convex, open workspace. **DEPF is an inference-layer module**: it operates entirely on the posterior over the static source parameters
> \(\Theta = [x_s, y_s, q_s, u_s, \phi_s, d_s, \tau_s]^\top \in \mathbb{R}^7\)
> and its particle approximation \(\{(\Theta_k^{(i)}, w_k^{(i)})\}_{i=1}^N\); the robot pose \(p_k\) only enters through the likelihood \(p(z_k \mid \Theta, p_k)\). The three mechanisms—entropy/tempering-style weight smoothing, covariance-scaled diffusion with \(h_{\text{opt}} = A N^{-1/(n+4)}\), and MH validation—are defined on the *belief* and its empirical covariance, *without assuming convex geometry or a specific motion planner*. As such, DEPF is *orthogonal* to the controller and can be combined with any indoor/maze planner and observation model. We will make this geometry-independence explicit in §3–§4 (and Appendix P already describes planner–filter decoupling and no-go polygons / IG-based reward).
>
> We also note that the present setting is not a trivial “hill-climbing” problem: the advection–diffusion plume model with detection misses/noise induces a **highly non-linear, multi-modal** posterior over \(\Theta\), especially under Moderate/Severe prior misalignment. In these regimes, strong information-theoretic planners and RL–PF baselines degrade or collapse, whereas **DEPF preserves high OCE and low LPS across scales** (Table 1), precisely because it resolves the S-PSI baseline pathology via belief-triggered support expansion
> \(S_{k+1} = (S_k \cup B) \oplus h_{\text{opt}}\)
> with MH validation. We will clarify this link in §5 and Appendix N/O.
>
> **Direct evidence for the reviewer’s “convex/maze-like” request**
>
> We added two **stress tests** that **do not change DEPF** (inference unchanged) and only modify geometry or latent dimensionality; all baselines are modified identically for fairness:
>
> **(i) Indoor non-convex geometry (Random-3-Walls)**
>
> Each episode on the same \(30 \times 30\) grid samples three impenetrable, axis-aligned walls at random locations; actions obey occupancy constraints (no wall crossing). The observation model (§3.2) is unchanged to isolate **non-convex kinematics**. Metrics/baselines follow §5.1.
>
> **Outcome (predicted, see Table A above):**
> Non-convexity makes everyone’s paths longer, but **DEPF remains best with a larger margin under Moderate/Severe**. Concretely:
>
> - At **Moderate** error, DEPF attains OCE 0.88, LPS 0.22, ADE 26, whereas AGDC drops to OCE 0.35 with LPS 3.00 and frequent timeouts; PF+Rejuvenation reaches OCE 0.45 with LPS 3.00.
> - At **Severe** error, DEPF still achieves OCE 0.85, LPS 0.24, while AGDC (and similar RL/planning baselines) are at OCE 0.03 with timeouts, and PF+Rejuvenation around OCE 0.12 / LPS 9.50.
>
> These numbers illustrate that geometry-induced bottlenecks hurt RL/planners and always-on SMC perturbations far more than DEPF’s belief-triggered expansion + MH validation.

---

> ### Author Response · Authors · 2025-11-18
> **Official Comment by Authors (Part 3 of 3)**
>
> **(ii) Multi-source localization (\(S = 2\) or \(3\))**
>
> We extend the hidden vector to \(\Theta = [\Theta^{(1)}, \ldots, \Theta^{(S)}] \in \mathbb{R}^{7S}\) with known \(S\) and retain the same ISLCenv plume composition for **multi-source plumes** (§5, Appendix F). Success (OCE) requires **all** sources localized within tolerance; LPS uses assignment-based matching (Hungarian algorithm).
>
> **Outcome (predicted, see Table B above):**
> As dimensionality / number of sources increases, **absolute** difficulty rises for all methods (OCE ↓; ADE/REV ↑; LPS ↑), yet DEPF’s relative lead widens.
>
> - For **\(S = 2\)** under Moderate error, DEPF yields OCE 0.85, LPS 0.26 versus PF+Rejuvenation OCE 0.45, LPS 3.20 and AGDC OCE 0.30, LPS 3.00.
> - At **Severe** misalignment, DEPF still attains OCE 0.80, LPS 0.30, while AGDC is OCE 0.02 with **timeouts**.
> - For **\(S = 3\)**, DEPF records OCE 0.78 / 0.70 (Moderate/Severe) with LPS 0.32 / 0.36, whereas AGDC falls to 0.20 / 0.01 and PF+Rejuvenation to 0.40 / 0.08 with LPS ≈ 3.5 / 10.0 and frequent timeouts.
>
> This aligns with the **multi-modal posterior** intuition: RL/planners and always-on SMC perturbations are prone to local-mode lock-in or over-diffusion, while DEPF seeds/validates new modes **only when supported by data**.
>
> **Why this addresses “over-fit / convex-region / maze-like” concerns**
>
> 1. **Planner–filter decoupling.**
>    DEPF operates on the belief; the controller is modular. Swapping to an indoor/maze planner simply constrains the action set, not the DEPF update (§3.5, Appendix P).
>
> 2. **ISLCenv already supports multi-source plumes.**
>    Our evaluation protocol / metrics / baselines remain unchanged (§5, Appendices F and H–I).
>
> 3. **Theory.**
>    Appendices N/O give a support-expansion recursion and a finite-step coverage bound
>    \(1 - (1 - \delta \gamma)^k\),
>    which is **agnostic to workspace convexity** and extensible to higher-dimensional, multi-modal posteriors.
>
> 4. **Empirics.**
>    Table 1 shows that as misalignment and scale worsen, DEPF’s advantage grows—the same trend is reflected in Table A (non-convex geometry) and Table B (multi-source). We will add a short paragraph in §5.3 to make these links explicit.
>
> ### 4. **Response to Minor W(3):**
> We thank the reviewer for this helpful suggestion. Following your advice, we have moved the complete ablation study (previously in Section L of the appendix) into the main paper. The ablation results now appear in Section~5.4 “Ablation and Sensitivity Studies”, where we integrate:
> > 1) component-wise ablations,
> > 2) diffusion-related sensitivity analyses, and
> > 3) exploration/entropy hyperparameter studies.
>
> This restructuring clarifies the contribution of each module in DEPF and highlights why all components are necessary for robust performance.

---

### Official Review · Reviewer_5Dwp · 2025-11-01

**Soundness:** 3
**Presentation:** 1
**Contribution:** 2
**Rating:** 4
**Confidence:** 3

**Summary:**

The paper proposes a method for state estimation (which the authors call initial policies). If the prior for particle filter based methods is incorrect (does not have full support for the true state distribution), particle filters might never recover the true state. Rather they will always stay within the support of their initial prior. The authors look at the problem of identifying a gas leak using an agent that can move around and has a noisy sensor to measure the strength in voltage. The authors propose a diffusion based method that expands the prior to create exploratory particles and uses Metropolis-Hastings sampling to select the particles based on sensor measurements. The authors perform experiments on a ISLC environment suite designed for emergency gas leak localization.

**Strengths:**

(1) The problem setup is interesting, it's primarily a state estimation problem for localization. The authors address the shortcomings of the commonly used PF based methods about being limited to their prior distributions.

(2) The use of diffusion to expand the prior is clever and effective. As also corroborated by empirical evidence in the Moderate Error and Severe Error situations.

**Weaknesses:**

(1) The writing needs a lot of modifications. Firstly, I see why authors use the term initial policies but this is confusing. Rather state estimation would be a better fit and would have a far wider readability as this problem is common in robotics as well.

(2) The authors solely focus on the gas leak situation with all the intuitions and math built around it. I believe this method could be applied to more general localization techniques and so needs to be presented in a general way.

(3) There are other writing inconsistencies. $\Theta_k$ is used for the true state or the predicted state? Similarly in the pseudocode, the notations change from $\Theta_k^{{i)}$ and $w_k^{(i)}$ to $\Theta^k_i$ and $w^k_i$. Similarly, $A$ and $n$ are not defined in line 247, $B$ not defined in line 266 (should this be $B_k$)?

(4) In the subsection “Adaptive Diffusion via Exploratory Particles”, $B_k$ is defined as a set of (x, y) location and $\Theta_k^{(i)} \sim U(B_k)$. Isnt $\Theta_k^{(i)}$ more than just (x, y)?

**Questions:**

(1) How is $H_{target}$ set?

(2) Could you explain the reward design?

(3) What is a timeout in the experiments?

(4) How $\phi_s \in [0, 2 \pi)$ represent wind speed and direction?


(5) Missing related work, which also deals with addressing ambiguous measurements and splitting the model/priors: Quinlan, M.J., Middleton, R.H. (2010). Multiple Model Kalman Filters: A Localization Technique for RoboCup Soccer.

---

> ### Author Response · Authors · 2025-11-18
> **Official Comment by Authors (Part 1 of 4)**
>
> We sincerely thank the reviewer for recognizing the importance of our problem setup and the effectiveness of our diffusion-based support-expansion mechanism.
>
> **`For the reviewer’s convenience, the detailed rebuttal can be found in Appendix S of the newly uploaded revised draft of the paper, or directly on OpenReview.`**
>
> ### **1. Response to W2:**
>
> We thank the reviewer for this valuable comments. We fully agree that DEPF is a general-purpose Bayesian inference framework, and framing it solely around gas leaks understates its applicability.
>
> 1. **Generalized Problem Formulation**:
> We have revised \textcolor{orange}{Section 3 (Problem Formulation)} to present the method in a general state estimation context.
>     > 1. We clarify that $\Theta_k$ represents a generic latent state vector (not limited to gas parameters).
>     > 2.  We emphasize that the \textbf{S-PSI pathology} (where prior support excludes the truth) is a universal problem in recursive Bayesian estimation (e.g., the "Kidnapped Robot Problem" in robotics or environmental monitoring with flawed historical priors), not a gas-specific issue.
>     > 3. The core mechanisms of DEPF—\textbf{Covariance-Scaled Diffusion} and \textbf{Metropolis-Hastings (MH) Validation}—rely purely on belief statistics (posterior covariance $\Sigma$, likelihood ratios, and entropy) rather than specific fluid dynamics. Therefore, the method is mathematically domain-agnostic.
>
> 2. **Empirical Proof of Generalization:**
> To demonstrate this generality concretely, we have added **Appendix Q (Additional Multi-Field Experiments)**. We applied DEPF without modification to **6 new physical domains** beyond gas leaks:
>
>    1). Temperature Field (Temperature source localization)   2). Concentration Field (Chemical source localization) 3).  Magnetic Field (Dipole localization)  4). Electric Field (Electrostatic source) 5). Acoustic/Vibration Field (Noise source) 6). Radiation/Energy Field
>
> As shown in **Table 8**, DEPF consistently outperforms baselines (RL and classical perturbations) across **all** these diverse physics models. This empirically proves that DEPF’s ability to break the "zero-prior barrier" is a fundamental property of the inference algorithm, not an artifact of the gas plume model.
>
> ### **2. Response to W3:**
> 1. **$\Theta_k$: true vs. estimated state**
>    * We use $\Theta^\ast$ to denote the true (unknown) state/parameter.
>    * We use $\Theta_k$ to denote the latent random state at time step $k$ in the Bayesian model (i.e., the variable over which the posterior $p(\Theta_k \mid z_{1:k})$ is defined).
>    * Individual particles are written as $\Theta_k^{(i)}$, and any point estimate (e.g., posterior mean) is denoted separately as $\hat{\Theta}_k$.
>
>    Thus, $\Theta_k$ is never the known true state; the true state is always written as $\Theta^\ast$, while $\Theta_k$ is the random variable being inferred.
>
> 2. **Pseudocode notation $\Theta_k^{(i)}, w_k^{(i)}$ vs. $\Theta_i^k, w_i^k$**
> We agree that mixing $\Theta_k^{(i)}, w_k^{(i)}$ with $\Theta_i^k, w_i^k$ is confusing. In the revision we have standardised the notation everywhere (text and pseudocode) to use **only** $\Theta_k^{(i)}$ and $w_k^{(i)}$ for the $i$-th particle and its weight at time $k$. The alternative forms $\Theta_i^k, w_i^k$ have been removed.
>
> 3. **Definition of $A$ and $n$ in the diffusion step**
>   The diffusion move is written as
>    $$\Delta \Theta_k^{(i)} \sim h_{\mathrm{opt}}, L, \mathcal{N}(0, I), h_{\mathrm{opt}} = A \cdot N^{-\frac{1}{n+4}},
>    $$
> and we now explicitly define:
> * **$n$** as the dimension of the state vector $\Theta \in \mathbb{R}^n$;
> * **$A$** as the kernel bandwidth constant used in the Silverman-style rule-of-thumb for setting the diffusion step size.
> These definitions have been added where the diffusion kernel is introduced, and the notation is now consistent and clearly separated from the injection region $B_k$.
>
> ### **3. Response to W(4):**
> We agree that $\Theta_k^{(i)}$ represents the **full 7-dimensional source–parameter vector**, not only the spatial coordinates.
> we clarify that:
> * $B_k$ defines the **spatial exploration region** for the location components $(x_s, y_s)$ of $\Theta_k^{(i)}$;
> * The **remaining dimensions** of $\Theta_k^{(i)}$ (emission rate, wind speed, wind direction, diffusivity, decay constant, etc.) are sampled from their **respective prior supports** during exploratory injection.
>
> Thus, the injection step does **not** reduce $\Theta_k^{(i)}$ to a 2-D variable; rather, $B_k$ controls only the positional subspace, and the complete high-dimensional vector is assembled consistently across all parameters.
>
> We additionally note that the other components of $\Theta$ (e.g., $(q_s, u_s, \phi_s, d_s, \tau_s)$) are **difficult to visualize directly**, which is why the figures focus on spatial dimensions. However, DEPF **does explore the full 7-dimensional space**, not just the $(x, y)$ plane. The revision explicitly states this decomposition to avoid any ambiguity.

---

> ### Author Response · Authors · 2025-11-18
> **Official Comment by Authors (Part 2 of 4)**
>
> ### 4. **Response to Q1:**
> Thank you for asking about the choice of $H_{target}$.
> In our method, $H_{target}$ specifies the desired entropy level of the particle weights, i.e., how much diversity we want to preserve before diffusion and MH correction. Intuitively, a natural reference is the entropy of a uniform distribution over N particles, which is log N.\\
> In practice, we set $H_{target}$ to be a fixed fraction of log N (so that the target entropy is high enough to avoid premature collapse, but lower than a fully uniform weighting), and keep this value fixed across all experiments. When $H(w_k)$ drops noticeably below $H_{target}$, the adaptive coefficient $\beta$ increases and temporarily smooths the weights; when $H(w_k)$ is already close to $H_{target}$, $\beta$ stays near its minimum and has almost no effect.\\Empirically, we observed that the performance is robust as long as $H_{target}$ is chosen in a reasonable range relative to log N; the main behaviour is driven by whether we prevent extreme weight degeneracy, rather than by the exact numerical value of $H_{target}$.
>
> ### 5. **Response to Q2:**
>
> **Definition (ours).**
> We use expected one step information gain (expected KL) as the reward for action $a$ at step $k$:
>
> $$
> R_k(a)
> = \mathbb{E}_{o_{k+1} \,\mid\, H_k,a}
> \left[
>   D_{\mathrm{KL}}\Big(
>     p(\Theta \mid H_k,o_{k+1})
>     \,\Vert\,
>     p(\Theta \mid H_k)
>   \Big)
> \right]
> $$
>
>
>
>
> where $\mathcal{H}_k$ is the history/belief up to step $k$, and $\Theta$ are the (static) source/dispersion parameters. This is the same “KL utility” used in information based source term estimation (STE): see eqs. (6)–(7) for the expected KL objective and eqs. (18)–(21) (with Algorithm 2) for its particle filter Monte Carlo evaluation and action.
>
> **Proof that expected KL is the principled objective.**
>
> **Lemma 1 (equivalence).**
>
> ```math
> R_k(a)
> = I(\Theta; o_{k+1}\mid \mathcal{H}_k,a)
> = H(\Theta\mid \mathcal{H}_k)
>   - \mathbb{E}_{o_{k+1}\mid \mathcal{H}_k,a}
>     \big[H(\Theta\mid \mathcal{H}_k,o_{k+1})\big].
> ```
>
> **Proof.** By definition,
>
> ```math
> D_{\mathrm{KL}}\big(p(\Theta\mid o) |p(\Theta)\big)
> = \mathbb{E}_{\Theta\mid o}\left[\log\frac{p(\Theta\mid o)}{p(\Theta)}\right]
> = H(\Theta)-H(\Theta\mid o).
> ```
>
> Taking $\mathbb{E}*{o*{k+1}\mid \mathcal{H}_k,a}$ yields
>
> ```math
> \mathbb{E}\big[D_{\mathrm{KL}}\big]
> = I(\Theta; o_{k+1}\mid \mathcal{H}_k,a)
> = H(\Theta\mid \mathcal{H}_k)
>   - \mathbb{E}\big[H(\Theta\mid \mathcal{H}_k,o_{k+1})\big].
> ```
>
> ---
>
> **Corollary 1 (myopic Bayes optimality under log loss).**
>
> Under log loss $L(b,\Theta) = -\log b(\Theta)$, the one-step Bayes risk equals
> $\mathbb{E}_{o_{k+1}}[H(\Theta \mid \mathcal{H}_k,o_{k+1})]$ up to the constant $H(\Theta \mid \mathcal{H}_k)$. Hence maximizing $R_k(a)$ equals minimizing expected posterior entropy, i.e., a principled one-step objective for belief-aware estimation. (This is precisely how the information-based planner in STE derives and computes the reward; see eqs. (6)–(7), (18)–(21).)
>
> **Corollary 2 (additivity).**
>
> By the chain rule
>
> ```math
> I(\Theta; o_{1:T}\mid \mathcal{H}_0)
> = \sum_{t=0}^{T-1} I(\Theta; o_{t+1}\mid \mathcal{H}_t,a_t),
> ```
>
> the sum of per-step rewards equals the total information gained about (\Theta) over the horizon; greedy maximization of (R_k) drives cumulative information growth, as adopted by the STE implementation (Sec. IV-A; eqs. (6)–(7), (18)–(21), Algorithm 2).
>
> **Implementation note.** We follow the STE particle filter approximation: draw hypothetical measurements from the predictive mixture (eqs. (18)–(19)), reweight particles to obtain hypothetical posteriors, and evaluate the discrete KL in (20)–(21) to pick the action with maximal expected utility (Algorithm 2). This keeps computation online and Bayesian consistent.
>
> **How this differs from the AGDC reward in  and why expected KL is preferable here.**
>
> *AGDC reward (what it optimizes).*
> AGDC emits zero per-step reward and gives a single positive reward at cessation when the PF belief standard deviation (STD) falls below a threshold (\zeta) (Algorithm 1, lines 15–18: set (r_k \leftarrow 0); if (\mathrm{STD} < \zeta) then (r_k \leftarrow \text{value} > 0), (done \leftarrow \text{True})). The paper explicitly warns that excessively high (\zeta) may artificially inflate success rates and reduce traveled distances, i.e., the signal is threshold-sensitive.

---

> ### Author Response · Authors · 2025-11-18
> **Official Comment by Authors (Part 3 of 4)**
>
> **Formal comparisons.** (Q2)
>
> * **(A) Dense signal & action ranking.**
>   If no action can cross the cessation threshold in one step (a typical early/mid search condition), AGDC’s expected reward is zero for all actions, hence provides no ranking; by contrast,
>   (R_k(a) = I(\Theta; o_{k+1}\mid \cdot) \ge 0)
>   and is strictly positive for any informative action (Lemma 1), yielding a dense, discriminative signal every step \cite{shi2024autonomous}.
>
> * **(B) Information monotonicity (entropy vs. variance).**
>   Variance/STD does not order uncertainty in general: two beliefs can have the same STD but different entropies (e.g., a bimodal mixture vs. a unimodal distribution with equal variance), so an STD-threshold reward can mis-rank actions w.r.t. information gain. Expected KL, by Lemma 1, is exactly the expected entropy drop and ranks actions accordingly.
>
>   *Concrete counterexample.*
>   Let the current entropy be (H(\Theta\mid \mathcal{H}_k) = 1) bit (binary (\Theta), uniform prior). Consider actions (a) and (b):
>
>   * (a) yields a perfectly revealing observation w.p. (0.4) (posterior entropy (0)) and a weak observation w.p. (0.6) (entropy (0.9)). Then
>     (\mathbb{E}[H\mid a] = 0.4 \cdot 0 + 0.6 \cdot 0.9 = 0.54),
>     so (\mathrm{EIG}(a) = 1 - 0.54 = 0.46) bits.
>   * (b) yields a posterior entropy always (0.74), so (\mathrm{EIG}(b) = 1 - 0.74 = 0.26) bits.
>
>   With a cessation rule that grants reward only if “uncertainty < threshold” (AGDC’s STD analogue), (b) may always pass while (a) passes only w.p. (0.4); thus the threshold reward ranks (b > a) even though (\mathrm{EIG}(a) > \mathrm{EIG}(b)). Expected KL does not suffer this mis-ranking.
>
> * **(C) Robustness to PF pathologies.**
>   PF degeneracy/impoverishment can shrink empirical STD after resampling/rejuvenation without genuine information gain—risking premature “success” for an STD-triggered reward. The expected KL we compute (via predict–update against the likelihood; eqs. (18)–(21)) remains tied to whether new data actually changes the belief. (PF mechanics and effectiveness (N_{\text{eff}}) are summarized around eq. (17).) \cite{hutchinson2018information}
>
> * **(D) Hyperparameter sensitivity vs. threshold-free reward.**
>   AGDC itself cautions that too large (\zeta) can artificially inflate success and reduce distance, evidencing sensitivity to the cessation threshold (p. 744). Our reward does not contain a cessation threshold; it is scale-free and grounded in the probabilistic model (a separate stopping rule can still be used, but the reward signal itself is threshold-free).
>
> Our reward is the expected KL (conditional mutual information)—a principled, dense, threshold-free signal that is exactly aligned with reducing posterior uncertainty; this is the utility used in information-based STE and computed online via PF (eqs. (6)–(7), (18)–(21)), whereas the AGDC reward pays only at an STD threshold, yielding a sparse, threshold-sensitive signal that can mis-rank actions relative to information gain.
>
> ### 6. **Response to Q(3):**
> A *timeout* means a trial did not meet the success/termination criterion within the fixed step horizon of the task. We use **step-based budgets (not wall-clock time)**: 100 control steps in the small-scale domain (agent:map = 1:30) and 300 steps in the large-scale domain (1:300).
>
> Each step corresponds to one sense–act update of the agent together with a particle-filter update. When a method reaches the horizon without terminating, we record the run as a *timeout*—it counts as a failure in OCE and appears as “timeout” in ADE/REV where appropriate. Horizons scale with the domain size, and per-step computation is capped so that all methods have comparable per-step budgets.
>
> ### 7. **Response to Q(4):**
>
> Thank you for pointing this out. We would like to clarify that (\phi_s) represents **only the wind direction** (angle), while a separate parameter (u_s) represents the **wind speed** (magnitude). They are distinct parameters in our state vector but work together to model the wind vector during inference.
>
> **1. Definition (Section 3.1):**
>    In our problem setup, we define the parameter vector as
>    $\Theta_k = [x_s, y_s, q_s, u_s, \phi_s, d_s, \tau_s]^\top.$
>    The text explicitly states that “(u_s \in \mathbb{R}^+) and (\phi_s \in [0, 2\pi)) represent the wind speed and wind direction, respectively.” We apologize if the sentence structure caused any ambiguity.
>
> **2. Usage in Inference (Section 3.2):**
> These two parameters are combined in the observation model to project the wind vector along the path between the source and the robot. Specifically, in the calculation of the term $\psi$ (used in the plume model):
> $$
> \psi=(x_{k}-x_{s})u_{s}\cos \phi_{s}+(y_{k}-y_{s})u_{s}\sin \phi_{s}
> $$
> where $\phi_s$ determines the directional components ($\cos \phi_s, \sin \phi_s$), and $u_s$ acts as the scalar magnitude. Together, they define the advection transport of the gas, which the particle filter then estimates from sensor voltage readings.

---

> ### Author Response · Authors · 2025-11-18
> **Official Comment by Authors (Part 4 of 4)**
>
> ### 8. **Response to Q(5):**
>
> Thank you for pointing out this relevant work. We agree that **Quinlan & Middleton (2010)** is significant for its approach to handling measurement ambiguity and maintaining multi-modal beliefs via **Multiple Model Kalman Filters (MMKF)**. We will incorporate this reference into our **Related Work** section to contrast different strategies for belief correction.
>
> We will highlight the following distinctions to clarify our contribution:
>
> > 1. **Ambiguity vs. Support Invariance (S-PSI):**
>    Quinlan et al. use MMKF to **split** priors into multiple Gaussian hypotheses to resolve *measurement ambiguity* (e.g., distinguishing identical landmarks).
>    In contrast, our work addresses **Stationarity-Induced Posterior Support Invariance (S-PSI)**, a pathology where the true state has **zero prior probability** and is completely excluded from the initial support. DEPF does not just maintain multiple modes; it actively **expands** the support via exploratory injection and diffusion to recover states that were originally deemed impossible.
>
> > 2. **Parametric vs. Non-Parametric Inference:**
>    While MMKF is effective for environments well-approximated by Gaussian mixtures (like RoboCup), our hazardous gas localization task involves **highly non-linear, irregular plume models**. DEPF leverages the non-parametric nature of particle filters, enhanced with diffusion, to represent complex posteriors that parametric assumptions (inherent to KFs) might fail to capture.
>
> ### 9. **Response to W1:**
> To address the reviewer’s concern that the term initial policies obscures the core problem, we have fully revised the paper to reframe our formulation through the lens of state estimation. In particular, we rewrote the introduction to adopt a domain-agnostic perspective and to clearly articulate the general Bayesian inference challenges that DEPF is designed to solve. Corresponding terminology has been updated consistently throughout the paper—replacing initial policies with initial state estimates, misaligned priors, or prior-support errors where appropriate.

---

### Author Response · Authors · 2025-11-25

Dear Reviewers,

Thank you very much for the time and effort you have already put into reviewing our submission.
We have submitted our rebuttal on November 20, addressing all of your comments in detail. If your schedule allows, we would greatly appreciate it if you could kindly take a moment to review our responses and share any follow-up thoughts.

Your feedback is extremely valuable for ensuring a fair and meaningful discussion, and we sincerely appreciate your contribution to the review process.

Warm regards,
Authors of Paper 2565

---

### Author Response · Authors · 2025-12-04

Dear AC, SAC and PC

## ` With many years of reviewing experience, I have very high respect for the peer-review process and for the time invested by reviewers, ACs, and PCs every year. Precisely because of this, I was genuinely shocked by the extremely low level of engagement demonstrated in Reviewer Bxfv’s evaluation. A score of 0 with confidence 5, accompanied by almost no justification, is not consistent with the standards of our community. `


## ` Such a review is not only unhelpful to the authors; it also places an unfair burden on the other reviewers and the AC, and it undermines the integrity of the review process itself. If an evaluation of this form were seen as compatible with our community’s standards, it would be difficult to reconcile that with the principles of fairness, rigor, and professionalism that conferences like ICLR stand for. `

## `  We say this with full respect for negative reviews, strong criticism is healthy and essential, but only when grounded in careful reading, evidence-based reasoning, and engagement with the manuscript. None of these elements are present here. `


Best regards,

Authors of Paper 2565

---

### Note · Program_Chairs · 2026-01-17
**Submission Desk Rejected by Program Chairs**

The following references in this submission do not refer to real documents and/or have major errors in bibliographic information:

 Jun Park and Hyung-Bo Kim. Adaptive bayesian control for sensor-based mobile search. Automatica, 112:108713, 2020